# Approximately Equivariant Graph Networks

**Ningyuan (Teresa) Huang**
Johns Hopkins University
`nhuang19@jhu.edu`

**Ron Levie**
Technion – Israel Institute of Technology
`levieron@technion.ac.il`

**Soledad Villar**
Johns Hopkins University
`svillar3@jhu.edu`

## Abstract

Graph neural networks (GNNs) are commonly described as being permutation equivariant with respect to node relabeling in the graph. This symmetry of GNNs is often compared to the translation equivariance of Euclidean convolution neural networks (CNNs). However, these two symmetries are fundamentally different: The translation equivariance of CNNs corresponds to symmetries of the fixed domain acting on the image signals (sometimes known as *active symmetries*), whereas in GNNs any permutation acts on both the graph signals and the graph domain (sometimes described as *passive symmetries*). In this work, we focus on the active symmetries of GNNs, by considering a learning setting where signals are supported on a fixed graph. In this case, the natural symmetries of GNNs are the automorphisms of the graph. Since real-world graphs tend to be asymmetric, we relax the notion of symmetries by formalizing approximate symmetries via graph coarsening. We present a bias-variance formula that quantifies the tradeoff between the loss in expressivity and the gain in the regularity of the learned estimator, depending on the chosen symmetry group. To illustrate our approach, we conduct extensive experiments on image inpainting, traffic flow prediction, and human pose estimation with different choices of symmetries. We show theoretically and empirically that the best generalization performance can be achieved by choosing a suitably larger group than the graph automorphism, but smaller than the permutation group.

## 1 Introduction

Graph Neural Networks (GNNs) are popular tools to learn functions on graphs. They are commonly designed to be permutation equivariant since the node ordering can be arbitrary (in the matrix representation of a graph). Permutation equivariance serves as a strong geometric prior and allows GNNs to generalize well [1–3]. Yet in many applications, the node ordering across different graphs is matched or fixed a priori, such as a time series of social networks where the nodes identify the same users, or a set of skeleton graphs where the nodes represent the same joints. In such settings, the natural symmetries arise from graph automorphisms, which effectively only act on the graph signals; This is inherently different from the standard equivariance in GNNs that concerns all possible permutations acting on both the signals and the graph domain. A permutation of both the graph and the graph signal can be seen as a *change of coordinates* since it does not change the object it represents, just the way to express it. This parallels the *passive symmetries* in physics, where physical observables are independent of the coordinate system one uses to express them [4, 5]. In contrast, a permutation of the graph signal on a fixed graph potentially transforms the object itself, not only its representation. This parallels the *active symmetries* in physics, where the coordinate system (in this case, the graph or domain) is fixed but a group transformation on the signal results in a predictable

37th Conference on Neural Information Processing Systems (NeurIPS 2023).

transformation of the outcome (e.g., permuting left and right joints in the human skeleton changes a left-handed person to right-handed). The active symmetries on a fixed graph are similar to the translation equivariance symmetries in Euclidean convolutional neural networks (CNNs), where the domain is a fixed-size grid and the signals are images.

In this work, we focus on active symmetries in GNNs. Specifically, we consider a fixed graph domain $G$ with $N$ nodes, an adjacency matrix $A \in \mathbb{R}^{N \times N}$, and input graph signals $X \in \mathbb{R}^{N \times d}$. We are interested in learning equivariant functions $f$ that satisfy (approximate) active symmetries

$$f(\Pi X) \approx \Pi f(X) \text{ for permutations } \Pi \in \mathcal{G} \subseteq \mathcal{S}_N, \tag{1}$$

where $\mathcal{G}$ is a subgroup of the permutation group $\mathcal{S}_N$ that depends on $G$. For example, $\mathcal{G}$ can be the graph automorphism group $\mathcal{A}_G = \{\Pi : \Pi A = A\Pi\}$. Each choice of $\mathcal{G}$ induces a hypothesis class $\mathcal{H}_\mathcal{G}$ for $f$: the smaller the group $\mathcal{G}$, the larger the class $\mathcal{H}_\mathcal{G}$. We aim to select $\mathcal{G}$ so that the learned function $f \in \mathcal{H}_\mathcal{G}$ generalizes well, also known as the *model selection* problem [6, Chp.4]. In contrast, standard graph learning methods (GNNs, spectral methods) are defined to satisfy passive symmetries, by treating $A$ as input and requiring $f(\Pi A \Pi^\top, \Pi X) = \Pi f(A, X)$ for all permutations $\Pi \in \mathcal{S}_N$ and all $N$. But for the fixed graph setting, we argue that active symmetries are more relevant. Thus we use $A$ to define the hypothesis class rather than treating it an input. By switching from passive symmetries to active symmetries, we will show how to design GNNs for signals supported on a fixed graph with different levels of expressivity and generalization properties.

While enforcing symmetries has been shown to improve generalization when the symmetry group is known a priori [7–13], the problem of *symmetry model selection* is not completely solved, particularly when the data lacks exact symmetries (see for instance [14–16] and references therein). Motivated by this, we study the symmetry model selection problem for learning on a fixed graph domain. This setting is particularly interesting since (1) the graph automorphism group serves as the natural *oracle* symmetry; (2) real-world graphs tend to be asymmetric, but admit cluster structure or local symmetries. Therefore, we define *approximate symmetries of graphs* using the cut distance between graphs from graphon analysis. An approximate symmetry of a graph $G$ is a symmetry of any other graph $G'$ that approximates $G$ in the cut distance. In practice, we take $G'$ as coarse-grainings (or clusterings) of $G$, as these are typically guaranteed to be close in cut distance to $G$. We show how to induce approximate symmetries for $G$ via the automorphisms of $G'$. Our main contributions include:

1. We formalize the notion of active symmetries and approximate symmetries of GNNs for signals supported on a fixed graph domain, which allows us to study the symmetry group model selection problem. (See Sections 2, 4)

2. We theoretically characterize the statistical risk depending on the hypothesis class induced from the symmetry group, and show a bias-variance tradeoff between the reduction in expressivity and the gain in regularity of the model. (See Sections 3, 4)

3. We illustrate our approach empirically for image inpainting, traffic flow prediction, and human pose estimation. (See Section 5 for an overview, and Appendix D for the details on how to implement equivariant graph networks with respect to different symmetry groups).

## 1.1 Related Work

**Graph Neural Networks and Equivariant Networks.** Graph Neural Networks (GNNs) [17–19] are typically permutation-equivariant (with respect to node relabeling). These include message-passing neural networks (MPNNs) [19–21], spectral GNNs [18, 22–24], and subgraph-based GNNs [25–30]. Permutation equivariance in GNNs can extend to edge types [31] and higher-order tensors representing the graph structure [32, 33]. Equivariant networks generalize symmetries on graphs to other objects, such as sets [34], images [35], shapes [36, 37], point clouds [38–42], manifolds [1, 43, 44], and physical systems [45, 46] among many others. Notably, many equivariant machine learning problems concern the setting where the domain is fixed, e.g., learning functions on a fixed sphere [47], and thus focus on active symmetries rather than passive symmetries such as the node ordering in standard GNNs. Yet learning on a fixed graph domain arises naturally in many applications such as molecular dynamics modeling [48]. This motivates us to consider active symmetries of GNNs for learning functions on a fixed graph. Our work is closely related to Natural Graph Networks (NGNs) in [49], which use global and local graph isomorphisms to design maximally expressive GNNs for distinguishing *different graphs*. In contrast, we focus on generalization and thus consider symmetry model selection on a *fixed* graph.

**Generalization of Graph Neural Networks and Equivariant Networks.** Most existing works focus on the graph-level tasks, where in-distribution generalization bounds of GNNs have been derived using statistical learning-theoretic measures such as Rademacher complexity [50] and Vapnik–Chervonenkis (VC) dimension [51, 52], or uniform generalization analysis based on random graph models [53]; Out-of-distribution generalization properties have been investigated using different notions including transferability [54, 55] (also known as size generalization [56]), and extrapolation [57]. In general, imposing symmetry constraints improves the sample complexity and the generalization error [7–13]. Recently, Petrache and Trivedi [58] investigated approximation-generalization tradeoffs using approximate symmetries for general groups. Their results are based on uniform convergence generalization bounds, which measure the worst-case performance of all functions in a hypothesis class and differ from our non-uniform analysis.

**Approximate Symmetries.** For physical dynamical problems, Wang et al. [45] formalized approximate symmetries by relaxing exact equivariance to allow for a small equivariance error. For reinforcement learning applications, Finzi et al. [59] proposed Residual Pathway Priors to expand network layers into a sum of equivariant layers and non-equivariant layers, and thus relax strict equivariance into approximate equivariance priors. In the context of self-supervised learning, Suau et al. [60], Gupta et al. [61] proposed to learn structural latent representations that satisfy approximate equivariance. Inspired by randomized algorithms, Cotta et al. [62] formalizes probabilistic notions of invariances and universal approximation. In [63], scattering transforms (a specific realization of a CNN) are shown to be approximately invariant to small deformations in the image domain, which can be seen as a form of approximate symmetry. Scattering transforms on graphs are discussed in [64–66]. Additional discussions on approximate symmetries can be found in [67, Section 6].

## 2   Problem Setup: Learning Equivariant Maps on a Fixed Graph

**Notations.** We let $\mathbb{R}, \mathbb{R}_+$ denote the reals and the nonegative reals, $I$ denote the identity matrix and $\mathbb{1}$ denote the all-ones matrix. For a matrix $Y \in \mathbb{R}^{N \times k}$, we write the Frobenious norm as $\|Y\|_F$. We denote by $X \odot Y$ the element-wise multiplication of matrices $X$ and $Y$. We write $[N] = \{1, \dots, N\}$ and $\mathcal{S}_N$ as the permutation group on the set $[N]$. Groups are typically noted by calligraphic letters. Given $\mathcal{H}, \mathcal{K}$ groups, we denote a semidirect product by $\mathcal{H} \rtimes \mathcal{K}$. For a group $\mathcal{G}$ with representation $\phi$, we denote by $\chi_{\phi|_\mathcal{G}}$ the corresponding character (see Definition 6 in Appendix A.1). Denoting $\mathcal{H} \leq \mathcal{G}$ means that $\mathcal{H}$ is a subgroup of $\mathcal{G}$.

**Equivariance.** We consider a compact group $\mathcal{G}$ with Haar measure $\lambda$ (the unique $\mathcal{G}$-invariant probability measure on $\mathcal{G}$). Let $\mathcal{G}$ act on spaces $\mathcal{X}$ and $\mathcal{Y}$ by representations $\phi$ and $\psi$, respectively. We say that a map $f : \mathcal{X} \to \mathcal{Y}$ is $\mathcal{G}$-equivariant if for all $g \in \mathcal{G}, x \in \mathcal{X}, \psi(g^{-1})f(\phi(g)\, x) = f(x)$. Given any map $f : \mathcal{X} \to \mathcal{Y}$, a projection of $f$ onto the space of $\mathcal{G}$-equivariant maps can be computed by averaging over orbits with respect to $\lambda$

$$(\mathcal{Q}_\mathcal{G} f)(x) = \int_\mathcal{G} \psi(g^{-1})\, f\left(\phi(g)x\right)\, \mathrm{d}\lambda(g). \tag{2}$$

For $u, v \in \mathcal{Y}$, let $\langle u, v \rangle$ be a $\mathcal{G}$-invariant inner product, i.e. $\langle \psi(g)u, \psi(g)v \rangle = \langle u, v \rangle$, for all $g \in \mathcal{G}$, for all $u, v \in \mathcal{Y}$. Given two maps $f_1, f_2 : \mathcal{X} \to \mathcal{Y}$, we define their inner product as $\langle f_1, f_2 \rangle_\mu = \int_\mathcal{X} \langle f_1(x), f_2(x) \rangle\, \mathrm{d}\mu(x)$, where $\mu$ is a $\mathcal{G}$-invariant measure on $\mathcal{X}$. Let $V$ be the space of all (measurable) map $f : \mathcal{X} \to \mathcal{Y}$ such that $\|f\|_\mu = \sqrt{\langle f, f \rangle_\mu} < \infty$.

**Graphs.** We consider edge-node weighted graphs $G = ([N], A, b)$, where $[N]$ is a finite set of nodes, $A \in [0, 1]^{N \times N}$ is the adjacency matrix describing the edge weights, and $b = \{b_1, \dots, b_N\} \subset \mathbb{R}$ are the node weights. An edge weighted graph is a special case of $G$ where all node weights are 1. A simple graph is a special case of an edge weighted graph, where all edge weights are binary. Let $\mathcal{A}_G$ be the automorphism group of a graph defined as $\mathcal{A}_G := \{\Pi \in \mathcal{S}_N : \Pi A \Pi^\top = A, \Pi b = b\}$, which characterizes the symmetries of $G$. Hereinafter, $\mathcal{G}$ is assumed to be a subgroup of $\mathcal{S}_N$.

**Graph Signals and Learning Task.** We consider graph signals supported in the nodes of a fixed graph $G$ and maps between graphs signals. Let $\mathcal{X} = \mathbb{R}^{N \times d}, \mathcal{Y} = \mathbb{R}^{N \times k}$ be the input and output graph signal spaces. We denote by $f$ a map between graph signals, $f : \mathcal{X} \to \mathcal{Y}$. Even though the functions $f$ depend on $G$, we don't explicitly write $G$ as part of the notation of $f$ because $G$ is fixed. Our goal is to learn a target map between graph signals $f^* : \mathcal{X} \to \mathcal{Y}$. To this end, we assume access to a training set $\{(X_i, Y_i)\}$ where the $X_i$ are i.i.d. sampled from an $\mathcal{S}_N$-invariant distribution $\mu$ on

$\mathcal{X}$ where $\mathcal{S}_n$ acts on $\mathcal{X}$ by permuting the rows, and $Y_i = f^*(X_i) + \xi_i$ for some noise $\xi_i$. A natural assumption is that $f^*$ is approximately equivariant with respect to $\mathcal{A}_G$ in some sense. Our symmetry model selection problem concerns a sequence of hypothesis class $\{\mathcal{H}_\mathcal{G}\}$ indexed by $\mathcal{G}$, where we choose the best class $\mathcal{H}_\mathcal{G}^*$ such that the estimator $\hat{f} \in \mathcal{H}_\mathcal{G}^*$ gives the best generalization performance.

**Equivariant Graph Networks.** We propose to learn the target function $f^*$ on the fixed graph using $\mathcal{G}$-*equivariant graph networks* ($\mathcal{G}$-Net), which are equivariant to a chosen symmetry group $\mathcal{G}$ depending on the graph domain. Using standard techniques from representation theory, such as Schur's lemma and projections to isotypic components [68], we can parameterize $\mathcal{G}$-Net by interleaving $\mathcal{G}$-equivariant linear layers $f_\mathcal{G} : \mathbb{R}^{N \times d} \to \mathbb{R}^{N \times k}$ with pointwise nonlinearity, where the weights in the linear map $f_\mathcal{G}$ are constrained in patterns depending on the group structure (also known as parameter sharing or weight tying [69], see Appendix A.1 for technical details). In practice, we can make $\mathcal{G}$-Net more flexible by systematically breaking the symmetry, such as incorporating graph convolutions (i.e., $A f_\mathcal{G}$) and locality constraints (i.e., $A \odot f_\mathcal{G}$). Compared to standard GNNs (described in Appendix D.1), $\mathcal{G}$-Net uses a more expressive linear map to gather global information (i.e., weights are not shared among all nodes). Importantly, $\mathcal{G}$-Net yields a suite of models that allows us to flexibly choose the hypothesis class (and estimator) reflecting the active symmetries in the data, and subsumes standard GNNs that are permutation-equivariant with respect to passive symmetries (but not necessarily equivariant with respect to active symmetries).

**Graphons.** A graphon is a symmetric measurable function $W : [0,1]^2 \to [0,1]$. Graphons represent (dense) graph limits where the number of nodes goes to infinity; they can also be viewed as random graph models where the value $W(x, y)$ represents the probability of having an edge between the nodes $x$ and $y$. Let $\mathcal{W}$ denote the space of all graphons and $\eta$ denote the Lebesgue measure. Lovász and Szegedy [70] introduced the cut norm on $\mathcal{W}$ as

$$\|W\|_\square := \sup_{\substack{S, T \subset [0,1] \\ S, T \text{ measurable}}} \left| \int_{S \times T} W(u, v) \, d\eta(u) \, d\eta(v) \right|. \tag{3}$$

Based on the cut norm, Borgs et al. [71] defined the cut distance between two graphons $W, U \in \mathcal{W}$,

$$\delta_\square(W, U) := \inf_{f \in S_{[0,1]}} \|W - U^f\|_\square, \tag{4}$$

where $S_{[0,1]}$ is the set of all measure-preserving bijective measurable maps between $[0,1]$ and itself, and $U^f(x, y) = U(f(x), f(y))$. Note that the cut distance is "permutation-invariant," where measure preserving bijections are seen as the continuous counterparts of permutations.

# 3 Generalization with Exact Symmetry

In this section, we assume the target map $f^*$ is $\mathcal{A}_G$-equivariant and study the symmetry model selection problem by comparing the (statistical) risk of different models. Concretely, the risk quantifies how a given model performs on average on any potential input. The smaller the risk, the better the model performs. Thus, we use the *risk gap* of two functions $f, f'$, defined as

$$\Delta(f, f') := \mathbb{E}\left[ \|Y - f(X)\|_F^2 \right] - \mathbb{E}\left[ \|Y - f'(X)\|_F^2 \right], \tag{5}$$

as our model selection metric. Following ideas from Elesedy and Zaidi [8], we analyze the risk gap between any function $f \in V$ and its $\mathcal{G}$-equivariant version, for $\mathcal{G} \subseteq \mathcal{S}_N$.

**Lemma 1** (Risk Gap). *Let $\mathcal{X} = \mathbb{R}^{N \times d}, \mathcal{Y} = \mathbb{R}^{N \times k}$ be the input and output graph signal spaces on a fixed graph $G$. Let $X \sim \mu$ where $\mu$ is a $\mathcal{S}_N$-invariant distribution on $\mathcal{X}$. Let $Y = f^*(X) + \xi$, where $\xi \in \mathbb{R}^{N \times k}$ is random, independent of $X$ with zero mean and finite variance and $f^* : \mathcal{X} \to \mathcal{Y}$ is $\mathcal{A}_G$-equivariant. Then, for any $f \in V$ and for any compact group $\mathcal{G} \subseteq \mathcal{S}_N$, we can decompose $f$ as*

$$f = \bar{f}_\mathcal{G} + f_\mathcal{G}^\perp,$$

*where $\bar{f}_\mathcal{G} = \mathcal{Q}_\mathcal{G} f, f_\mathcal{G}^\perp = f - \bar{f}_\mathcal{G}$. Moreover, the risk gap satisfies*

$$\Delta(f, \bar{f}_\mathcal{G}) = \mathbb{E}\left[ \|Y - f(X)\|_F^2 \right] - \mathbb{E}\left[ \|Y - \bar{f}_\mathcal{G}(X)\|_F^2 \right] = \underbrace{-2\langle f^*, f_\mathcal{G}^\perp \rangle_\mu}_{\text{mismatch}} + \underbrace{\left\| f_\mathcal{G}^\perp \right\|_\mu^2}_{\text{constraint}}.$$

Lemma 1, proven in Appendix B.1, adapts ideas from [8] to equate the risk gap to the symmetry "mismatch" between the chosen group $\mathcal{G}$ and the target group $\mathcal{A}_G$, plus the symmetry "constraint" captured by the norm of the anti-symmetric part of $f$ with respect to $\mathcal{G}$. Our symmetry model selection problem aims to find the group $\mathcal{G} \subseteq \mathcal{S}_N$ such that the risk gap is maximized. Note that for $\mathcal{G} \leq \mathcal{A}_G$, the mismatch term vanishes since $f^*$ is $\mathcal{A}_G$-equivariant, but the constraint term decreases with $\mathcal{G}$; When $\mathcal{G} = \mathcal{A}_G$, we recover Lemma 6 in [8]. On the other hand, for $\mathcal{G} > \mathcal{A}_G$, the mismatch term can be positive, negative or zero (depending on $f^*$) whereas the constraint term increases with $\mathcal{G}$.

### 3.1 Linear Regression

In this section, we focus on the linear regression setting and analyze the risk gap of using the equivariant estimator versus the vanilla estimator. We consider linear estimator $\hat{\Theta} : \mathbb{R}^{N \times d} \to \mathbb{R}^{N \times k}$ that predicts $\hat{Y} = f_{\hat{\Theta}}(X) = \hat{\Theta}^\top X$. Given a compact group $\mathcal{G}$ and any linear map $\Theta$, we obtain its $\mathcal{G}$-equivariant version via projection to the $\mathcal{G}$-equivariant space (also known as intertwiner average),

$$\Psi_{\mathcal{G}}(\Theta) = \int_{\mathcal{G}} \phi(g) \, \Theta \, \psi\left(g^{-1}\right) \mathrm{d}\lambda(g). \tag{6}$$

We denote $\Psi_{\mathcal{G}}^{\perp}(\Theta) = \Theta - \Psi_{\mathcal{G}}(\Theta)$ as the projection of $\hat{\Theta}$ to the orthogonal complement of the $\mathcal{G}$-equivariant space. Here $\Psi_{\mathcal{G}}$ instantiates the orbit-average operator $\mathcal{Q}_{\mathcal{G}}$ (eqn. 2) for linear functions.

**Theorem 2** (Bias-Variance-Tradeoff). *Let $\mathcal{X} = \mathbb{R}^{N \times d}, \mathcal{Y} = \mathbb{R}^{N \times k}$ be the graph signals spaces on a fixed graph $G$. Let $\mathcal{S}_N$ act on $\mathcal{X}$ and $\mathcal{Y}$ by permuting the rows with representations $\phi$ and $\psi$. Let $\mathcal{G}$ be a subgroup of $\mathcal{S}_N$ acting with restricted representations $\phi|_{\mathcal{G}}$ on $\mathcal{X}$ and $\psi|_{\mathcal{G}}$ on $\mathcal{Y}$. Let $X_{[i,j]} \overset{i.i.d.}{\sim} \mathcal{N}\left(0, \sigma_X^2\right)$ and $Y = f^*(X) + \xi$ where $f^*(x) = \Theta^\top x$ is $\mathcal{A}_G$-equivariant and $\Theta \in \mathbb{R}^{Nd \times Nk}$. Assume $\xi_{[i,j]}$ is random, independent of $X$, with mean 0 and $\mathbb{E}\left[\xi\xi^\top\right] = \sigma_\xi^2 I < \infty$. Let $\hat{\Theta}$ be the least-squares estimate of $\Theta$ from $n$ i.i.d. examples $\{(X_i, Y_i) : i = 1, \ldots, n\}$, $\Psi_{\mathcal{G}}(\hat{\Theta})$ be its equivariant version with respect to $\mathcal{G}$. Let $\left(\chi_{\psi|_{\mathcal{G}}} \mid \chi_{\phi|_{\mathcal{G}}}\right) = \int_{\mathcal{G}} \chi_{\psi|_{\mathcal{G}}}(g)\chi_{\phi|_{\mathcal{G}}}(g)\mathrm{d}\lambda(g)$ denote the inner product of the characters. If $n > Nd + 1$ the risk gap is*

$$\mathbb{E}\left[\Delta\left(f_{\hat{\Theta}}, f_{\Psi_{\mathcal{G}}(\hat{\Theta})}\right)\right] = \underbrace{-\sigma_X^2 \left\|\Psi_{\mathcal{G}}^{\perp}(\Theta)\right\|_F^2}_{bias} + \underbrace{\sigma_\xi^2 \frac{N^2 dk - \left(\chi_{\psi|_{\mathcal{G}}} \mid \chi_{\phi|_{\mathcal{G}}}\right)}{n - Nd - 1}}_{variance}.$$

The bias term depends on the anti-symmetric part of $\Theta$ with respect to $\mathcal{G}$, whereas the variance term captures the difference of the dimension between the space of linear maps $\mathbb{R}^{Nd} \to \mathbb{R}^{Nk}$ (measured by $N^2 dk$), and the space of $\mathcal{G}$-equivariant linear maps (measured by $\left(\chi_{\psi|_{\mathcal{G}}} \mid \chi_{\phi|_{\mathcal{G}}}\right)$; a proof can be found in [72, Section 2.2]). The bias term reduces to zero for $\mathcal{G} \leq \mathcal{A}_G$ and we recover Theorem 1 in [8] when $\mathcal{G} = \mathcal{A}_G$. Notably, by using a larger group $\mathcal{G}$, the dimension of the equivariant space measured by $\left(\chi_{\psi|_{\mathcal{G}}} \mid \chi_{\phi|_{\mathcal{G}}}\right)$ is smaller, and thus the variance term increases; meanwhile, the bias term decreases due to extra (symmetry) constraints on the estimator. We remark that $\left(\chi_{\psi|_{\mathcal{G}}} \mid \chi_{\phi|_{\mathcal{G}}}\right)$ depends on $d, k$ as well: For $d = k = 1$, $\phi(g) = \psi(g) \in \mathbb{R}^{N \times N}$ are standard permutation matrices; For general $d > 1, k > 1$, $\phi(g) \in \mathbb{R}^{Nd \times Nd}, \psi(g) \in \mathbb{R}^{Nk \times Nk}$ are block-diagonal matrices, where each block is a permutation matrix.

To understand the significance of the risk gap, we note that (see Appendix B.1) when $n > Nd + 1$, the risk of the least square estimator is

$$\mathbb{E}\left[\|Y - \hat{\Theta}^\top X\|_F^2\right] = \sigma_\xi^2 \frac{Nd}{n - Nd - 1} + \sigma_\xi^2. \tag{7}$$

Thus, when $n$ is small enough so that the risk gap in Theorem 2 is dominated by the variance term, enforcing equivariance gives substantial generalization gain of order $\frac{N^2 dk - \left(\chi_{\psi|_{\mathcal{G}}} \mid \chi_{\phi|_{\mathcal{G}}}\right)}{n - Nd - 1}$.

**Example 3.1.** In Appendix C.1 we construct an example with $\mathcal{X} = \mathbb{R}^3, \mathcal{Y} = \mathbb{R}^3$, and $x \sim \mathcal{N}(0, \sigma_X^2 I_d)$. We consider a target linear function $f^*$ that is $\mathcal{S}_2$-equivariant and *approximately* $\mathcal{S}_3$-equivariant, and compare the estimators $f_{\psi_{\mathcal{S}_2}(\hat{\Theta})}$ versus $f_{\psi_{\mathcal{S}_3}(\hat{\Theta})}$. When the number of training samples $n$ is small,

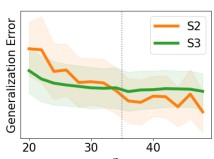

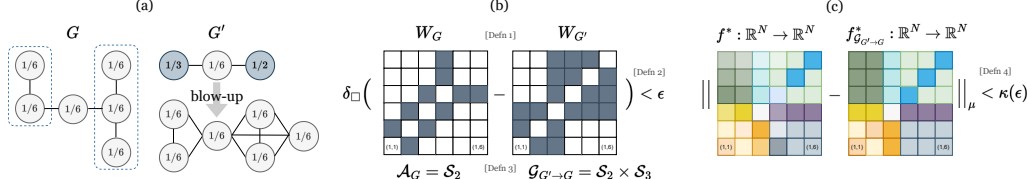

Figure 1: Overview of Definitions for Approximate Symmetries: (a) Graph $G$ and its coarsened graph $G'$; (b) Induced graphons $\mathcal{W}_G, \mathcal{W}_{G'}$ with small cut distance $\delta_\square$ and their symmetries $\mathcal{A}_G = \mathcal{S}_2, \mathcal{A}_{G'} = \{e\}$. The induced symmetry group $\mathcal{G}_{G\to G'}$ yields more symmetries for $G$; (c) Approximate equivariant mapping $f^*$ that is close to $f^*_{\mathcal{G}_{G\to G'}}$.

using a $\mathcal{S}_3$-equivariant least square estimator yields better test error than the $\mathcal{S}_2$-equivariant one, as shown in figure inset. The dashed vertical line denotes the theoretical threshold $n^* \approx 35$, before which using $\mathcal{S}_3$ yields better generalization than $\mathcal{S}_2$. This is an example where enforcing more symmetries than the target (symmetry) results on better generalization properties.

## 4 Generalization with Approximate Symmetries

Large graphs are typically asymmetric. Therefore, the assumption of $f^*$ being $\mathcal{A}_G$-equivariant in Section 3 becomes trivial. Yet, graphon theory asserts that graphs can be seen as living in a "continuous" metric space, with the cut distance (Definition 1). Since graphs are continuous entities, the combinatorial notion of exact symmetry of graphs is not appropriate. We hence relax the notion of exact symmetry to a property that is "continuous" in the cut distance. The regularity lemma [73, Theorem 5.1] asserts that any large graph can be approximated by a coarser graph in the cut distance. Since smaller graphs tend to exhibit more symmetries than larger ones, we are motivated to consider the symmetries of coarse-grained versions of graphs as their approximate symmetries (Definition 2, 3). We then present the notion of approximately equivariant mappings (Definition 4), which allows us to precisely characterize the bias-variance tradeoff (Corollary 3) in the approximate symmetry setting based on Lemma 1. Effectively, we reduce selecting the symmetry group to choosing the coarsened graph $G'$. Moreover, our symmetry model selection perspective allows for exploring different coarsening procedures and choosing the one that works best for the problem.

**Definition 1** (Induced graphon)**.** *Let $G$ be a (possibly edge weighted) graph with node set $[N]$ and adjacency matrix $A = \{a_{i,j} \in [0,1]\}_{i,j=1}^N$. Let $\mathcal{P}_N = \{I_N^n = (\frac{n-1}{N}, \frac{n}{N})\}_{n=1}^N$ be the equipartition of $[0,1]$ to $N$ intervals (where formally the last interval is the closed interval $[\frac{N-1}{N}, 1]$). We define the graphon $W_G$ induced by $G$ as*

$$W_G(x,y) = \sum_{i,j=1}^N a_{i,j} \mathbb{1}_{I_N^i}(x) \mathbb{1}_{I_N^j}(y),$$

*where $\mathbb{1}_{I_N^i}$ is the indicator function of the set $I_N^i \subset [0,1]$, $i = 1, \ldots, N$.*

We induce a graphon from an edge-node weighted graph $G' = ([M], A, b)$, with rational node weights $b_i$, as follows. Let $N \in \mathbb{N}$ be a value such that the node weights can be written in the form $b = \{b_m = \frac{q_m}{N}\}_{m=1}^M$, for some $q_m \in \mathbb{N}_{\geq 0}$, $m = 1, \ldots, M$. We *blow-up* $G'$ to an edge weighted graph $\overline{G}_N$ of $\sum q_m$ nodes by splitting each node $m$ of $G'$ into $q_m$ nodes $\{n_m^j\}_{j=1}^{q_m}$ of weight 1, and defining the adjacency matrix $\overline{A}_N$ with entries $\overline{a}_{n_m^j, n_{m'}^{j'}} = a_{m,m'}$. Note that for any two choices $N_1, N_2 \in \mathbb{N}$ in the above construction, $\delta_\square(W_{\overline{G}_{N_1}}, W_{\overline{G}_{N_2}}) = 0$, where the infimum in the definition of $\delta_\square$ is realized on some permutation of intervals in $[0,1]$, as explained below. We hence define the induced graphon of the edge-node weighted graph $G'$ by $W_{G'} := W_{\overline{G}_N}$ for some $N$, where, in fact, each choice of $N$ gives a representative of an equivalence class of piecewise constant graphons with zero $\delta_\square$ distance.

**Definition 2.** *Let $G'$ be an edge-node weighted graph with $M$ nodes and node weights $b = \{b_m = \frac{q_m}{N'}\}_{m=1}^M$, satisfying $\{q_m\}_m \subset \mathbb{N}_{\geq 0}$ and $\sum q_m = N$, and let $G$ be an edge weighted graph with $N$*

*nodes. We say that $G'$ coarsens $G$ up to error $\epsilon$ if*

$$\delta_\square(W_G, W_{G'}) < \epsilon.$$

Suppose that $G'$ coarsens $G$. This implies the existence of an assignment of nodes of $G$ to nodes of $G'$. Namely, the supremum underlying the definition of $\delta_\square$ over the measure preserving bijections $\phi : [0,1] \to [0,1]$ is realized by a permutation $\Pi : [N] \to [N]$ applied on the intervals $\{I_N^i\}_{i=1}^N$. With this permutation, the assignment $C_{G \to G'}$ defined by $[N] \ni \Pi(n_m^j) \mapsto m \in [M]$, for every $m = 1, \ldots, M$ and $j = 1, \ldots, q_m$, is called a *cluster assignment* of $G$ to $G'$.

Let $G'$, with $M$ nodes, be a edge-node weighted graph that coarsens the simple graph $G$ of $N$ nodes. Let $C_{G \to G'}$ be a cluster assignment of $G$ to $G'$. Let $\mathcal{A}_{G'}$ be the automorphism group of $G'$. Note that two nodes can belong to the same orbit of $\mathcal{A}_{G'}$ only if they have the same node weight. Hence, for every two nodes $m, m'$ in the same orbit of $\mathcal{A}_{G'}$ (i.e., equivalent clusters), there exists a bijection $\psi_{m,m'} : C_{G \to G'}^{-1}\{m\} \to C_{G \to G'}^{-1}\{m'\}$. We choose a set of such bijections, such that for every $m, m'$ on the same orbit of $\mathcal{A}_{G'}$

$$\psi_{m,m'} = \psi_{m',m}^{-1}.$$

We moreover choose $\psi_{m,m}$ as the identity mapping for each $m = 1, \ldots, M$. We identify each element $g$ of $\mathcal{A}_{G'}$ with an element $\overline{g}$ in a permutation group $\overline{\mathcal{A}}_{G'}$ of $[N]$, called the *blown-up symmetry group*, as follows. Given $g \in \mathcal{A}_{G'}$, for every $m \in [M]$ and every $n \in C_{G \to G'}^{-1}\{m\}$, define

$$\overline{g}n := \psi_{m,gm}n.$$

**Definition 3.** *Let $G'$, with $M$ nodes, be a edge-node weighted graph that coarsen the (simple or weighted) graph $G$ of $N$ nodes. Let $C_{G \to G'}$ be a cluster assignment of $G$ to $G'$ with cluster sizes $c_1, \ldots, c_M$. Let $\mathcal{A}_{G'}$ be the automorphism group of $G'$, and $\overline{\mathcal{A}}_{G'}$ be the blown-up symmetry group of $[N]$. For every $m = 1, \ldots M$, let $\mathcal{S}_{c_m}$ be the symmetry group of $C_{G \to G'}^{-1}\{m\}$. We call the group of permutations*

$$\mathcal{G}_{G \to G'} := \left( \mathcal{S}_{c_1} \times \mathcal{S}_{c_2} \ldots \times \mathcal{S}_{c_M} \right) \rtimes \overline{\mathcal{A}}_{G'} \subseteq \mathcal{S}_N$$

*the symmetry group of $G$ induced by the coarsening $G'$. We call any element of $\mathcal{G}_{G \to G'}$ an approximate symmetry of $G$.*

Specifically, every element $s \in \mathcal{G}_{G \to G'}$ can be written in coordinates by

$$s = s_1 s_2 \ldots s_M a \sim (s_1, \ldots, s_M, a),$$

for a unique choice of $s_j \in \mathcal{S}_{c_j}$, $j = 1, \ldots, M$, and $a \in \overline{\mathcal{A}}_{G'}$ (see Appendix B.2 for details).

In words, we consider the symmetry of the graph $G$ of $N$ nodes not by its own automorphism group, but via the symmetry of its coarsened graph $G'$. The nodes in the same orbit in $G$ are either in the same cluster of $G'$ (i.e., from the same coarsened node), or in the equivalent clusters of $G'$ (i.e., they belong to the same orbit of the coarsened graph and share the same cluster size). See Figure 1 (and Figure 9 in Appendix A.2) for examples.

**Definition 4.** *Let $G$ be a graph with $N$ nodes, and $\mathcal{X} = \mathbb{R}^{N \times d}$ be the space of graph signals. Let $X \sim \mu$ where $\mu$ is a $\mathcal{S}_N$-invariant measure on $\mathbb{R}^{N \times d}$. We call $f^* : \mathbb{R}^{N \times d} \to \mathbb{R}^{N \times k}$ an approximately equivariant mapping if there exists a function $\kappa : \mathbb{R}_+ \to \mathbb{R}_+$ satisfying $\lim_{\epsilon \to 0} \kappa(\epsilon) = 0$ (called the equivariance rate), such that for every $\epsilon > 0$ and every edge-node weigted graph $G'$ that coarsen $G$ up to error $\epsilon$,*

$$\|f^* - f^*_{\mathcal{G}_{G \to G'}}\|_\mu \leq \kappa(\epsilon),$$

*where $f^*_{\mathcal{G}_{G \to G'}} = \mathcal{Q}_{\mathcal{G}_{G \to G'}}(f)$ is the intertwining projection of $f^*$ with respect to $\mathcal{G}_{G \to G'}$.*

**Example 4.1.** Here we give an example of an approximately equivariant mapping in a natural setting. Suppose that $G$ is a random geometric graph, namely, a graph that was sampled from a metric-probability space $\mathcal{M}$ by randomly and independently sampling nodes $\{x_n\}_{n=1}^N \subset \mathcal{M}$. Hence, $G$ can be viewed as a discretization of $\mathcal{M}$, and the graphs that coarsen $G$ can be seen as coarser discretizations of $\mathcal{M}$. Therefore, the approximate symmetries are *local deformations* of $G$, namely, permutations that swap nodes if they are close in the metric of $\mathcal{M}$. This now gives an interpretation for approximately equivariance mappings of geometric graphs: these are mappings that are stable to local deformations. In Appendix C.2 we develop this example in detail.

We are ready to present the bias-variance tradeoff in the setting with approximate symmetries. The proofs are deferred to Appendix B.2.

**Corollary 3** (Risk Gap via Graph Coarsening). *Let $\mathcal{X} = \mathbb{R}^{N \times d}, \mathcal{Y} = \mathbb{R}^{N \times k}$ be the input and output graph signal spaces on a fixed graph $G$. Let $X \sim \mu$ where $\mu$ is a $\mathcal{S}_N$-invariant distribution on $\mathcal{X}$. Let $Y = f^*(X) + \xi$, where $\xi \in \mathbb{R}^{N \times k}$ is random, independent of $X$ with zero mean and finite variance, and $f^* : \mathbb{R}^{N \times d} \to \mathbb{R}^{N \times k}$ be an approximately equivariant mapping with equivariance rate $\kappa$. Then, for any $G'$ that coarsens $G$ up to error $\epsilon$, for any $f \in V$, we have*

$$\Delta(f, \bar{f}_{\mathcal{G}_{G \to G'}}) = \underbrace{-2\langle f^*, f^{\perp}_{\mathcal{G}_{G \to G'}} \rangle_\mu}_{mismatch} + \underbrace{\left\| f^{\perp}_{\mathcal{G}_{G \to G'}} \right\|^2_\mu}_{constraint} \geq -2\kappa(\epsilon) \left\| f^{\perp}_{\mathcal{G}_{G \to G'}} \right\|_\mu + \left\| f^{\perp}_{\mathcal{G}_{G \to G'}} \right\|^2_\mu$$

Notably, Corollary 3 illustrates the tradeoff explicitly in the form of choosing the coarsened graph $G'$: If we choose $G'$ close to $G$ such that the coarsening error $\epsilon$ and $\|f^{\perp}_{\mathcal{G}_{G \to G'}}\|_\mu$ are small, then the mismatch term is close to zero; meanwhile the constraint term is also small since $\mathcal{A}_\mathcal{G}$ is typically trivial. On the other hand, if we choose $G'$ far from $G$ that yields large coarsening error $\epsilon$, then the constraint term $\|f^{\perp}_{\mathcal{G}_{G \to G'}}\|^2_\mu$ also increases.

**Corollary 4** (Bias-Variance-Tradeoff via Graph Coarsening). *Consider the same linear regression setting in Theorem 2, except now $f^*$ is an approximately equivariant mapping with equivariance rate $\kappa$, and $\mathcal{G} = \mathcal{G}_{G \to G'}$ is controlled by $G'$ that coarsens $G$ up to error $\epsilon$. Denote the canonical permutation representations of $\mathcal{G}_{G \to G'}$ on $\mathcal{X}, \mathcal{Y}$ as $\phi', \psi'$, respectively. Let $(\chi_{\psi'} \mid \chi_{\phi'}) = \int_{\mathcal{G}_{G \to G'}} \chi_{\psi'}(g) \chi_{\phi'}(g) \mathrm{d}\lambda(g)$ denote the inner product of the characters. If $n > Nd + 1$ the risk gap is bounded by*

$$\mathbb{E}\left[ \Delta\left( f_{\hat{\Theta}}, f_{\Psi_{\mathcal{G}_{G \to G'}}(\hat{\Theta})} \right) \right] \geq -2\kappa(\epsilon) \sqrt{\sigma_\xi^2 \frac{N^2 dk - (\chi_{\psi'} \mid \chi_{\psi'})}{n - Nd - 1}} + \sigma_\xi^2 \frac{N^2 dk - (\chi_{\psi'} \mid \chi_{\psi'})}{n - Nd - 1}.$$

## 5 Experiments

We illustrate our theory in three real-world tasks for learning on a fixed graph: image inpainting, traffic flow prediction, and human pose estimation[1]. For image inpainting, we demonstrate the bias-variance tradeoff via graph coarsening using $\mathcal{G}$-Net; For the other two applications, we show that the tradeoff not only emerges from $\mathcal{G}$-Net that enforces strict equivariance, but also $\mathcal{G}$-Net augmented with symmetry-breaking modules, allowing us to *recover* standard GNN architectures as a special case. Concretely, we consider the following variants (more details in Appendix D.4.2):

1. $\mathcal{G}$-Net with strict equivariance using equivariant linear map $f_\mathcal{G}$.
2. $\mathcal{G}$-Net augmented with graph convolution $A f_\mathcal{G}(x)$, denoted as $\mathcal{G}$-Net(gc).
3. $\mathcal{G}$-Net augmented with graph convolution and learnable edge weights: $\mathcal{G}$-Net(gc+ew).
4. $\mathcal{G}$-Net augmented with graph locality constraints $(A \odot f_\mathcal{G})(x)$ and learnable edge weights, denoted as $\mathcal{G}$-Net(pt+ew).

$\mathcal{G}$-Net serves as a baseline to validate our generalization analysis for equivariant estimators (see Section 3). Yet in practice, we observe that augmenting $\mathcal{G}$-Net with symmetry breaking can further improve performance, thereby justifying the analysis for approximate symmetries (see Section 4). In particular, $\mathcal{G}$-Net(gc) and $\mathcal{G}$-Net(gc+ew) are motivated by graph convolutions used in standard GNNs; $\mathcal{G}$-Net(pt+ew) is inspired by the concept of locality in CNNs, where $A \odot f_\mathcal{G}$ effectively restricts the receptive field to the 1-hop neighrborhood induced by the graph $A$.

### 5.1 Application: Image Inpainting

We consider a $28 \times 28$ grid graph as the fixed domain, with grey-scale images as the graph signals. The learning task is to reconstruct the original images given *masked* images as inputs (i.e., image inpainting). We use subsets of MNIST [74] and FashionMNIST [75], each comprising 100 training samples and 1000 test samples. The input and output graph signals are $(m_i \odot x_i, x_i)$, where $x_i \in \mathbb{R}^{28 \times 28} \equiv \mathbb{R}^{784}$ denotes the image signals and $m_i$ denotes a random mask (size $14 \times 14$ for MNIST and $20 \times 20$ for FashionMNIST). We investigate the symmetry model selection problem by clustering

---

[1]The source code is available at `https://github.com/nhuang37/Approx_Equivariant_Graph_Nets`.

the grid into $M$ patches with size $d \times d$, where $d \in \{28, 14, 7, 4, 2, 1\}$. Here $d = 28$ means one cluster (with $\mathcal{S}_N$ symmetry); $d = 1$ is 784 singleton clusters with no symmetry (trivial).

We consider $\mathcal{G}_{G \to G'}$-equivariant networks $\mathcal{G}$-Net with ReLU nonlinearity. We parameterize the equivariant linear layer $f : \mathbb{R}^N \to \mathbb{R}^N$ with respect to $\mathcal{G}_{G \to G'} = (\mathcal{S}_{c_1} \times \ldots \times \mathcal{S}_{c_M}) \rtimes \overline{\mathcal{A}}_{G'}$ using the following block-matrix form (assuming the nodes are ordered by their cluster assignment), with $f_{kl}$ denoting block matrices, and $a_k, b_k, e_{kl}$ representing scalars:

$$ f = \begin{bmatrix} f_{11} & \cdots & f_{1M} \\ & \cdots & \\ f_{M1} & \cdots & f_{MM} \end{bmatrix}, \; f_{kk} = a_k I + b_k \mathbb{1}\mathbb{1}^\top, \; f_{kl} = e_{kl} \mathbb{1}\mathbb{1}^\top \text{ for } k \neq l. \quad (8) $$

The coarsened graph symmetry $\mathcal{A}_{G'}$ induces constraints on $a_k, b_k, e_{kl}$. If $\mathcal{A}_{G'}$ is trivial, then these scalars are unconstrained. In the experiment, we consider a reflection symmetry on the coarsened grid graph, i.e., $\mathcal{A}_{G'} = \mathcal{S}_2$ which acts by reflecting the left (coarsened) patches to the right (coarsened) patches. Suppose the reflected patch pairs are ordered consecutively, then $a_k = a_{k+1}, b_k = b_{k+1}$ for $k \in \{1, 3, \ldots, M-1\}$, and $e_{kl} = e_{k+1,l-1}$

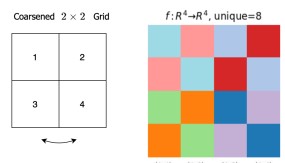

for $k \in \{1, 3, \ldots, M-1\}, l \in \{2, 4, \ldots, M\}$ (see Figure inset for an illustration). In practice, we extend the parameterization to $f : \mathbb{R}^{N \times d} \to \mathbb{R}^{N \times k}$. More details can be found in Appendix D.2.

Figure 2 shows the empirical risk first decreases and then increases as the group decreases, illustrating the bias-variance tradeoff from our theory. Figure 2 (left) compares a 2-layer $\mathcal{G}$-Net with a 1-layer linear $\mathcal{G}$-Net, demonstrating that the tradeoff occurs in both linear and nonlinear models. Figure 2 (right) shows that using reflection symmetry of the coarsened graph outperforms the trivial symmetry baseline, highlighting the utility of modeling coarsened graph symmetries.

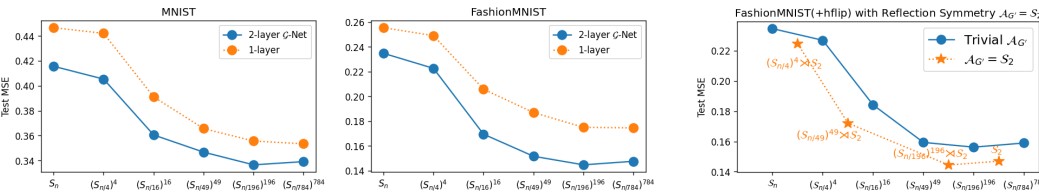

Figure 2: Bias-variance tradeoff via graph coarsening. Left:2-layer $\mathcal{G}$-Net(blue) and 1-layer linear $\mathcal{G}$-equivariant functions (orange), assuming the coarsened graph is asymmetric; Right: 2-layer $\mathcal{G}$-Net with both trivial and non-trivial coarsened graph symmetry. See Table 7 for more numerical details.

## 5.2 Application: Traffic Flow Prediction

The traffic flow prediction problem can be formulated as follows: Let $X^{(t)} \in \mathbb{R}^{n \times d}$ represent the traffic graph signal (e.g., speed or volume) observed at time $t$. The goal is to learn a function $h(\cdot)$ that maps $T'$ historical traffic signals to $T$ future graph signals, assuming the fixed graph domain $G$:

$$ \left[ X^{(t-T'+1)}, \ldots, X^{(t)}; G \right] \xrightarrow{h(\cdot)} \left[ X^{(t+1)}, \ldots, X^{(t+T)} \right]. $$

The traffic graph is typically large and asymmetric. Therefore we leverage our approximate symmetries to study the symmetry model selection problem (Section 4). We use the METR-LA dataset which represents traffic volume of highways in Los Angeles (see figure inset). We use both the original graph $G$ from [76], and its sparsified version $G_s$ which is more faithful to the road geometry. In $G_s$, the $(i, j)$-edge exists if and only if nodes $i, j$ lie on the

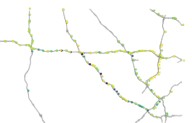

same highway. We first validate our Definition 4 in such dataset (see Appendix D.3.1). In terms of the models, we consider a simplified version of DCRNN model proposed in [76], where the spatial module is modelled by a standard GNN, and the temporal module is modelled via an encoder-decoder architecture for sequence-to-sequence modelling. We follow the same temporal module and extending its GNN module using $\mathcal{G}$-Net(gc), which recovers DCRNN when choosing $\mathcal{S}_N$. We then consider different choices of the coarsened graph $G'$ (to induce approximate symmetries). As shown in Table 3, using 2 clusters as approximate symmetries yields better generalization error than using 9 clusters, or the full permutation group $\mathcal{S}_N$.

| $\mathcal{G}$-Net(gc) | $\mathcal{S}_N$ | $\mathcal{S}_{c_1} \times \mathcal{S}_{c_2}$ | $\mathcal{S}_{c_1} \times \ldots \times \mathcal{S}_{c_9}$ |
| --- | --- | --- | --- |
| Graph $G_s$ | $3.173 \pm 0.013$ | $\mathbf{3.150 \pm 0.008}$ | $3.204 \pm 0.006$ |
| Graph $G$ | $3.106 \pm 0.013$ | $\mathbf{3.092 \pm 0.008}$ | $3.174 \pm 0.013$ |

Table 3: Traffic forecasting with different choices of graph clustering. Table shows Mean Absolute Error (MAE) across 3 runs.

| $\mathcal{G}$-Net(gc+ew) | $\mathcal{S}_{16}$ | Relax-$\mathcal{S}_{16}$ | $\mathcal{A}_G = (\mathcal{S}_2)^2$ | Trivial |
| --- | --- | --- | --- | --- |
| MPJPE $\downarrow$ | $42.55 \pm 0.88$ | $\mathbf{39.87 \pm 0.46}$ | $42.18 \pm 0.49$ | $41.60 \pm 0.32$ |
| P-MPJPE $\downarrow$ | $34.48 \pm 0.44$ | $\mathbf{31.38 \pm 0.14}$ | $32.08 \pm 0.20$ | $31.69 \pm 0.17$ |

Table 4: Human pose estimation with different choices of symmetries. Table shows mean per-joint position error (MPJPE) and MPJPE after alignment (P-MPJPE) across 3 runs.

## 5.3 Application: Human Pose Estimation

We consider the simple (loopy) graph $G$ with adjacency matrix $A \in \{0,1\}^{16 \times 16}$ representing 16 human joints, illustrated on the right. The input graph signals $X \in \mathbb{R}^{16 \times 2}$ describe the joint 2D spatial location. The goal is to learn a map to reconstruct the 3D pose $Y \in \mathbb{R}^{16 \times 3}$. We use the standard benchmark dataset Human3.6M [77] and follow the evaluation protocol in [78]. The generalization performance is measured by mean per joint position error (MPJPE) and MPJPE after alignment (P-MPJPE). We implement a 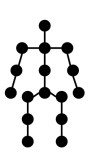 4-Layer $\mathcal{G}$-Net(gc+ew), which recovers the same model architecture in SemGCN [78] when choosing $\mathcal{G} = \mathcal{S}_{16}$ (further details are provided in Appendix D.4.1).

We consider the following symmetry groups: the permutation group $\mathcal{S}_{16}$, the graph automorphism $\mathcal{A}_G$, and the trivial symmetry. Note that the human skeleton graph has $\mathcal{A}_G = (\mathcal{S}_2)^2$ (corresponding to the arm flip and leg flip). Additionally, we investigate an approximate symmetry called Relax-$\mathcal{S}_{16}$; This relaxes the global weight sharing in $\mathcal{S}_{16}$ (that only learns 2 scalars per equivariant linear layer $f_{\mathcal{S}_{16}} : \mathbb{R}^N \to \mathbb{R}^N$, where $f_{\mathcal{S}_{16}}[i,i] = a, f_{\mathcal{S}_{16}}[i,j] = b$ for $i \neq j$) to local weight sharing (that learns $2 \times 16 = 32$ scalars, where $f_{\text{Relax-}\mathcal{S}_{16}}[i,i] = a_i, f_{\text{Relax-}\mathcal{S}_{16}}[i,j] = b_i$ for $i \neq j$). Table 4 shows that the best performance is achieved at the hypothesis class induced from Relax-$\mathcal{S}_{16}$, which is smaller than $\mathcal{S}_{16}$ but differs from $\mathcal{A}_G$. Furthermore, using $\mathcal{G}$-Net(gc+ew) with Relax-$\mathcal{S}_{16}$ gives a substantial improvement over $\mathcal{S}_{16}$ (representing SemGCN in [78]). This demonstrates the utility of enforcing active symmetries in GNNs that results in more expressive models. More details and additional ablation studies can be found in Appendix D.4.

## 6 Discussion

In this paper, we focus on learning tasks where the graph is fixed, and the dataset consists of different signals on the graph. We developed an approach for designing GNNs based on active symmetries and approximate symmetries induced by the symmetries of the graph and its coarse-grained versions. A layer of an approximately equivariant graph network uses a linear map that is equivariant to the chosen symmetry group; the graph is not used as an input but rather induces a hypothesis class. In practice, we further break the symmetry by incorporating the graph in the model computation, thus combining symmetric and asymmetric components.

We theoretically show a bias-variance tradeoff between the loss of expressivity due to imposing symmetries, and the gain in regularity, for settings where the target map is assumed to be (approximately) equivariant. For simplicity, the theoretical analysis focuses on the equivariant models without symmetry breaking; Theoretically analyzing the combination of symmetric and asymmetric components in machine learning models is an interesting open problem. The bias-variance formula is computed only for a simple linear regression model with white noise and in the underparamterized setting; Extending it to more realistic models and overparameterized settings is a promising direction.

As a proof of concept, our approximately equivariant graph networks only consider symmetries of the fixed graph. An interesting future direction is to extend our approach to also account for (approximate) symmetries in node features and labels, using suitably generalized cut distance (e.g., graphon-signal cut distance in [79]; see Appendix C.2 for an overview). Our network architecture consists only of linear layers and pointwise nonlinearity. Another promising direction is to incorporate pooling layers (e.g., [22, 80]) and investigate them through the lens of approximate symmetries. Finally, extending our analysis to general groups is a natural next step. Our techniques are based on compact groups with orthogonal representation. While we believe that the orthogonality requirement can be lifted straightforwardly, relaxing the compactness requirement appears to be more challenging (see related discussion in [46]).

## Acknowledgments

This research project was started at the BIRS 2022 Workshop "Deep Exploration of non-Euclidean Data with Geometric and Topological Representation Learning" held at the UBC Okanagan campus in Kelowna, B.C. The authors thank Ben Blum-Smith (JHU), David Hogg (NYU), Erik Thiede (Cornell), Wenda Zhou (Open AI), Carey Priebe (JHU), and Rene Vidal (UPenn) for valuable discussions, and Hong Ye Tan (Cambridge) for pointing out typos in an earlier version of the manuscript. The authors also thank the anonymous NeurIPS reviewers and area chair for helpful feedback. Ningyuan Huang is partially supported by the MINDS Data Science Fellowship from Johns Hopkins University. Ron Levie is partially funded by ISF (Israel Science Foundation) grant #1937/23. Soledad Villar is partially funded by the NSF–Simons Research Collaboration on the Mathematical and Scientific Foundations of Deep Learning (MoDL) (NSF DMS 2031985), NSF CISE 2212457, ONR N00014-22-1-2126 and an Amazon AI2AI Faculty Research Award.

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

# A   Parameterization of equivariant linear Maps

We consider a group $\mathcal{G}$ acting on spaces $\mathcal{X}$ and $\mathcal{Y}$ via representations $\phi$ and $\psi$, respectively. Our goal is to find the equivariant linear maps $f : \mathcal{X} \to \mathcal{Y}$ such that $f(\phi(g)x) = \psi(g)f(x)$ for all $g \in \mathcal{G}$ and $x \in \mathcal{X}$. The standard way to do this, used extensively in the equivariant machine learning literature (e.g. [40, 43]), is to decompose $\phi$ and $\psi$ in irreducibles and use Schur's lemma.

In a nutshell, a group representation $\varphi$ is an homomorphism $\mathcal{G} \to \mathrm{GL}(V)$, where $\mathrm{GL}(V)$ denotes the General Linear group of the vector space $V$ (sometimes mathematicians say that $V$ is a representation of $\mathcal{G}$, but we need to know the homomorphism $\varphi$ too). One way to interpret the group homomorphism (i.e. $\varphi(gh) = \varphi(g) \circ \varphi(h)$) is that the group multiplication corresponds to the composition of linear invertible maps (i.e. matrix multiplication). A linear subspace $W$ of $V$ is said to be a subrepresentation of $\varphi$ if $\varphi(\mathcal{G})(W) \subset W$. An irreducible representation is one that only has itself and the trivial subspace as subrepresentations.

Schur's lemma states that if $V, W$ are vector spaces over $\mathbb{C}$ and $\varphi_V$, $\varphi_W$ are irreducible representations, then either (1) $\varphi_V$ and $\varphi_W$ are not isomorphic as representations (and the only equivariant linear map between $V, W$ is the zero map), or (2) $\varphi_V$ and $\varphi_W$ are isomorphic and the only non-trivial equivariant maps are of the form $\lambda I$ where $\lambda \in \mathbb{C}$ and $I$ is the identity (See Chapter 1 of [68]).

Given $\mathcal{G}$ acting on spaces $\mathcal{X}$ and $\mathcal{Y}$ via representations $\phi$ and $\psi$, respectively, one can decompose $\phi$ and $\psi$ in irreducibles over $\mathbb{C}$

$$\phi = \oplus_{k=1}^{\ell} a_k \mathcal{T}_k \quad \psi = \oplus_{k=1}^{\ell} b_k \mathcal{T}_k$$

(this notation assumes the same irreducibles appear in both decompositions, which can be done if we allow some of the $a_k$ and $b_k$ to be zero). Then one can parameterize the equivariant linear maps by having one complex parameter per irreducible that appears in both decompositions.

These ideas can be applied to real spaces too. By Maschke's theorem we can decompose the representation in irreducibles over $\mathbb{R}$. Then we can check further how to decompose these irreducibles over $\mathbb{C}$, and apply Schur's lemma. We have 3 cases for the decomposition:

1. The irreducible over $\mathbb{R}$ is also irreducible over $\mathbb{C}$. In this case we directly apply Schur's lemma.

2. The irreducible over $\mathbb{R}$ decomposes in two different irreducibles over $\mathbb{C}$. In this case we can send each $\mathbb{C}$-irreducible to their isomorphic counterpart.

3. The irreducibles over $\mathbb{R}$ decompose in two copies of the same irreducible over $\mathbb{C}$. In this case we can send each irreducible to any isomorphic copy independently.

Therefore, finding the equivariant linear maps reduces to decomposing the corresponding representations in irreducibles. In the next sections we explain in detail how to do this for the specific problems described in this paper. The appendix is organized as follows: We first show how to parameterize equivariant linear layers for abelian group using isotypical decomposition (Section A.1), and then discuss the case for the symmetry group induced by graph coarsening, which further considers parameterizing permutation-equivariant linear maps to model the within-cluster symmetries (Section A.2).

## A.1   Equivariant Linear Maps via Isotypical Decomposition

In this section, we assume that the graph adjacency matrix $A$ has distinct eigenvalues $\lambda_1 > \lambda_2 > \ldots > \lambda_n$. Then $\mathcal{A}_G$ is an abelian group [81, Lemma 3.8.1]. Under this assumption, we present the construction of $\mathcal{A}_G$-equivariant linear maps using isotypical decomposition (i.e. decomposition into isomorphism classes of irreducible representations) and group characters. We remark that such construction extends to non-abelian groups and refer the interested reader to [82], but we omit it here for the ease of exposition.

We consider the simplest setting where $f : \mathbb{R}^N \to \mathbb{R}^N$ is a linear function that maps graph signals. Let $x \in \mathbb{R}^N$ be the node features, then $\mathcal{A}_G$-equivariance requires

$$f(g\, x) = g\, f(x) \quad \text{for all } g \in \mathcal{A}_G. \tag{9}$$

To construct equivariant linear functions $f$, our roadmap is outlined as follows:

1. Decompose the vector space $\mathcal{X} = \mathbb{R}^N$ into a sum of components such that different components cannot be mapped to each other equivariantly (also known as the isotypic decomposition);

2. Given $\mathcal{X} = \oplus_i \mathcal{X}_i$ an isotypic representation, we then parameterize $f$ by linear maps at each $\mathcal{X}_i$ such that for all $i$, $f(\mathcal{X}_i) \subseteq \mathcal{X}_i$.

To this end, we need the following definitions.

**Definition 5.** *($\mathcal{G}$-module, [82, Defn 1.3.1]) Let $\mathcal{X}$ be a vector space and $\mathcal{G}$ be a group. We say the vector space $\mathcal{X}$ is a $\mathcal{G}$-module or $\mathcal{X}$ carries a representation of $\mathcal{G}$ if there is a group homomorphism $\rho : \mathcal{G} \to GL(\mathcal{X})$, where $GL$ denotes the General Linear group. Equivalently, if the following holds:*

1. *$gv \in \mathcal{X}$,*

2. *$g(cv + dw) = c(gv) + d(gw)$,*

3. *$(gh)v = g(hv)$,*

4. *$ev = v$*

*for all $g, h \in \mathcal{G}; v, w \in \mathcal{X}$ and scalars $c, d \in \mathbb{C}$ ($e \in \mathcal{G}$ denotes the identity element).*

In what follows, we consider $\mathcal{X} = \mathbb{R}^N$ carries a representation of $G$.

**Definition 6.** *(Group characters) Given a group $\mathcal{G}$ and a representation $\phi : G \to GL(V)$, the group character is a function $\chi : \mathcal{G} \to \mathbb{R}$ (or $\mathbb{C}$) defined as $\chi(g) := \operatorname{tr} \phi(g)$.*

**Definition 7.** *(Group orbits) Let $\mathcal{X}$ be a vector space and $\mathcal{G}$ be a group. The group orbit of an element $x \in \mathcal{X}$ is $O(x) := \{gx : g \in \mathcal{G}\}$.*

Let $g_1, \ldots, g_s$ be the generators of $\mathcal{A}_G \subset (\mathcal{S}_2)^n$, or simply $\mathcal{A}_G \equiv (\mathcal{S}_2)^k$ for some $k \leq n$. Since $\mathcal{A}_G$ is abelian, any irreducible representation is 1-dimensional [68, p.8]. In other words, the irreducible representations of an abelian group are homomorphisms

$$\rho : \mathcal{A}_G \to \mathbb{C}. \tag{10}$$

Since all the elements of the group $\mathcal{A}_G = (\mathcal{S}_2)^k$ is of order 1 or 2, the homomorphisms are $\rho : \mathcal{A}_G \to \{\pm 1\} \subset \mathbb{R}$. By Definition 6, the irreducible characters (i.e., characters of irreducible matrix representation) are also homomorphisms $\rho : \mathcal{A}_G \to \{\pm 1\}$. In other words, $\chi(g) \in \{\pm 1\}$ for all $g \in \mathcal{A}_G$. Then we can write down the $2^k \times 2^k$ character table, where the rows are the characters $\chi$, and the columns are the group elements $g \in \mathcal{A}_G$ (see Table 1 as an example). Now, define the projection onto the isotypic component of the representation $X$ as

$$P_\chi := \frac{\deg(X)}{|\mathcal{A}_G|} \sum_{g \in \mathcal{A}_G} \overline{\chi(g)} \, g = \frac{1}{|\mathcal{A}_G|} \sum_{g \in \mathcal{A}_G} \chi(g) \, g, \tag{11}$$

where the second equality uses the fact that $\mathcal{A}_G$ is abelian.

Intuitively, applying $P_\chi$ on $\mathcal{X} = \operatorname{span}(\{e_1, \ldots, e_N\})$ picks out all $v \in \mathcal{X}$ that stays in the same subspace defined by the group character $\chi$. (Note that for the $(\mathcal{S}_2)^k$ case $\chi^{-1}(g) = \chi(g)$ since $\chi(g) \in \{\pm 1\}$).

Algorithm 1 presents the construction of equivariant linear map $f$ with respect to an abelian group. Such construction can parameterize all abelian $\mathcal{A}_G$-equivariant linear maps, as shown in Lemma 5.

**Lemma 5.** *$f$ is linear, equivariant with respect to the abelian group $\mathcal{A}_G$ if and only if $f$ can be written as* (13) *in Algorithm 1.*

*Proof.* By construction in Algorithm 1, $f$ is linear and equivariant. To show the converse, we first recall some useful facts and notations. Since $\mathcal{A}_G$ is abelian with all irreducible representations being one-dimensional, for isotypic components $\mathcal{X}_{\chi_1} \not\cong \mathcal{X}_{\chi_2}$, we have

$$g \, v_1 = \lambda_1(g) \, v_1, \quad \text{for all } g \in \mathcal{G}, v_1 \in \mathcal{X}_{\chi_1}, \tag{14}$$

$$g \, v_2 = \lambda_2(g) \, v_2, \quad \text{for all } g \in \mathcal{G}, v_2 \in \mathcal{X}_{\chi_2}, \tag{15}$$

where there exists some $g \in \mathcal{G}$ such that $\lambda_1(g) \neq \lambda_2(g)$. To show $f$ being linear and equivariant implies for all $v \in \mathcal{X}_\chi$, $f(v) \in \mathcal{X}_\chi$, we prove by contradiction. Without loss of generality, suppose

$$f(v_{\chi_1}) = \alpha_1 v_{\chi_1} + \alpha_2 v_{\chi_2}, \tag{16}$$

**Algorithm 1** Parameterizing equivariant linear functions $f : \mathbb{R}^N \to \mathbb{R}^N$ for abelian group

**Require:** Abelian group $\mathcal{A}_G = (\mathcal{S}_2)^k$
 1. Construct the character table of $\chi_{\text{irreps}}$ for $\mathcal{A}_G$, i.e. $\chi_i : \mathcal{A}_G \to \{\pm 1\}$ $i = 1, \dots \ell$;
 2. For each character $\chi_i$ in the character table, compute the projection matrix

$$P_{\chi_i}(\mathcal{X}) = [P_{\chi_i}(e_1); \dots; P_{\chi_i}(e_N)] \in \mathbb{R}^{N \times N}; \tag{12}$$

 followed by computing the basis from $P_{\chi_i}(\mathcal{X})$ and call it $\mathcal{X}_{\chi_i} = [b_{\chi_i}^{(1)}, \dots, b_{\chi_i}^{(K_i)}]$.
 3. $\mathcal{X} = \oplus_{i=1}^{\ell} \mathcal{X}_{\chi_i}$ where $\mathcal{X}_{\chi_i}$ are the isotypic component. Then $f$ is any linear function satisfying that $f(\mathcal{X}_{\chi_i}) \subseteq \mathcal{X}_{\chi_i}$ for all $i = 1, \dots, \ell$. In particular, in the basis $[b_{\chi_i}^{(s)}]_{1 \le i \le \ell, 1 \le s \le K_i}$ $f$ can be written as a block diagonal matrix $\mathbb{R}^{n \times n}$ with each block $M_{\chi_i}$ being the linear map from $\mathcal{X}_{\chi_i} \to \mathcal{X}_{\chi_i}$,

$$f = \begin{bmatrix} M_{\chi_1} & & & \\ & M_{\chi_2} & & \\ & & \ddots & \\ & & & M_{\chi_\ell} \end{bmatrix}. \tag{13}$$

 **return** $f$

|  | $e$ | $\sigma$ |
|---|---|---|
| $\chi_e$ | 1 | 1 |
| $\chi_2$ | 1 | $-1$ |

Table 1: Character table for $\mathrm{aut}(P_4) \cong Z_2$

for some scalars $\alpha_1, \alpha_2$ and $v_{\chi_1} \in \mathcal{X}_{\chi_1}, v_{\chi_2} \in \mathcal{X}_{\chi_2}$. Then by (14), for all $g \in \mathcal{G}$,

$$f(g\, v_{\chi_1}) = f(\lambda_1(g)\, v_{\chi_1}) = \lambda_1(g) f(v_{\chi_1}) = \lambda_1(g) \alpha_1 v_{\chi_1} + \lambda_1(g) \alpha_2 v_{\chi_2}. \tag{17}$$

Now, since $f$ is equivariant, for all $g \in \mathcal{G}$,

$$f(g\, v_{\chi_1}) = g f(v_{\chi_1}) = g(\alpha_1 v_{\chi_1} + \alpha_2 v_{\chi_2}) = \lambda_1(g) \alpha_1 v_{\chi_1} + \lambda_2(g) \alpha_2 v_{\chi_2}. \tag{18}$$

But there exists some $g' \in \mathcal{G}$ such that $\lambda_1(g') \ne \lambda_2(g')$, which leads to $f(g' v_{\chi_1}) \ne f(g' v_{\chi_1})$, a contradiction. One can easily extend the proof strategy to the general case for $f(v_{\chi_1}) = \sum_{i=1}^{l} v_{\chi_i}$.
$\square$

**Example A.1.** Consider the path graph on 4 nodes (i.e., $P_4$). We have $\mathrm{aut}(P_4) = \{e, (14)(23)\} \cong Z_2$.

Steps 1: Note that $Z_2$ is abelian and thus all irreducible characters $\chi(g) \in \{\pm 1\}$, for all $g \in Z_2$. The character table is shown in Table 1.

Step 2: using (11) we have

$$P_{\chi_e}[e_1; e_2; e_3; e_4] = \frac{1}{2} \begin{bmatrix} 1 & 0 & 0 & 1 \\ 0 & 1 & 1 & 0 \\ 0 & 1 & 1 & 0 \\ 1 & 0 & 0 & 1 \end{bmatrix} \text{ which yields basis } \mathcal{B}(P_{\chi_e}) = [e_1 + e_4; e_2 + e_3].$$

$$P_{\chi_2}[e_1; e_2; e_3; e_4] = \frac{1}{2} \begin{bmatrix} 1 & 0 & 0 & -1 \\ 0 & 1 & -1 & 0 \\ 0 & -1 & 1 & 0 \\ -1 & 0 & 0 & 1 \end{bmatrix} \text{ which yields basis } \mathcal{B}(P_{\chi_2}) = [e_1 - e_4; e_2 - e_3].$$

Step 3: Parameterize $f : \mathbb{R}^4 \to \mathbb{R}^4$ by $f : \mathcal{B}(P_{\chi_e}) \to \mathcal{B}(P_{\chi_e})$ and $f : \mathcal{B}(P_{\chi_2}) \to \mathcal{B}(P_{\chi_2})$. For all $v \in \mathbb{R}^4$, write $v = c_1(e_1 + e_4) + c_2(e_2 + e_3) + c_3(e_1 - e_4) + c_4(e_2 - e_3)$, then

$$f(v) = \begin{bmatrix} \alpha_1 & \alpha_2 \\ \alpha_3 & \alpha_4 \end{bmatrix} \begin{bmatrix} c_1 \\ c_2 \end{bmatrix} + \begin{bmatrix} \alpha_5 & \alpha_6 \\ \alpha_7 & \alpha_8 \end{bmatrix} \begin{bmatrix} c_3 \\ c_4 \end{bmatrix}, \tag{19}$$

where $\alpha_1, \dots, \alpha_8$ are (learnable) real scalars. Now $f$ is linear, equivariant by construction.

## A.2 Equivariant Linear Map for Symmetries Induced by Graph Coarsening

In this section, we provide additional details of constructing equivariant linear maps for using the symmetry group induced by graph coarsening (Definition 3). Recall the symmetry group with $M$ clusters of $G$ (with the associated coarsened graph $G'$) is given by

$$\mathcal{G}_{G \to G'} := \left(\mathcal{S}_1 \times \mathcal{S}_2 \ldots \times \mathcal{S}_M\right) \rtimes \overline{\mathcal{A}}_{G'} \subset \mathcal{S}_N.$$

We first assume that $\overline{\mathcal{A}}_{G'}$ is trivial and show how to parameterize equivariant functions with respect to products of permutations. Then we discuss more general cases, for instance if $\overline{\mathcal{A}}_{G'}$ is abelian. For the ease of exposition, we consider $X \in \mathbb{R}^N, Y \in \mathbb{R}^N$.

Suppose $\overline{\mathcal{A}}_{G'}$ is trivial, we claim that a linear function $f : \mathbb{R}^N \to \mathbb{R}^N$ is equivariant with respect to $\mathcal{G}_{G \to G'}$ if and only if it admits the following block-matrix form:

$$f = \begin{bmatrix} f_{11} & f_{12} & \cdots & f_{1M} \\ f_{21} & f_{22} & \cdots & f_{2M} \\ & & \ddots & \\ f_{M1} & f_{M2} & \cdots & f_{MM} \end{bmatrix}, \; f_{kk} = a_k I_{c_k} + b_k \mathbb{1}_{c_k} \mathbb{1}_{c_k}^\top, \; f_{kl} = e_{kl} \mathbb{1}_{c_k} \mathbb{1}_{c_l}^\top \text{ for } k \neq l, \quad (20)$$

where $f_{kl}$ are block matrices, $I_k$ is a size-$k$ identity matrix, $\mathbb{1}_k$ is a length-$k$ vector of all ones, and $a_k, b_k, e_{kl}$ are scalars. The subscript $c_k$ denotes the size of the $k$-th cluster. Figure 5 illustrates the block structure of $f$.

We provide a proof sketch of our claim. Since $\overline{\mathcal{A}}_{G'}$ is trivial by assumption, it remains to show any function that is equivariant to the product of permutations $\left(\mathcal{S}_1 \times \mathcal{S}_2 \ldots \times \mathcal{S}_M\right)$ admits the form in eqn. 20. We justify as follows:

1. (within-cluster) $f_{kk}$ is a linear permutation-equivariant function if and only if its diagonal elements are the same and its off-diagonal elements are the same ([34, Lemma 3.]);
2. (across-cluster) $f_{kl}$ for $k \neq l$ is a constant matrix since nodes within a cluster are indistinguishable; moreover, $e_{kl}$ are unconstrained since the coarsened symmetry $\overline{\mathcal{A}}_{G'}$ is trivial.

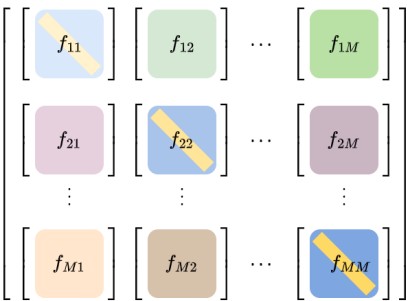

Figure 5: The block structure of equivariant linear function $f : \mathbb{R}^n \to \mathbb{R}^n$ with respect to $\mathcal{G}_{G \to G'}$ (where $G, G'$ are asymmetric): Each diagonal block $f_{kk}$ is diagonally constant and off-diagonally constant; Each off-diagonal block $f_{kl}$ is a constant matrix.

As an example, we illustrate the equivariant linear layer for two-cluster graph coarsening. Without loss of generality, assume that the node signals $X$ are ordered according to the cluster assignment (e.g., $X_{[1:|V_1|]}$ are node features for the first cluster, etc). Let $X_{(1)}, X_{(2)}$ denote the node features for the first and second cluster, $W_{(1)}^s, W_{(2)}^s$ denote the weights on the block diagonal for self-feature transformation, $W_{(1)}^n, W_{(2)}^n$ denote the weights on the block diagonal for within-cluster neighbors, and $W_{(12)}^n, W_{(21)}^n$ denote the weights off the block diagonal for across-cluster neighbors. Let $I$ denote the identity matrix, $\mathbf{1}_{(1)}, \mathbf{1}_{(2)}$ denote the all-ones matrices with the same size as the corresponding

cluster, and $\mathbf{1}_{(12)}, \mathbf{1}_{(21)}$ denote the all-ones matrices mapping across clusters. Recall $\odot$ denotes the element-wise multiplication of two matrices. Then the equivariant linear layer is parameterized as

$$I \odot \begin{bmatrix} X_{(1)}W_{(1)}^s \\ X_{(2)}W_{(2)}^s \end{bmatrix} + \left( \begin{bmatrix} \mathbf{1}_{(1)} & 0 \\ 0 & \mathbf{1}_{(2)} \end{bmatrix} - I \right) \odot \begin{bmatrix} X_{(1)}W_{(1)}^n \\ X_{(2)}W_{(2)}^n \end{bmatrix} + \begin{bmatrix} 0 & \mathbf{1}_{(12)} \\ \mathbf{1}_{(21)} & 0 \end{bmatrix} \odot \begin{bmatrix} X_{(1)}W_{(12)}^n \\ X_{(2)}W_{(21)}^n \end{bmatrix}. \quad (21)$$

For cases where $\overline{\mathcal{A}}_{G'}$ is nontrivial, if $\overline{\mathcal{A}}_{G'}$ is abelian, we can use a construction by Serre ([83] Section 8.2). Observe that the symmetry of $\overline{\mathcal{A}}_{G'}$ effectively constrains the patterns of $e_{kl}$ in equation 20. We provide an example where $\overline{\mathcal{A}}_{G'} = \mathcal{S}_2$ in our image inpainting experiment (see Section 5.1), and defer the investigation of general cases for future work.

# B  Proofs of Our Theoretical Results

## B.1  Proofs of Generalization with Exact Symmetries

**Lemma 1** (Risk Gap). *Let $\mathcal{X} = \mathbb{R}^{N \times d}, \mathcal{Y} = \mathbb{R}^{N \times k}$ be the input and output graph signal spaces on a fixed graph $G$. Let $X \sim \mu$ where $\mu$ is a $\mathcal{S}_N$-invariant distribution on $\mathcal{X}$. Let $Y = f^*(X) + \xi$, where $\xi \in \mathbb{R}^{N \times k}$ is random, independent of $X$ with zero mean and finite variance and $f^* : \mathcal{X} \to \mathcal{Y}$ is $\mathcal{A}_G$-equivariant. Then, for any $f \in V$ and for any compact group $\mathcal{G} \subseteq \mathcal{S}_N$, we can decompose $f$ as*

$$f = \bar{f}_{\mathcal{G}} + f_{\mathcal{G}}^{\perp},$$

*where $\bar{f}_{\mathcal{G}} = \mathcal{Q}_{\mathcal{G}} f, f_{\mathcal{G}}^{\perp} = f - \bar{f}_{\mathcal{G}}$. Moreover, the risk gap satisfies*

$$\Delta(f, \bar{f}_{\mathcal{G}}) = \mathbb{E}\left[\|Y - f(X)\|_F^2\right] - \mathbb{E}\left[\|Y - \bar{f}_{\mathcal{G}}(X)\|_F^2\right] = \underbrace{-2\langle f^*, f_{\mathcal{G}}^{\perp}\rangle_\mu}_{mismatch} + \underbrace{\left\|f_{\mathcal{G}}^{\perp}\right\|_\mu^2}_{constraint}.$$

Lemma 1 is a straightforward extension of Lemma 6 in [8], which makes use of Lemma 1 in [8].

**Lemma 1 in [8].** *Let $U$ be any subspace of $V$ that is closed under $\mathcal{Q}$. Define the subspaces $S$ and $A$ of, respectively, the $\mathcal{G}$-symmetric and $\mathcal{G}$-anti-symmetric functions in $U$ : $S = \{f \in U : f$ is $\mathcal{G}$-equivariant $\}$ and $A = \{f \in U : \mathcal{Q}f = 0\}$. Then $U$ admits admits an orthogonal decomposition into symmetric and anti-symmetric parts*

$$U = S \oplus A$$

*Proof of Lemma 1.* The first part of Lemma 1 $f = \bar{f}_{\mathcal{G}} + f_{\mathcal{G}}^{\perp}$ follows from Lemma 1 in [8]. For the second part, by the assumption that the noise $\xi$ is independent of $X$ with zero mean and finite variance, we can simplify the risk gap as

$$\Delta(f, \bar{f}_{\mathcal{G}}) := \mathbb{E}\left[\|Y - f(X)\|_F^2\right] - \mathbb{E}\left[\|Y - \bar{f}_{\mathcal{G}}(X)\|_F^2\right]$$
$$= \mathbb{E}\left[\|f^*(X) - f(X)\|_F^2\right] - \mathbb{E}\left[\|f^*(X) - \bar{f}_{\mathcal{G}}(X)\|_F^2\right]. \quad (22)$$

Substituting $f = \bar{f}_{\mathcal{G}} + f_{\mathcal{G}}^{\perp}$ yields

$$\mathbb{E}\left[\|f^*(X) - \bar{f}_{\mathcal{G}}(X) - f_{\mathcal{G}}^{\perp}(X)\|_F^2\right] - \mathbb{E}\left[\|f^*(X) - \bar{f}_{\mathcal{G}}(X)\|_F^2\right]$$
$$= -2\langle f^*(X) - \bar{f}_{\mathcal{G}}(X), f_{\mathcal{G}}^{\perp}(X)\rangle_\mu + \mathbb{E}\left[\|f_{\mathcal{G}}^{\perp}(X)\|_F^2\right]$$
$$= -2\langle f^*, f_{\mathcal{G}}^{\perp}\rangle_\mu + \left\|f_{\mathcal{G}}^{\perp}\right\|_\mu^2. \quad (23)$$

$\square$

We remark that Lemma 6 in [8] assumes that $f^*$ is $\mathcal{G}$-equivariant, so the first term in (23) vanishes. We are motivated from the symmetry model selection problem, and thereby relax the assumption of the chosen symmetry group $\mathcal{G}$ can differ from the target symmetry group $\mathcal{A}_{\mathcal{G}}$.

**Theorem 2** (Bias-Variance-Tradeoff). *Let $\mathcal{X} = \mathbb{R}^{N \times d}, \mathcal{Y} = \mathbb{R}^{N \times k}$ be the graph signals spaces on a fixed graph $G$. Let $\mathcal{S}_N$ act on $\mathcal{X}$ and $\mathcal{Y}$ by permuting the rows with representations $\phi$ and $\psi$. Let $\mathcal{G}$ be a subgroup of $\mathcal{S}_N$ acting with restricted representations $\phi|_{\mathcal{G}}$ on $\mathcal{X}$ and $\psi|_{\mathcal{G}}$ on $\mathcal{Y}$.*

Let $X_{[i,j]} \overset{i.i.d.}{\sim} \mathcal{N}\left(0, \sigma_X^2\right)$ and $Y = f^*(X) + \xi$ where $f^*(x) = \Theta^\top x$ is $\mathcal{A}_G$-equivariant and $\Theta \in \mathbb{R}^{Nd \times Nk}$. Assume $\xi_{[i,j]}$ is random, independent of $X$, with mean 0 and $\mathbb{E}\left[\xi\xi^\top\right] = \sigma_\xi^2 I < \infty$. Let $\hat{\Theta}$ be the least-squares estimate of $\Theta$ from $n$ i.i.d. examples $\{(X_i, Y_i) : i = 1, \ldots, n\}$, $\Psi_\mathcal{G}(\hat{\Theta})$ be its equivariant version with respect to $\mathcal{G}$. Let $\left(\chi_{\psi|_\mathcal{G}} \mid \chi_{\phi|_\mathcal{G}}\right) = \int_\mathcal{G} \chi_{\psi|_\mathcal{G}}(g) \chi_{\phi|_\mathcal{G}}(g) \mathrm{d}\lambda(g)$ denote the inner product of the characters. If $n > Nd + 1$ the risk gap is

$$\mathbb{E}\left[\Delta\left(f_{\hat{\Theta}}, f_{\Psi_\mathcal{G}(\hat{\Theta})}\right)\right] = \underbrace{-\sigma_X^2 \left\|\Psi_\mathcal{G}^\perp(\Theta)\right\|_F^2}_{bias} + \underbrace{\sigma_\xi^2 \frac{N^2 dk - \left(\chi_{\psi|_\mathcal{G}} \mid \chi_{\phi|_\mathcal{G}}\right)}{n - Nd - 1}}_{variance}.$$

Theorem 2 presents the risk gap in expectation, which follows from Lemma 1, by taking $f$ as the least-squares estimator and using assumptions in the linear regression setting. To this end, we denote $\boldsymbol{X} \in \mathbb{R}^{n \times Nd}, \boldsymbol{Y} \in \mathbb{R}^{n \times Nk}, \boldsymbol{\xi} \in \mathbb{R}^{n \times Nk}$ as the training data arranged in matrix form, where $\boldsymbol{Y} = f^*(\boldsymbol{X}) + \boldsymbol{\xi}$. Recall that the least-squares estimator of $\Theta$ in the classic regime ($n > Nd$) is given by

$$\hat{\Theta} := (\boldsymbol{X}^\top \boldsymbol{X})^\dagger \boldsymbol{X}^\top \boldsymbol{Y} \overset{a.e.}{=} \Theta + (\boldsymbol{X}^\top \boldsymbol{X})^{-1} \boldsymbol{X}^\top \boldsymbol{\xi}, \tag{24}$$

while its equivariant map is

$$\Psi_\mathcal{G}(\hat{\Theta}) = \int_\mathcal{G} \phi(g)\, \hat{\Theta}\, \psi\left(g^{-1}\right) \mathrm{d}\lambda(g). \tag{25}$$

Our proof makes use of the following results in [8], which we restate adapted versions here for our setting.

**Proposition 11 in [8].** *Let* $V = \{f_W : f_W(x) = W^\top x, W \in \mathbb{R}^{d \times k}, x \in \mathbb{R}^d\}$ *denote the space of linear functions. Let* $X \sim \mu$ *with* $\mathbb{E}[XX^\top] = \Sigma$. *For any linear functions* $f_{W_1}, f_{W_2} \in V$, *the inner product on* $V$ *satisfies*

$$\langle f_{W_1}, f_{W_2} \rangle_\mu = \mathrm{Tr}(W_1^\top \Sigma W_2). \tag{26}$$

**Theorem 13 in [8]** (Simplified, Adapted). *Consider the same setting as Theorem 2. For* $n > Nd + 1$,

$$\sigma_X^2 \mathbb{E}\left[\left\|\Psi_\mathcal{G}^\perp\left(\left(\boldsymbol{X}^\top \boldsymbol{X}\right)^+ \boldsymbol{X}^\top \boldsymbol{\xi}\right)\right\|_F^2\right] = \sigma_\xi^2 \frac{N^2 dk - \left(\chi_{\psi|_\mathcal{G}} \mid \chi_{\phi|_\mathcal{G}}\right)}{n - Nd - 1}.$$

*Proof of Theorem 2.* We first plug in the least-squares expressions $\hat{\Theta}, \Psi_\mathcal{G}(\hat{\Theta})$ to Lemma 1 and treat the mismatch term and constraint term separately; We complete the proof by collecting common terms together.

For the mismatch term, our goal is to compute

$$-2\,\mathbb{E}\left[\langle \Theta, \hat{\Theta} - \Psi_\mathcal{G}(\hat{\Theta}) \rangle_\mu\right], \tag{27}$$

where the expectation is taken over the test point $X$ and the training data $\boldsymbol{X}, \boldsymbol{\xi}$.

To that end, we write

$$\left(\hat{\Theta} - \Psi_\mathcal{G}(\hat{\Theta})\right) x \overset{a.e.}{=} \Theta^\top x + \boldsymbol{\xi}^\top \boldsymbol{X}(\boldsymbol{X}^\top \boldsymbol{X})^{-1} x - \int_\mathcal{G} \psi(g^{-1}) \left(\Theta^\top + \boldsymbol{\xi}^\top \boldsymbol{X}(\boldsymbol{X}^\top \boldsymbol{X})^{-1}\right) \phi(g) x\, \mathrm{d}\lambda(g). \tag{28}$$

Taking expectation yields

$$\mathbb{E}_{X, \boldsymbol{X}, \boldsymbol{\xi}}\left[\langle \Theta, \hat{\Theta} - \Psi_\mathcal{G}(\hat{\Theta}) \rangle_\mu\right] = \|\Theta\|_\mu^2 + \mathbb{E}_{X, \boldsymbol{X}, \boldsymbol{\xi}}\left[\langle \Theta^\top X, \boldsymbol{\xi}^\top \boldsymbol{X}(\boldsymbol{X}^\top \boldsymbol{X})^{-1} x \rangle\right]$$
$$- \mathbb{E}_{X, \boldsymbol{X}, \boldsymbol{\xi}}\left[\langle \Theta^\top x, \int_\mathcal{G} \psi(g^{-1}) \left(\Theta^\top + \boldsymbol{\xi}^\top \boldsymbol{X}(\boldsymbol{X}^\top \boldsymbol{X})^{-1}\right) \phi(g) x\, \mathrm{d}\lambda(g) \rangle\right]. \tag{29}$$

Note that $\boldsymbol{\xi}$ is independent with $\boldsymbol{X}$ and mean 0, so the second term in (29) vanishes. Similarly, the part $\mathbb{E}_{X, \boldsymbol{X}, \boldsymbol{\xi}} \int_\mathcal{G} \psi(g^{-1}) \left(\boldsymbol{\xi}^\top \boldsymbol{X}(\boldsymbol{X}^\top \boldsymbol{X})^{-1}\right) \phi(g) x\, \mathrm{d}\lambda(g)$ also vanishes (by first taking conditional

expectation of $\boldsymbol{\xi}$ conditioned on $\boldsymbol{X}$). Thus, we arrive at

$$\mathbb{E}\left[\langle\Theta,\hat{\Theta}-\Psi_{\mathcal{G}}(\hat{\Theta})\rangle_\mu\right] = \|\Theta\|_\mu^2 - \mathbb{E}_x\left[\langle\Theta^\top x, \int_{\mathcal{G}}\psi(g^{-1})\,\Theta^\top\,\phi(g)x\,\mathrm{d}\lambda(g)\rangle\right]$$
$$= \|\Theta\|_\mu^2 - \langle\Theta,\Psi_{\mathcal{G}}(\Theta)\rangle_\mu$$
$$= \|\Psi_{\mathcal{G}}^\perp(\Theta)\|_\mu^2$$
$$= -2\,\sigma_X^2\|\Psi_{\mathcal{G}}^\perp(\Theta)\|_F^2, \tag{30}$$

where the last equality follows from Proposition 11 in [8] with the assumption that $\Sigma = \sigma_X^2$. This finishes the computation for the mismatch term.

Now for the constraint term, we have

$$\|f_{\mathcal{G}}^\perp\|_\mu^2 = \|\Psi_{\mathcal{G}}^\perp(\hat{\Theta})\|_\mu^2 \tag{31}$$
$$= \sigma_X^2\,\mathbb{E}_{\boldsymbol{X},\boldsymbol{\xi}}\|\Psi_{\mathcal{G}}^\perp\left(\Theta + (\boldsymbol{X}^\top\boldsymbol{X})^{-1}\boldsymbol{X}^\top\boldsymbol{\xi}\right)\|^2 \tag{32}$$
$$= \sigma_X^2\|\Psi_{\mathcal{G}}^\perp(\Theta)\|_F^2 + \sigma_X^2\,\mathbb{E}_{\boldsymbol{X},\boldsymbol{\xi}}\|\Psi_{\mathcal{G}}^\perp\left((\boldsymbol{X}^\top\boldsymbol{X})^{-1}\boldsymbol{X}^\top\boldsymbol{\xi}\right)\|^2, \tag{33}$$

where the last equality follows from linearity of expectation, $\mathbb{E}[\boldsymbol{\xi}] = 0$ and $\boldsymbol{\xi}$ independent of $x$.

Combining the mismatch term in (30) with the constraint term in (33), the risk gap becomes

$$\mathbb{E}\left[\Delta\left(f_{\hat{\Theta}}, f_{\Psi_{\mathcal{G}}(\hat{\Theta})}\right)\right] = -\sigma_X^2\|\Psi_{\mathcal{G}_L}^\perp(\Theta)\|^2 + \sigma_X^2\,\mathbb{E}_{\boldsymbol{X},\boldsymbol{\xi}}\|\Psi_{\mathcal{G}_L}^\perp\left((\boldsymbol{X}^\top\boldsymbol{X})^{-1}\boldsymbol{X}^\top\boldsymbol{\xi}\right)\|^2, \tag{34}$$

Applying Theorem 13 in [8], the second term in (34) reduces to

$$\sigma_X^2\,\mathbb{E}_{\boldsymbol{X},\boldsymbol{\xi}}\|\Psi_{\mathcal{G}_L}^\perp\left((\boldsymbol{X}^\top\boldsymbol{X})^{-1}\boldsymbol{X}^\top\boldsymbol{\xi}\right)\|^2 = \sigma_\xi^2\frac{N^2dk - \left(\chi_{\psi|\mathcal{G}}\mid\chi_{\phi|\mathcal{G}}\right)}{n - Nd - 1}, \tag{35}$$

from which the theorem follows immediately.

$\square$

Finally, we state a well-known result for the risk of (Ordinary) Least-Squares Estimator (see [84, 85] and references therein).

**Lemma 6** (Risk of Least-Squares Estimator). *Consider the same set-up as Theorem 2. For $n > Nd + 1$,*

$$\mathbb{E}\left[\|Y - \hat{\Theta}^\top X\|_F^2\right] = \sigma_\xi^2\frac{Nd}{n - Nd - 1} + \sigma_\xi^2.$$

*Proof of Lemma 6.* Recall $X, Y$ denote the test sample. We denote the risk of the least-squares estimator *conditional on the training data* $\boldsymbol{X} \in \mathbb{R}^{n \times Nd}$ as $\mathcal{R}(\hat{\Theta}\mid\boldsymbol{X})$, which has the following bias-variance decomposition:

$$\mathcal{R}(\hat{\Theta}\mid\boldsymbol{X}) = \mathbb{E}\left[\|Y - \hat{\Theta}^\top X\|_F^2\mid\boldsymbol{X}\right] \tag{36}$$
$$= \mathbb{E}\left[\|\Theta^\top X + \xi - \hat{\Theta}^\top X\|_F^2\mid\boldsymbol{X}\right] \tag{37}$$
$$= \mathbb{E}\left[\|(\Theta - \hat{\Theta})^\top X\|_F^2\mid\boldsymbol{X}\right] + \sigma_\xi^2, \tag{38}$$

where the last equality follows from $\xi$ being zero mean and independent with $X$. The second term $\sigma_\xi^2$ is also known as *irreducible error*. We decompose the first term into

$$\mathbb{E}\left[\|(\Theta - \hat{\Theta})^\top X\|_F^2\mid\boldsymbol{X}\right] = \mathbb{E}\left[\|(\Theta - \mathbb{E}[\hat{\Theta}])^\top X\|_F^2 + \|(\mathbb{E}[\hat{\Theta}] - \hat{\Theta})^\top X\|_F^2\mid\boldsymbol{X}\right]. \tag{39}$$

Recall that $\hat{\Theta}\stackrel{a.e.}{=}(\boldsymbol{X}^\top\boldsymbol{X})^{-1}\boldsymbol{X}^\top\boldsymbol{Y} = (\boldsymbol{X}^\top\boldsymbol{X})^{-1}\boldsymbol{X}^\top(\boldsymbol{X}\Theta + \xi) = \Theta + (\boldsymbol{X}^\top\boldsymbol{X})^{-1}\boldsymbol{X}^\top\xi$. Thus $\mathbb{E}[\hat{\Theta}] = \Theta$ and (39) simplifies to $\mathbb{E}\left[\|(\mathbb{E}[\hat{\Theta}] - \hat{\Theta})^\top X\|_F^2\mid\boldsymbol{X}\right]$.

We finish computing the risk by taking expectation over $\boldsymbol{X}$, and using $\mathbb{E}[\hat{\Theta}] - \hat{\Theta} = (\boldsymbol{X}^\top \boldsymbol{X})^{-1} \boldsymbol{X}^\top \xi$,

$$\mathbb{E}\left[\|Y - \hat{\Theta}^\top X\|_F^2\right] = \mathbb{E}\left[\mathcal{R}(\hat{\Theta} \mid \boldsymbol{X})\right] \tag{40}$$

$$= \mathbb{E}_{\boldsymbol{X}}\left[\mathbb{E}_{X,\xi}\left[\|(\mathbb{E}[\hat{\Theta}] - \hat{\Theta})^\top X\|_F^2 \mid \boldsymbol{X}\right]\right] + \sigma_\xi^2 \tag{41}$$

$$= \mathbb{E}\left[\|\left((\boldsymbol{X}^\top \boldsymbol{X})^{-1} \boldsymbol{X}^\top \xi\right)^\top X\|_F^2\right] + \sigma_\xi^2 \tag{42}$$

$$= \sigma_\xi^2 \operatorname{tr}\left(\mathbb{E}[(\boldsymbol{X}^\top \boldsymbol{X})^{-1}] \sigma_X^2 I\right) + \sigma_\xi^2. \tag{43}$$

By [86, Lemma 2.3], for $n > Nd + 1$, $\mathbb{E}[(\boldsymbol{X}^\top \boldsymbol{X})^{-1}] = \frac{Nd}{n-Nd-1} I$. Putting this in (43) completes the proof. $\qquad\square$

## B.2 Proofs of Generalization with Approximate Symmetries

In Definition 3, we construct the *symmetry group of $G$ induced by the coarsening $G'$* via a semidirect product,

$$\mathcal{G}_{G \to G'} = \left(\mathcal{S}_{c_1} \times \mathcal{S}_{c_2} \ldots \times \mathcal{S}_{c_M}\right) \rtimes \overline{\mathcal{A}}_{G'}.$$

We explain the construction in more details here. We first recall the definition of semidirect product. Given two groups $\mathcal{G}_1, \mathcal{G}_2$ and a group homomorphism $\varphi : \mathcal{G}_2 \to \operatorname{aut}(\mathcal{G}_1)$, we can construct a new group $\mathcal{G}_1 \rtimes_\varphi \mathcal{G}_2$, called the semidirect product of $\mathcal{G}_1, \mathcal{G}_2$ with respect to $\varphi$ as follows:

1. The underlying set is the Cartesian product $\mathcal{G}_1 \times \mathcal{G}_2$;

2. The group operation $\circ$ is determined by the homomorphism $\varphi$, such that

$$\circ : (\mathcal{G}_1 \rtimes_\varphi \mathcal{G}_2) \times (\mathcal{G}_1 \rtimes_\varphi \mathcal{G}_2) \to (\mathcal{G}_1 \rtimes_\varphi \mathcal{G}_2)$$
$$(g_1, g_2) \circ (g_1', g_2') = (g_1\, \varphi_{g_2}(g_1'), g_2 g_2'), \quad g_1, g_1' \in \mathcal{G}_1; g_2, g_2' \in \mathcal{G}_2,$$

Take $\mathcal{G}_1 = \left(\mathcal{S}_{c_1} \times \mathcal{S}_{c_2} \ldots \times \mathcal{S}_{c_M}\right), \mathcal{G}_2 = \overline{\mathcal{A}}_{G'}$. Note that $\mathcal{G}_1$ is a normal subgroup in $\mathcal{G}_{G \to G'}$; Namely, for all $s \in \mathcal{G}_{G \to G'}, g_1 \in \mathcal{G}_1$, we have $sg_1 s^{-1} \in \mathcal{G}_1$. Thus, the map $g_1 \mapsto sg_1 s^{-1}$ is an automorphism of $\mathcal{G}_1$. In particular, the homomorphism $\varphi_{g_2}(g_1) = g_2\, g_1\, g_2^{-1}$ for $g_2 \in \mathcal{G}_2$ describes the action of $\mathcal{G}_2$ on $\mathcal{G}_1$ by conjugation [2]. In the context of a graph $G$ with $N$ nodes and its coarsening $G'$ with $M$ clusters, the homomorphism $\varphi$ describes how across-cluster permutations in $G'$ act on the original graph $G$. Note that in the special case where $\mathcal{A}_{G'}$ acts transitively on the coarsened nodes, we recover the wreath product — a special kind of semidirect product. Our construction of $\mathcal{G}_{G \to G'}$ can be seen as a natural generalization of the wreath product.

**Corollary 3** (Risk Gap via Graph Coarsening). *Let $\mathcal{X} = \mathbb{R}^{N \times d}, \mathcal{Y} = \mathbb{R}^{N \times k}$ be the input and output graph signal spaces on a fixed graph $G$. Let $X \sim \mu$ where $\mu$ is a $\mathcal{S}_N$-invariant distribution on $\mathcal{X}$. Let $Y = f^*(X) + \xi$, where $\xi \in \mathbb{R}^{N \times k}$ is random, independent of $X$ with zero mean and finite variance, and $f^* : \mathbb{R}^{N \times d} \to \mathbb{R}^{N \times k}$ be an approximately equivariant mapping with equivariance rate $\kappa$. Then, for any $G'$ that coarsens $G$ up to error $\epsilon$, for any $f \in V$, we have*

$$\Delta(f, \bar{f}_{\mathcal{G}_{G \to G'}}) = \underbrace{-2\langle f^*, f_{\mathcal{G}_{G \to G'}}^\perp \rangle_\mu}_{\text{mismatch}} + \underbrace{\left\|f_{\mathcal{G}_{G \to G'}}^\perp\right\|_\mu^2}_{\text{constraint}} \geq -2\kappa(\epsilon) \left\|f_{\mathcal{G}_{G \to G'}}^\perp\right\|_\mu + \left\|f_{\mathcal{G}_{G \to G'}}^\perp\right\|_\mu^2$$

---

[2]In our context, it is sensible to write $\varphi_{g_2}(g_1) = g_2\, g_1\, g_2^{-1}$ given that $\mathcal{G}_1, \mathcal{G}_2$ are originally inside a common group $\mathcal{S}_N$. Yet semidirect product applies to two arbitrary groups — not necessarily inside a common group initially, where $\varphi_{g_2}(g_1)$ is an abstraction of $g_2\, g_1\, g_2^{-1}$.

*Proof of Corollary 3.* We start by simplifying the mismatch term in Lemma 1,

$$-2\mathbb{E}\left[\langle f^*(x), f^\perp_{\mathcal{G}_{G\to G'}}(x)\rangle\right] = -2\mathbb{E}\left[\langle f^*(x) - f^*_{\mathcal{G}_{G\to G'}}(x) + f^*_{\mathcal{G}_{G\to G'}}(x), f^\perp_{\mathcal{G}_{G\to G'}}(x)\rangle\right]$$

$$= -2\mathbb{E}\left[\langle \underbrace{f^*(x) - f^*_{\mathcal{G}_{G\to G'}}(x)}_{\mathcal{G}_L\text{-anti-symmetric part of }f^*}, \underbrace{f^\perp_{\mathcal{G}_{G\to G'}}(x)}_{\mathcal{G}_L\text{-anti-symmetric part of }f}\rangle\right]$$

$$\geq -2\,\|f^* - f^*_{\mathcal{G}_{G\to G'}}\|_\mu\,\|f^\perp_{\mathcal{G}_{G\to G'}}\|_\mu \quad \text{(By Cauchy Schwarz Ineq.)}$$

$$\geq -2\,\kappa(\epsilon)\,\|f^\perp_{\mathcal{G}_{G\to G'}}\|_\mu. \quad \text{(By Definition 4 Approx. Equiv. Map)}$$

Putting this together with the constraint term completes the proof. $\square$

**Corollary 4** (Bias-Variance-Tradeoff via Graph Coarsening). *Consider the same linear regression setting in Theorem 2, except now $f^*$ is an approximately equivariant mapping with equivariance rate $\kappa$, and $\mathcal{G} = \mathcal{G}_{G\to G'}$ is controlled by $G'$ that coarsens $G$ up to error $\epsilon$. Denote the canonical permutation representations of $\mathcal{G}_{G\to G'}$ on $\mathcal{X}, \mathcal{Y}$ as $\phi', \psi'$, respectively. Let $(\chi_{\psi'} \mid \chi_{\phi'}) = \int_{\mathcal{G}_{G\to G'}} \chi_{\psi'}(g)\chi_{\phi'}(g)\mathrm{d}\lambda(g)$ denote the inner product of the characters. If $n > Nd + 1$ the risk gap is bounded by*

$$\mathbb{E}\left[\Delta\left(f_{\hat\Theta}, f_{\Psi_{\mathcal{G}_{G\to G'}}(\hat\Theta)}\right)\right] \geq -2\kappa(\epsilon)\sqrt{\sigma_\xi^2\frac{N^2dk - (\chi_{\psi'} \mid \chi_{\psi'})}{n - Nd - 1}} + \sigma_\xi^2\frac{N^2dk - (\chi_{\psi'} \mid \chi_{\psi'})}{n - Nd - 1}.$$

*Proof of Corollary 4.* It follows immediately from applying Theorem 13 in [8] to Corollary 3 with $\mathcal{G} = \mathcal{G}_{G\to G'}$. $\square$

# C Examples

## C.1 Example: Gaussian data model with approximate symmetries

Example 3.1 considers $\mathcal{G} = \mathcal{S}_3, \mathcal{G} = \mathcal{S}_2, \mathcal{X} = \mathbb{R}^3, \mathcal{Y} = \mathbb{R}^3$, and $x \sim \mathcal{N}(0, \sigma_X^2 I_d)$. The target function is linear, i.e., $f^*(x) = \Theta^\top x$ for some $\Theta \in \mathbb{R}^{3\times3}$. In other words, we are learning linear functions on a fixed graph domain with 3 nodes. Suppose the target function is $\mathcal{S}_2$-equivariant such that it has the form

$$\Theta = \begin{bmatrix} a & b & c \\ b & a & c \\ d & d & e \end{bmatrix}, \quad a, b, c, d, e \in \mathbb{R}. \tag{44}$$

Now, we project $\Theta$ in (44) to $\mathcal{S}_3$-equivariant space using the intertwined average 6 with the canonical permutation representation of $\mathcal{S}_3$. A direct calculation yields

$$\Psi_{\mathcal{S}_3}(\Theta) = \begin{bmatrix} \frac{1}{3}(2a+e) & \frac{1}{3}(b+c+d) & \frac{1}{3}(b+c+d) \\ \frac{1}{3}(b+c+d) & \frac{1}{3}(2a+e) & \frac{1}{3}(b+c+d) \\ \frac{1}{3}(b+c+d) & \frac{1}{3}(b+c+d) & \frac{1}{3}(2a+e) \end{bmatrix} \tag{45}$$

$$\Psi^\perp_{\mathcal{S}_3}(\Theta) = \Theta - \Psi_{\mathcal{S}_3}(\Theta) = \begin{bmatrix} \frac{1}{3}(a-e) & \frac{1}{3}(2b-c-d) & \frac{1}{3}(-b+2c-d) \\ \frac{1}{3}(2b-c-d) & \frac{1}{3}(a-e) & \frac{1}{3}(-b+2c-d) \\ \frac{1}{3}(-b-c+2d) & \frac{1}{3}(-b-c+2d) & \frac{1}{3}(-2a+2e). \end{bmatrix} \tag{46}$$

Therefore, the bias term evaluates to

$$-\sigma_X^2\,\|\Psi^\perp_{\mathcal{S}_3}(\Theta)\|^2 = -\sigma_X^2\left(\frac{2(a-e)^2}{3} + \frac{2(-2b+c+d)^2}{9} + \frac{2(b-2c+d)^2}{9} + \frac{2(b+c-2d)^2}{9}\right). \tag{47}$$

For the variance term, recall $\chi_{\psi_{\mathcal{S}_3}}, \chi_{\phi_{\mathcal{S}_3}}$ are both the canonical permutation representations of $\mathcal{S}_3$, we have

$$\left(\chi_{\psi_{\mathcal{S}_3}} \mid \chi_{\phi_{\mathcal{S}_3}}\right) = \frac{1}{6}(3^2 + 1^2 + 1^2 + 1^2 + 0^2 + 0^2) = 2. \tag{48}$$

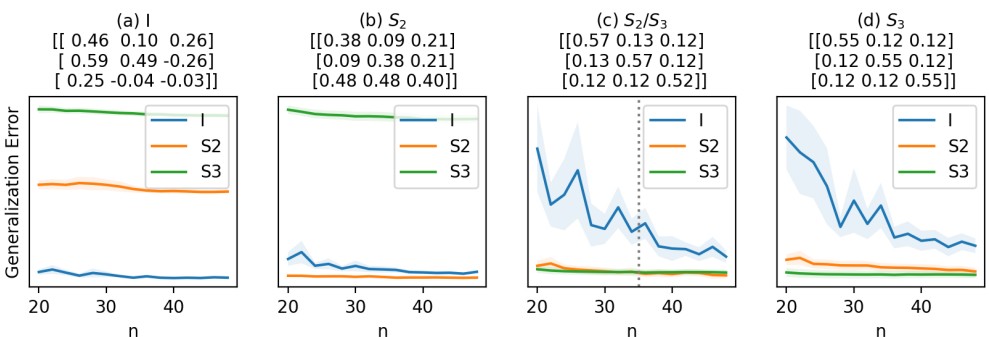

Figure 6: Choosing the symmetry group corresponding to the target function usually yields the best generalization ((a), (b), (d)), but not always: when the number of training data $n$ is small and the target function $f$ is approximately equivariant with respect to a larger group, choosing the larger symmetry group could yield further generalization gain, as shown in (c) empirically. The dashed gray vertical line highlights the theoretical threshold $n^* \approx 35$, before which using $\mathcal{S}_3$ yields better generalization than $\mathcal{S}_2$, validating our theoretical analysis. We set $\sigma_X^2 = 1, \sigma_\xi^2 = \frac{1}{64}$, conduct 10 random runs and compute the generalization error based on 300 test points. We obtain the estimators via stochastic gradient descent, and enforce the symmetry via tying weights. The titles of each subplot indicate the symmetry of the target function, and display the target function values.

Therefore, the variance term evaluates to

$$\sigma_\xi^2 \frac{N^2 - \left(\chi_{\psi|\mathcal{G}} \mid \chi_{\psi|\mathcal{G}}\right)}{n - N - 1} = \sigma_\xi^2 \frac{7}{n - 4}. \tag{49}$$

Putting (47) and (49) together yields the generalization gap of for the least square estimator $f_{\hat{\Theta}}$ compared to its $\mathcal{S}_3$-equivariant version $f_{\Psi_{\mathcal{S}_3}(\hat{\Theta})}$.

As a comparison, when choosing the symmetry group of the target function $\mathcal{G} = \mathcal{S}_2$, the bias vanishes and note that $\left(\chi_{\psi_{\mathcal{S}_2}} \mid \chi_{\phi_{\mathcal{S}_2}}\right) = \frac{1}{2}(3^2 + 1^2) = 5$, so generalization gap is

$$\mathbb{E}\left[\Delta\left(f_{\hat{\Theta}}, f_{\Psi_{\mathcal{S}_2}(\hat{\Theta})}\right)\right] = \sigma_\xi^2 \frac{4}{n - 4}. \tag{50}$$

We see that choosing $\mathcal{G} = \mathcal{S}_3$ is better if $a \approx e, b \approx c \approx d$ (i.e., $f^*$ is approximately $\mathcal{S}_3$-invariant) and the training sample size $n$ small, whereas $\mathcal{S}_2$ is better vice versa. This analysis illustrates the advantage of choosing a (suitably) larger symmetry group to induce a smaller hypothesis class when learning with limited data, and introduce useful inductive bias when the target function is approximately symmetric with respect to a larger group. We further illustrate our theoretical analysis via simulations, with details and results shown in Figure 6.

### C.2  Example: Approximately Equivariant Mapping on a Geometric Graph

In this section, we illustrate a construction of an approximately equivariant mapping. We focus on a version of Definition 3 that does not take to account the symmetries of $G'$. Namely, we consider a definition of the approximate symmetries as

$$\mathcal{G}_{G \to G'} := \mathcal{S}_{c_1} \times \mathcal{S}_{c_2} \dots \times \mathcal{S}_{c_M} \subset \mathcal{S}_N.$$

Equivalently, we restrict the analysis to coarsening graphs $G'$ that are asymmetric.

**Background from graphon-signal analysis.** To support our construction, we cite some definitions and results from [79].

**Definition 8.** *Let $r > 0$. The graphon-signal space with signals bounded by $r$ is $\mathcal{WL}_r := \mathcal{W} \times L_r^\infty[0,1]$, where $L_r^\infty[0,1]$ is the ball of radius $r$ in $L^\infty[0,1]$. The distance in $\mathcal{WL}_r$ is defined for $(W, s), (V, g) \in \mathcal{WL}_r$ by*

$$d_\square\big((W, s), (V, g)\big) := \|(W, s) - (V, g)\|_\square := \|W - V\|_\square + \|s - g\|_1.$$

*Moreover,*

$$\delta_\square\big((W,s),(V,g)\big) = \inf_\phi d_\square\big((W,s),(V^\phi,g^\phi)\big),$$

*where* $g^\phi(x) = g(\phi(x))$ *and* $\phi$ *is a measure preserving bijection.*

Any graph-signal induces a graphon signal in the natural way, as in Definition 1. The cut norm and distance between two graph-signals is defined to be the cut norm and distance between the two induced graphon-siganl respectively. Similarly, the $L_1$ distance between a signal $q$ on a graph and a signal $s$ on $[0,1]$ is defined to be the $L_1$ distance between the induced signal from $q$ and $s$. The supremum in the definition of cut distance between two induced graphon-signals is realized by some measure preserving bijection.

**Sampling graphon-signals.** The following construction is from [79, Section 3.4]. Let $\Lambda = (\lambda_1, \ldots \lambda_N) \in [0,1]^N$ be $N$ independent uniform random samples from $[0,1]$, and $(W,s) \in \mathcal{WL}_r$. We define the *random weighted graph* $W(\Lambda)$ as the weighted graph with $N$ nodes and edge weight $w_{i,j} = W(\lambda_i, \lambda_j)$ between node $i$ and node $j$. We similarly define the *random sampled signal* $s(\Lambda)$ with value $s_i = s(\lambda_i)$ at each node $i$. Note that $W(\Lambda)$ and $s(\Lambda)$ share the sample points $\Lambda$. We then define a random simple graph as follows. We treat each $w_{i,j} = W(\lambda_i, \lambda_j)$ as the parameter of a Bernoulli variable $e_{i,j}$, where $\mathbb{P}(e_{i,j} = 1) = w_{i,j}$ and $\mathbb{P}(e_{i,j} = 0) = 1 - w_{i,j}$. We define the *random simple graph* $\mathbb{G}(W, \Lambda)$ as the simple graph with an edge between each node $i$ and node $j$ if and only if $e_{i,j} = 1$. The following theorem is [79, Theorem 3.6]

**Theorem 3.6 from [79]** (Sampling lemma for graphon-signals). *Let* $r > 1$. *There exists a constant* $N_0 > 0$ *that depends on* $r$, *such that for every* $N \geq N_0$, *every* $(W,s) \in \mathcal{WL}_r$, *and for* $\Lambda = (\lambda_1, \ldots \lambda_N) \in [0,1]^N$ *independent uniform random samples from* $[0,1]$, *we have*

$$\mathbb{E}\left(\delta_\square\big((W,s),(\mathbb{G}(W,\Lambda),s(\Lambda))\big)\right) < \frac{15}{\sqrt{\log(N)}}. \tag{51}$$

By Markov's inequality and (51), for any $0 < p < 1$, there is an event of probability $1 - p$ (regarding the choice of $\Lambda$) in which

$$\delta_\square\big((W,s),(\mathbb{G}(W,\Lambda),s(\Lambda))\big) < \frac{15}{p\sqrt{\log(N)}}. \tag{52}$$

**Stability to deformations of mappings on geometric graphs.** Let $\mathcal{M}$ be a metric space with an atomless standard probability measure defined over the Borel sets (up to completion of the measure). Such a probability space is equivalent to the standard probabiltiy space $[0,1]$ with Lebesgue measure. Namely, there are co-null sets $A \subset \mathcal{M}$ and $B \subset [0,1]$, and a measure preserving bijection $\phi : A \to B$. Hence, graphon analysis applied as-is when replacing the domain $[0,1]$ with $\mathcal{M}$. Suppose that we are interested in a target function $f_\mathcal{M} : L^1(\mathcal{M}) \to L^1(\mathcal{M})$ that is stable to deformations in the following sense.

**Definition 9.** *Let* $\epsilon > 0$. *A measurable bijection* $\nu : \mathcal{M} \to \mathcal{M}$ *is called a* deformation up to $\epsilon$, *if there exists an event* $B_\epsilon \subset \mathcal{M}$ *with probability greater than* $1 - \epsilon$ *such that for every* $x \in B_\epsilon$

$$d_\mathcal{M}\big(\nu(x),x\big) < \epsilon.$$

*The mapping* $f_\mathcal{M} : L^1(\mathcal{M}) \to L^1(\mathcal{M})$ *is called* stable to deformations *with stability constant* $C$, *if for any deformation* $\nu$ *up to* $\epsilon$, *and every* $s \in L^1(\mathcal{M})$, *we have*

$$\|f_\mathcal{M}(s) - f_\mathcal{M}(s \circ \nu) \circ \nu^{-1}\|_1 < C\epsilon.$$

Suppose that we observe a discretized version of the domain $\mathcal{M}$, defined as follows. There is a graphon $W : \mathcal{M}^2 \to [0,1]$ defined as

$$W(x,y) = r\big(d(x,y)\big), \tag{53}$$

where $r : \mathbb{R}_+ \to [0,1]$ is a decreasing function with support $[0,\rho]$. Instead of observing $W$, we observe a graph $G = \mathbb{G}(W,\Lambda)$ with node set $[N]$, sampled from $W$ on the random independent points $\Lambda = \{\lambda_n\}_{n=1}^N \subset \mathcal{M}$ as above. Suppose moreover that any graph signal is sampled from a signal in $L^1(\mathcal{M})$, on the same random points $\Lambda$, as above. Suppose that the target $f_\mathcal{M}$ on the continuous

domain is well approximated by some mapping $f^* : L^1[N] \to L^1[N]$ on the discrete domain in the following sense. For every $s \in L^1(\mathcal{M})$, let $s_G$ be the graph signal sampled on the random samples $\{\lambda_n\}_n$. Then there is an event of high probability such that

$$\|f^* s_G - \{(f_{\mathcal{M}}(s))(x_n)\}_n\|_1 < e$$

for some small $e$. We hence consider $f^*$ as the target mapping of the learning problem. One example of such a scenario is when there exists some Lipschitz continuous mapping $\Theta : \mathcal{WL}_r \to \mathcal{WL}_r$ with Lipschitz constant $L$, such that $f_{\mathcal{M}} = \Theta(W, \cdot)$ and $f^* = \Theta(G, \cdot)$. Indeed, by (52), for some $p$ as small as we like, there is an event of probability $1 - p$ in which, up to a measure preserving bijection,

$$\|f_{\mathcal{M}} s - f^* s_G\|_1 \leq \delta_\square\Big((W, f_{\mathcal{M}} s), (G, f^* s_G)\Big)$$

$$\leq L\delta_\square\Big((W, s), (G, s_G)\Big) < \frac{15L}{p\sqrt{\log(N)}} = e. \tag{54}$$

A concrete example is when $\Theta$ is a message passing neural network (MPNN) with Lipschitz continuous message and update functions, and normalized sum aggregation [79, Theorem 4.1].

Let $G'$ be a graph that coarsens $G$ up to error $\epsilon$. In the same event as above, by (52), up to a measure preserving bijection,

$$\delta_\square(W_{G'}, W) \leq \delta_\square(W_{G'}, W_G) + \delta_\square(W_G, W) \leq \epsilon + e = u. \tag{55}$$

We next show an approximation property that we state here informally: Since $W(x, y) \approx 0$ for $x$ away from $y$, we must have $W_{G'}(x, y) \approx 0$ as well for a set of high measure. Otherwise, $\delta_\square(W_{G'}, W)$ cannot be small. By this, any approximate symmetry of $G$ is a small deformation, and, hence, $f^*$ is an approximately equivariant mapping.

**Equivariant mappings on geometric graphs.** In the following, we construct a scenario in which $f^*$ can be shown to be approximately equivariant in a restricted sense. For simplicity, we assume $f^*(s_G) \in L^2[0, 1]$, and restrict to the case $r = \mathbb{1}_{[0,\rho]}$ in the geometric graphon $W$ of (53). Denote the induced graphon $W_{G'} = T$. Given $h > 0$, define the $h$-*diagonal*

$$d_h = \{(x, y) \in \mathcal{M}^2 \mid d_{\mathcal{M}}(x, y) \leq h\}.$$

In the following, all distances are assumed to be up to the best measure preserving bijection.

If there is a domain $S' \times T' \in \mathcal{M}^2$ outside the $\rho$-diagonal in which $T(x, y) > c$ for some $c > 0$, by reverse triangle inequality, we must have

$$\|W - T\|_\square \geq \int_{S'} \int_{T'} T(x, y) dy dx = c\mu(S')\mu(T').$$

Hence, since by (55), $\|W - T\|_\square < u$, for every $S' \times T'$ that does not intersect $d_\rho$, we must have

$$\int_{S'} \int_{T'} T(x, y) dy dx \leq u.$$

In other words, for any two sets $S, T$ with distance more than $\rho$ ($\inf_{s \in S, t \in T} d_\mu(s, t) > \rho$), we have

$$\int_{S} \int_{T} T(x, y) dy dx \leq u.$$

This formalizes the statement "$W_{G'}(x, y) \approx 0$ for $x$ away from $y$" from above.

Next, we develop the analysis for the special case $\mathcal{M} = [0, 1]$ with the standard metric and Lebesgue probability measure. We note that the analysis can be extended to $\mathcal{M} = [0, 1]^D$ for a general dimension $D \in \mathbb{N}$. For every $z \in [0, 1]$, we have

$$\int_{[z+\rho/\sqrt{2},1]} \int_{[0,z-\rho/\sqrt{2}]} T(x, y) dy dx \leq u,$$

and

$$\int_{[0,z-\rho/\sqrt{2}]} \int_{[z+\rho/\sqrt{2},1]} T(x, y) dy dx \leq u.$$

Let $\nu > 0$. We take a grid $\{x_j\} \in [0,1]$ of spacing $\sqrt{2}\nu$. The sets

$$\bigcup_j [x_j + \rho/\sqrt{2}, 1] \times [0, x_j - \rho/\sqrt{2}], \quad \bigcup_j [0, x_j - \rho/\sqrt{2}] \times [x_j + \rho/\sqrt{2}, 1]$$

cover $d_\nu^c$ (where $d_\nu^c$ is the complement of $d_\nu$). Hence,

$$\iint_{d_\nu^c} T(x,y)dydx \leq \sum_{j=1}^{1/\sqrt{2}\nu} \int_{[x_j + \rho/\sqrt{2}, 1]} \int_{[0, x_j - \rho/\sqrt{2}]} T(x,y)dydx$$

$$+ \sum_{j=1}^{1/\sqrt{2}\nu} \int_{[0, x_j - \rho/\sqrt{2}]} \int_{[x_j + \rho/\sqrt{2}, 1]} T(x,y)dydx$$

$$\leq \frac{2}{\sqrt{2}\nu} u.$$

We take $\frac{2}{\sqrt{2}\nu} u = t$, for $u \ll t \ll 1$, namely, $\nu = \sqrt{2} \frac{u}{t}$. For example, we may take $t = \sqrt{2} u^{1/3}$, and $\nu = u^{2/3}$, assuming that $\rho < u^{1/3}$. Hence, we have

$$\iint_{d_{u^{2/3}}^c} T(x,y) \leq \sqrt{2} u^{1/3}.$$

To conclude, the probability of having an edge between nodes $\lambda_i$ and $\lambda_j$ in $\overline{G'}_N$ which are further away than $u^{2/3}$, namely, $d_{\mathcal{M}}(\lambda_i, \lambda_j) > u^{2/3}$, is less than $\sqrt{2} u^{1/3}$.

Suppose that $G'$ is asymmetric. This means that symmetries of $\mathcal{G}_{G \to G'}$ can only permute between nodes that have an edge between them in the blown-up graph $\overline{G'}_N$. The probability of having an edge between nodes further away than $u^{2/3}$ is less than $\sqrt{2} u^{1/3}$. Hence, a symmetry in $\mathcal{G}_{G \to G'}$ can be seen as a small deformation, where for each node $\lambda_i$ and a random uniform $g \in \mathcal{G}_{G \to G'}$, the probability that $\lambda_i$ it is mapped by $g$ to a node of distance less than $u^{2/3}$ is more than $1 - \sqrt{2} u^{1/3}$.

Any symmetry $g$ in $\mathcal{G}_{G \to G'}$ induces a measure preserving bijection $\nu$ in $\mathcal{M} = [0,1]$, by permuting the intervals of the partition $\mathcal{P}_N$ of Definition 1. As a result, the set of points that are mapped further away than $u^{2/3}$ under $\nu$ has probability upper bounded by $\sqrt{2} u^{1/3}$, and symmetries in $\mathcal{G}_{G \to G'}$ can be seen as a small deformation $\nu$ according to Definition 9 (in high probability). This means that, for any $g \in \mathcal{G}_{G \to G'}$,

$$\|f_{\mathcal{M}}(s) - f_{\mathcal{M}}(s \circ g) \circ g^{-1}\|_1 < C\sqrt{2} u^{1/3},$$

so by the triangle inequality, combining with equation 54, we have

$$\|f^*(s_G) - g^{-1} f^*(g s_G)\|_1 < 2e + C\sqrt{2} u^{1/3} = \epsilon'. \tag{56}$$

Equation (56) leads to

$$\|f^*(s_G) - \mathcal{Q}_{\mathcal{G}_{G \to G'}}(f^*)(s_G)\|_1 = \|f^*(s_G) - \frac{1}{|\mathcal{G}_{G \to G'}|} \sum_{g \in \mathcal{G}_{G \to G'}} g^{-1} f^*(g s_G)\|_1 \tag{57}$$

$$\leq \frac{1}{|\mathcal{G}_{G \to G'}|} \sum_{g \in \mathcal{G}_{G \to G'}} \|f^*(s_G) - g^{-1} f^*(g s_G)\|_1 < \epsilon'. \tag{58}$$

Since for any $q \in L^2[0,1] \cap L^\infty[0,1]$ we have $\|q\|_2^2 \leq \|q\|_\infty \|q\|_1$, we can bound

$$\|f^*(s_G) - \mathcal{Q}_{\mathcal{G}_{G \to G'}}(f^*)(s_G)\|_2 < \sqrt{\|f^*(s_G) - \mathcal{Q}_{\mathcal{G}_{G \to G'}}(f^*)(s_G)\|_\infty} \sqrt{\epsilon'}$$

$$< \sqrt{\|f^*(s_G)\|_\infty + \|\mathcal{Q}_{\mathcal{G}_{G \to G'}}(f^*)(s_G)\|_\infty} \sqrt{\epsilon'}$$

$$< \sqrt{2\|f^*(s_G)\|_\infty} \sqrt{\epsilon'}, \tag{59}$$

where the last inequality follows from $\|\mathcal{Q}_{\mathcal{G}_{G \to G'}}(f^*)(s_G)\|_\infty \leq \|f^*(s_G)\|_\infty$.

Denote $\|f^*\|_\infty := \int \|f^*(s_G)\|_\infty d\mu(s_G)$, and suppose that $\|f^*\|_\infty$ is finite. Hence, if $\mu$ is a probability measure, we have

$$\left\| f^* - \mathcal{Q}_{\mathcal{G}_{G \to G'}}(f^*) \right\|_\mu < \sqrt{2\|f^*\|_\infty} \sqrt{\epsilon'}.$$

This shows an example of approximately equivariant mapping based on random geometric graph, where the approximation rate is also a function of the size of the graph $N$, and goes to zero as $N \to \infty$ and $\epsilon \to 0$.

In future work, we will extend this example to more general metric space $\mathcal{M}$ and to non-trivial symmetry groups $\overline{\mathcal{A}}_{G'}$. Intuitively, most random geometric graphs are "close to asymmetric". This means that for "most" $G'$, the symmetries of $\overline{\mathcal{A}}_{G'}$ can only permute between nodes connected by an edge, and so are the symmetries of $\mathcal{G}_{G \to G'}$. For this, we need to extend Definition 9 by treating $G'$ probabilistically.

# D    Experiment Details

In this section, we provide additional details of our experiments. We first give a brief introduction of standard graph neural networks (Section D.1), followed by in-depth explanations of our applications in image inpainting (Section D.2), traffic flow prediction (Section D.3), and human pose estimation (Section D.4). All experiments were conducted on a server with 256 GB RAM and 4 NVIDIA RTX A5000 GPU cards.

## D.1    Graph Neural Networks (GNNs)

We consider standard message-passing graph neural networks (MPNNs) [19–21] defined as follows. A $L$-layer MPNN maps input $X \in \mathbb{R}^{N \times d}$ to output $Y \in \mathbb{R}^{N \times k}$ following an iterative scheme: At initialization, $\mathbf{h}^{(0)} = X$; At each iteration $l$, the embedding for node $i$ is updated to

$$\mathbf{h}_i^{(l)} = \phi \left( \mathbf{h}_i^{(l-1)}, \sum_{j \in \mathcal{N}(i)} \psi \left( \mathbf{h}_i^{(l-1)}, \mathbf{h}_j^{(l-1)}, A_{[i,j]} \right) \right), \tag{60}$$

where $\phi, \psi$ are the update and message functions, $\mathcal{N}(i)$ denotes the neighbors of node $i$, and $A_{[i,j]}$ represents the $(i,j)$-edge weight. MPNNs typically have two key design features: (1) $\phi, \psi$ are *shared* across all nodes in the graph, typically chosen to be a linear transformation or a multi-layer perceptions (MLPs), known as *global weight sharing*; (2) the graph $A$ is used for (spatial) convolution.

## D.2    Application: Image Inpainting

We provide additional details of the data, model, and optimization procedure used in the experiment.

**Data.** We consider the datasets MNIST [74] and FashionMNIST [75]. For each dataset, we take 100 training samples and 1000 test samples via stratified random sampling. The input and output graph signals are $(m_i \odot x_i, x_i)$ ($\odot$ is entrywise multiplication). Here $x_i \in \mathbb{R}^{28 \times 28} \equiv \mathbb{R}^{784}$ denotes the image signals and $m_i$ denotes a random mask (size $14 \times 14$ for MNIST and $20 \times 20$ for FashionMNIST). For experiments with reflection symmetry on the coarsened graph, we further transform each image in the FashionMNIST subset using horizontal flip with probability $0.5$ (FashionMNIST+hflip). We remark that it is possible to model the dihedral group $\mathcal{D}_4$ on the square as the coarsened graph symmetry, yet we only consider the reflection symmetry $\mathcal{S}_2$ to simplify the parameterization (since it is Abelien while $\mathcal{D}_4$ is not) while capturing the most relevant symmetries.

**Model.** We consider a 2-layer $\mathcal{G}_{G \to G'}$-equivariant networks $\mathcal{G}$-Net, a composition of $f_{\text{out}} \circ \texttt{ReLU} \circ f_{\text{in}}$, where $f_{\text{in}}, f_{\text{out}}$ denote the input/output equivariant linear layers. The input and output feature dimension is 1 (since the signals are grayscale images), and the hidden dimension is set as 28. For comparison, we also consider a simple 1-layer $\mathcal{G}_{G \to G'}$-equivariant networks $\mathcal{G}$-Net.

We train the models with ADAM (learning rate $0.01$, no weight decay, at most 1000 epochs). We report the best test accuracy at the model checkpoint selected by the best validation accuracy (with a $80/20$ training-validation split).

We supplement Figure 2 with Table 7 for further numerical details.

| MSE ($\times 1e^{-2}$)$\downarrow$ | $\mathcal{S}_{28^2} = \mathcal{S}_n$ | $(\mathcal{S}_{14^2})^4 = (S_{n/4})^4$ | $(\mathcal{S}_{7^2})^{16} = (S_{n/16})^{16}$ | $(\mathcal{S}_{4^2})^{49} = (S_{n/49})^{49}$ | $(\mathcal{S}_{2^2})^{196} = (S_{n/196})^{196}$ | Trivial $= (S_{n/784})^{784}$ |
|---|---|---|---|---|---|---|
| MNIST | $41.56 \pm 0.16$ | $40.53 \pm 0.26$ | $36.06 \pm 0.24$ | $34.68 \pm 0.5$ | $\mathbf{33.67 \pm 0.07}$ | $33.92 \pm 0.04$ |
| Fashion | $23.48 \pm 0.14$ | $22.26 \pm 0.02$ | $16.94 \pm 0.08$ | $15.16 \pm 0.1$ | $\mathbf{14.47 \pm 0.11}$ | $14.75 \pm 0.11$ |

Table 7: Image inpainting using $\mathcal{G}$-Net with different levels of coarsening. Table shows mean squared error (MSE) across 3 runs on the test set, supplementing Figure 2 (Left, blue curves).

### D.3 Application: Traffic Flow Prediction

**Data.** The METR-LA traffic dataset, [76], contains traffic information collected from 207 sensors in the highway of Los Angeles County from Mar 1st 2012 to Jun 30th 2012 [87]. We use the same traffic data normalization and $70/10/20$ train/validation/test data split as [76]. We consider two different traffic graphs constructed from the pairwise road network distance matrix: (1) the sensor graph $G$ introduced in [76] based on applying a thresholded Gaussian kernel (degree distribution in Figure 8e); (2) the sparser graph $G_s$ based on applying the binary mask where the $(i, j)$ entry is nonzero if and only if nodes $i, j$ lie on the same highway (degree distribution in Figure 8d). We construct the second variant to more faithfully model the geometry of the highway, illustrated in Figure 8a.

**Graph coarsening.** We choose 2 clusters based on highway intersection and flow direction, indicated by colors (Figure 8b (b)), and 9 clusters based on highway labels (Figure 8c (c)).

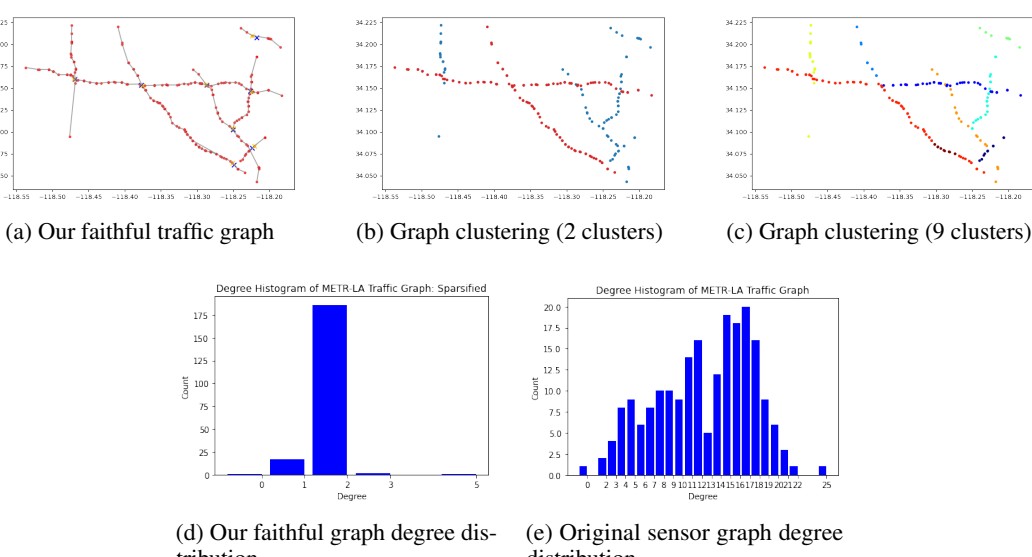

(a) Our faithful traffic graph   (b) Graph clustering (2 clusters)   (c) Graph clustering (9 clusters)

(d) Our faithful graph degree distribution

(e) Original sensor graph degree distribution

Figure 8: METR-LA traffic graph: visualization, clustering, and degree distribution

**Model.** We use a standard baseline, DCRNN proposed in [76]. DCRNN is built on a core recurrent module, DCGRU cell, which iterates as follows: Let $x_{i,t}, h_{i,t}$ denote the $i$-th node feature and hidden state vector at time $t$; Let $X_t, R_t, H_{t-1}$ be the matrices of stacking feature vectors $x_{i,t}, r_{i,t}, h_{i,t-1}$ as rows.

$$z_{i,t} = \sigma_g \left( W_z\, x_{i,t} + U_z\, h_{i,t-1} + b_z \right) \tag{61}$$

$$r_{i,t} = \sigma_g \left( W_r\, x_t + U_r\, h_{t-1} + b_r \right) \tag{62}$$

$$\hat{h}_{i,t} = \phi_h \left( [A\, X\, W_h]_{[i,:]}^\top + [A\, (R_t \odot H_{t-1})\, U_h]_{[i,:]}^\top + b_h \right) \tag{63}$$

$$h_{i,t} = z_t \odot h_{t-1} + (1 - z_t) \odot \hat{h}_t, \tag{64}$$

where $W_z, U_z, b_z, U_r, W_r, b_r, W_h, U_h, b_h$ are learnable weights and biases, $\sigma_g$ is the sigmoid function, $\phi_g$ is the hyperbolic tangent, and $h_{i,0} = 0$ for all $i$ at initialization. The crucial different from a vanilla GRU lies in eqn (63) where graph convolution replaces matrix multiplication.

We then modify the graph convolution in (63) from global weight sharing to tying weights among clusters of nodes, similar to the implementation in Appendix D.4 for Relax-$\mathcal{S}_{16}$. For example, in the

case of two clusters (orbits), we change $XW_h$ to

$$\text{swap}\left(\text{concat}[X_{c_1}W_{h,c_1}; X_{c_2}W_{h,c_2}]\right), \tag{65}$$

where $X_{c_i}$ denotes the submatrix of $X$ including the rows of nodes from cluster $i$ only, and $W_{h,c_1}, W_{h,c_2}$ are two learnable matrices. In words, we perform cluster-specific linear transformation, combine the transformed features, and reorder the rows (i.e., swap) to ensure compatibility with the graph convolution.

**Experiment Set-up.** For our experiments, we use DCRNN model with 1 RNN layer and 1 diffusion step. We choose $T' = 3$ (i.e., 3 historical graph signals) and $T = 3$ (i.e., predict the next 3 period graph signals). We train all variants for 30 epochs using ADAM optimizer with learning rate 0.01. We report the test set performance selected by the best validation set performance.

### D.3.1 Assumption Validation: Approximate Equivariant Map

Before applying our construction of approximate symmetries, we validate the assumption of the target function $f^*$ being an approximately equivariant mapping using a trained DCRNN model as a proxy. We proceed as follows:

**Data.** We use the validation set of METR-LA (traffic graph signals in LA), which has 207 nodes and consists of $14,040$ input and output signals. Each input $X \in \mathbb{R}^{207 \times 2}$ represents the traffic volume and speed in the past at the 207 stations, and output $Y \in \mathbb{R}^{207}$ representing future traffic volume.

**Model.** We use a trained DCRNN model on our faithful graph, with input being 3 historical signals $\boldsymbol{X} = (X_{T-3}, X_{T-2}, X_{T-1}) \in \mathbb{R}^{3 \times 207 \times 2}$ to predict the future signals $\boldsymbol{Y} = (X_T, X_{T+1}, X_{T+2}) \in \mathbb{R}^{3 \times 207}$. We denote this model as $f$. It gives reasonable performance with Mean Absolute Error $\approx 3$, and serves as a good proxy for the target (unknown) function $f^*$.

**Neighbors.** We take our faithful traffic graph that originally has 397 non-loop edges, and only consider a subset of 260 edges by thresholding the distance values to eliminate geometrically far-away nodes. This defines our 260 neighboring node pairs.

**Equivariance error.** For each node pair $(i, j)$, we swap their input signals by interchanging the $(i, j)$-th slices in the node dimension of the tensor $\boldsymbol{X}$, denoted as $\boldsymbol{X}_{(i,j)}$, and check if the swapped output $\hat{\boldsymbol{Y}}_{(i,j)} = f(\boldsymbol{X}_{(i,j)})$ is close to the original output $\hat{\boldsymbol{Y}} = f(\boldsymbol{X})$ with $(i, j)$-th slices swapped. We measure "closeness" via the relative equivariant error at the node pair. Concretely, let $\boldsymbol{X}[i, j]$ denote the tensor slices at the $(i, j)$ node pair, and $\boldsymbol{X}[j, i]$ being the swapped version by interchanging $(i, j)$-th slices. The relative different is computed as

$$\left|\hat{\boldsymbol{Y}}_{(i,j)}[j, i] - \hat{\boldsymbol{Y}}[i, j]\right|/\hat{\boldsymbol{Y}}[i, j],$$

where $/$ denotes element-wise division. We then compute the mean relative equivariance error over all instances in the validation set, which equals to 5.17%. This gives concrete justification to enforce approximate equivariance in the traffic flow prediction problems.

## D.4 Application: Human Pose Estimation

### D.4.1 Equivariant Layer for Human Skeleton Graph

We apply the constructions in Section A.1 to our human skeleton graph. We first show how to parameterize all linear $\mathcal{A}_G$-equivariant functions. Observe that $\mathcal{A}_G \cong (\mathcal{S}_2)^2 = \{e, a, l, al\}$, where the nontrivial actions correspond to the **a**rm flip with respect to the spine, the **l**eg flip with respect to the spine, and their composition. To fix ideas, we first treat both input and output graph signals as vectors, and construct $\mathcal{A}_G$-equivariant linear maps $f : \mathbb{R}^{16} \to \mathbb{R}^{16}$.

Step 1: Obtain the character table for $(\mathcal{S}_2)^2$

|          | $e$ | $a$ | $l$ | $al$ |
| -------- | --- | --- | --- | ---- |
| $\chi_e$ | 1   | 1   | 1   | 1    |
| $\chi_2$ | 1   | 1   | $-1$ | $-1$ |
| $\chi_3$ | 1   | $-1$ | 1   | $-1$ |
| $\chi_4$ | 1   | $-1$ | $-1$ | 1   |

Table 2: Character table for $(\mathcal{S}_2)^2$

Step 2: Construct the basis for isotypic decomposition. Here we choose to index the leg joint pairs as $(1, 4), (2, 5), (3, 6)$, arm joint pairs as $(10, 13), (11, 14), (12, 15)$, and spline joints $0, 7, 8, 9$.

$$B = [\mathcal{B}(P_{\chi_e}); \mathcal{B}(P_{\chi_2}); \mathcal{B}(P_{\chi_3}); \mathcal{B}(P_{\chi_4})] \text{ where}$$

$$\mathcal{B}(P_{\chi_e}) = [(e_1 + e_4)/\sqrt{2}; \dots; (e_{12} + e_{15})/\sqrt{2}; e_0; e_7; e_8; e_9] \in \mathbb{R}^{16 \times 10}.$$

$$\mathcal{B}(P_{\chi_2}) = [(e_1 - e_4)/\sqrt{2}; (e_2 - e_5)/\sqrt{2}; (e_3 - e_6)/\sqrt{2}] \in \mathbb{R}^{16 \times 3};$$

$$\mathcal{B}(P_{\chi_3}) = [(e_{10} - e_{13})/\sqrt{2}; (e_{11} - e_{14})/\sqrt{2}; (e_{12} - e_{15})/\sqrt{2}] \in \mathbb{R}^{16 \times 3};$$

$$\mathcal{B}(P_{\chi_4}) = \emptyset \tag{66}$$

Step 3: Parameterize $f : \mathbb{R}^{16} \to \mathbb{R}^{16}$ by $f : \mathcal{B}(P_{\chi_e}) \to \mathcal{B}(P_{\chi_e})$ and $f : \mathcal{B}(P_{\chi_2}) \to \mathcal{B}(P_{\chi_2})$, i.e. for all $v \in \mathbb{R}^{16}$, let $v = \mathcal{B}(P_{\chi_e}) \, \boldsymbol{c_e} + \mathcal{B}(P_{\chi_2}) \, \boldsymbol{c_2} + \mathcal{B}(P_{\chi_3}) \, \boldsymbol{c_3}$, then

$$f(v) = W_e \, \boldsymbol{c_e} + W_2 \, \boldsymbol{c_2} + W_3 \, \boldsymbol{c_3}, \tag{67}$$

where $W_e \in \mathbb{R}^{10 \times 10}, W_2 \in \mathbb{R}^{3 \times 3}, W_3 \in \mathbb{R}^{3 \times 3}$ are (learnable) weight matrices. Now $f$ expresses all linear, equivariant maps w.r.t $(\mathcal{S}_2)^2$.

The following calculation based on $f : \mathbb{R}^{16} \to \mathbb{R}^{16}$ shows how much degree of freedom (measured by learnable parameters) is gained by relaxing the symmetry from global (group $\mathcal{S}_{16}$), exact $\mathcal{A}_G \cong (\mathcal{S}_2)^2$, to trivial group (i.e., no symmetry).

$$f_{\mathcal{S}_{16}} = w \, \mathbf{I}_{16} + w'(\mathbf{1} - \mathbf{I}_{16}), \quad (2 \text{ parameters}); \tag{68}$$

$$f_{\mathcal{A}_G} = W_e \oplus W_2 \oplus W_3, \quad (118 \text{ parameters on the isotypic components}); \tag{69}$$

$$f_{\text{triv.}} = W, \quad (256 \text{ parameters}). \tag{70}$$

To parameterize equivariant linear function $f : \mathbb{R}^{16 \times d} \to \mathbb{R}^{16 \times d'}$, we proceed by decoupling the input space into $\mathbb{R}^{10 \times d}, \mathbb{R}^{3 \times d}, \mathbb{R}^{3 \times d}$ and the output space into $\mathbb{R}^{10 \times d'}, \mathbb{R}^{3 \times d'}, \mathbb{R}^{3 \times d'}$. Now the learnable weight matrices for multidimensional input/output become $W_e \in \mathbb{R}^{10d \times 10d'}, W_2 \in \mathbb{R}^{3d \times 3d'}, W_3 \in \mathbb{R}^{3d \times 3d'}$. The construction is summarized in Algorithm 2.

---

**Algorithm 2** Equivariant layer $f_{\mathcal{A}_G} : \mathbb{R}^{16 \times d} \to \mathbb{R}^{16 \times d'}$ for $\mathcal{A}_G \cong (\mathcal{S}_2)^2$

---

**Require:** The basis $B \in \mathbb{R}^{16 \times 16}$ in (66) for isotypic decomposition of $\mathcal{A}_G = (\mathcal{S}_2)^2$, input $h^{(l)} \in \mathbb{R}^{16 \times d}$.

**Initialize:** The learnable weights $W_e^{(l)} \in \mathbb{R}^{10d' \times 10d}; W_2^{(l)}, W_3^{(l)} \in \mathbb{R}^{3d' \times 3d}; M^{(l)} \in \mathbb{R}^{16 \times 16}$.

1. Project $h^{(l)}$ to the isotypic component: $z^{(l)} = B^\top h^{(l)}$;
2. Perform block-wise linear transformation:
   - $z_e = W_e \, \text{flatten}(z_{[:,:10]}^{(l)})$
   - $z_2 = W_2 \, \text{flatten}(z_{[:,10:13]}^{(l)})$
   - $z_3 = W_3 \, \text{flatten}(z_{[:,13:]}^{(l)})$
   - $z^{(l+1)} = \text{concat}[z_e; z_2; z_3] \in \mathbb{R}^{16 \times d'}$
3. Project back to the standard basis: $\bar{h}^{(l+1)} = B \, z^{(l+1)}$.
4. Perform pointwise nonlinearity: $h^{(l+1)} = \sigma(\bar{h}^{(l+1)})$.

**return** $h^{(l+1)}$

---

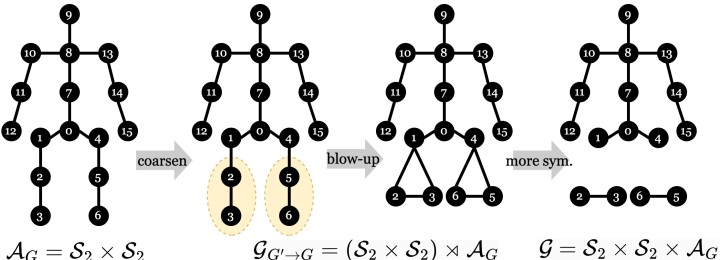

$$\mathcal{A}_G = \mathcal{S}_2 \times \mathcal{S}_2 \qquad \mathcal{G}_{G' \to G} = (\mathcal{S}_2 \times \mathcal{S}_2) \rtimes \mathcal{A}_G \qquad \mathcal{G} = \mathcal{S}_2 \times \mathcal{S}_2 \times \mathcal{A}_G$$

Figure 9: Human skeleton graph $G$, its coarsened graph $G'$ (clustering leg joints), and blow-up of $G'$

### D.4.2 Experiment Details

**Data.** We use the standard benchmark dataset, Human3.6M [77], with the same protocol as in [78]: We train the models on $1.56\mathrm{M}$ poses (from human subjects $S1, S5, S6, S7, S8$) and evaluate them on $0.54\mathrm{M}$ poses (from human subjects $S9, S11$). We use the method described in [88] to normalize the inputs (2D joint poses) to $[-1, 1]$ and align the targets (3d joint poses) with the root joint.

**Model.** We give a detailed description of $\mathcal{G}$-Net and its variants used in the experiments. Figure inset illustrates the architecture of $\mathcal{G}$-Net.

For the human skeleton graph with $N = 16$, we have $f_{\mathcal{G}} : \mathbb{R}^{16 \times d} \to \mathbb{R}^{16 \times k}$, where $d, k$ represent the input dimension and output dimension (for each layer). Let $f_{\mathcal{G}}[i,j] : \mathbb{R}^{16} \to \mathbb{R}^{16}$ denote its $(i,j)$-th slice.

1. $\mathcal{G}$-Net with strict equivariance using equivariant linear map $f_{\mathcal{G}}$ (see Table 3):

- $\mathcal{S}_{16}$: $f_{\mathcal{S}_{16}}[i,j] \in \mathbb{R}^{16 \times 16}$ is a diagonal matrix, with one learnable scalar $a$ on diagonal and another learnable scalar $b$ off diagonal.
- Relax-$\mathcal{S}_{16}$: We relax $f_{\mathcal{S}_{16}}[i,j]$ by having 16 different pairs of scalars $(a_i, b_i), i \in [16]$, such that each node $i$ can map to itself and communicate to its neighbors in a different way (controlled by $(a_i, b_i)$), while still treat all neighbors equally (by using the same $b_i$ for nodes $j \neq i$).
- $\mathcal{A}_G = \mathcal{S}_2{}^2$: We use Algorithm 2.
- Trivial: We allow $f[i,j] \in \mathbb{R}^{16 \times 16}$ to be arbitrary, i.e., it has $16 \times 16$ learnable scalars.

We remark that for $\mathcal{S}_{16}$ and Relax-$\mathcal{S}_{16}$, we implement them by tying weights; for $\mathcal{A}_G$, we implement them by projecting to isotypic component as shown in Algorithm 2.

2. $\mathcal{G}$-Net augmented with graph convolution $A f_{\mathcal{G}}(x)$, denoted as $\mathcal{G}$-Net(gc) (see Table 3): We apply the equivariant linear map $f_{\mathcal{G}}$ in 1. and obtain the output $f_{\mathcal{G}}(x) \in \mathbb{R}^{16 \times k}$; We then apply graph convolution by multiplication from the left, i.e., $A f_{\mathcal{G}}(x) \in \mathbb{R}^{16 \times k}$.

3. $\mathcal{G}$-Net augmented with graph convolution and learnable edge weights, denoted as $\mathcal{G}$-Net(gc+ew) (see Table 4): We further learn the edge weights for the adjacency matrix $A$, by $\mathrm{softmax}(M \odot A)$ where $M \in \mathbb{R}^{16}$ represents the learnable edge weights, and $M_{i,j}$ is nonzero when $A_{i,j} \neq 0$ and $0$ elsewhere. This is inspired from SemGCN [78]. Besides the groups discussed above, we also implemented Relax-$(\mathcal{S}_6)^2$ which corresponds to tying weights among the coarsened graph orbits, consists of 4 spline nodes (singleton orbits) and 2 orbits for the left/right arm and leg nodes.

4. $\mathcal{G}$-Net augmented with graph locality constraints $(A \odot f_{\mathcal{G}})(x)$ and learnable edge weights, denoted as $\mathcal{G}$-Net(pt+ew) (see Table 3): We perform pointwise multiplication $A \odot f_{\mathcal{G}}[i,j]$ at each $(i,j)$-th slice of $f_{\mathcal{G}}$. In practice, we also allow learnable edge weights as done in 3.

**Experimental Set-up.** We design $\mathcal{G}$-Net to have 4 layers (with batch normalization and residual connections in between the hidden layers), 128 hidden units, and use ReLU nonlinearity. This allows $\mathcal{G}$-Net(gc+ew) to recover SemGCN [78] when choosing $\mathcal{G} = \mathcal{S}_{16}$. We train our models for at most 30

epochs with early stopping. For comparison purpose, we use the same optimization routines as in SemGCN [78] and perform the hyper-parameter search of learning rates $\{0.001, 0.002\}$.

**Evaluation.** Table 3 shows results of $\mathcal{G}$-Net and its variants when varying the choice of $\mathcal{G}$. We observe that using the automorphism group $\mathcal{A}_G$ does not give the best performance, while imposing no symmetries (Trivial) or a relaxed version of $\mathcal{S}_{16}$ yields better results. Here, enforcing no symmetry achieves better performance since the human skeleton graph is very small with 16 nodes only. As shown in other experiments with larger graphs (e.g. image inpainting), enforcing symmetries indeed yields better performance.

Table 3: 3D human pose prediction using $\mathcal{G}$-Net and its variants. Error ($\pm$ std) measured by Mean Per-Joint Position Error (MPJPE) and MPJPE after rigid alignment (P-MPJPE) across 3 runs. All methods use the same hidden dimension $d = 128$. Bold type indicates the top-2 performance among each variant. "NA" indicates the loss fails to converge.

| **MPJPE** $\downarrow$ | $\mathcal{S}_{16}$ | Relax-$\mathcal{S}_{16}$ | $\mathcal{A}_G = (\mathcal{S}_2)^2$ | Trivial |
|---|---|---|---|---|
| $\mathcal{G}$-Net | NA | $\mathbf{47.97 \pm 0.47}$ | $48.30 \pm 0.69$ | $\mathbf{42.86 \pm 0.64}$ |
| $\mathcal{G}$-Net(gc) | NA | $54.50 \pm 4.33$ | $\mathbf{49.40 \pm 1.37}$ | $\mathbf{43.24 \pm 0.82}$ |
| $\mathcal{G}$-Net(pt+ew) | $41.54 \pm 0.47$ | $\mathbf{40.44 \pm 0.61}$ | $40.63 \pm 0.26$ | $\mathbf{38.41 \pm 0.31}$ |

| **P-MPJPE** $\downarrow$ | $\mathcal{S}_{16}$ | Relax-$\mathcal{S}_{16}$ | $\mathcal{A}_G = (\mathcal{S}_2)^2$ | Trivial |
|---|---|---|---|---|
| $\mathcal{G}$-Net | NA | $\mathbf{36.45 \pm 0.56}$ | $37.17 \pm 0.59$ | $\mathbf{32.59 \pm 0.62}$ |
| $\mathcal{G}$-Net(gc) | NA | $40.61 \pm 0.99$ | $\mathbf{37.62 \pm 1.32}$ | $\mathbf{33.05 \pm 0.81}$ |
| $\mathcal{G}$-Net(pt+ew) | $32.31 \pm 0.03$ | $\mathbf{31.11 \pm 0.68}$ | $31.35 \pm 0.14$ | $\mathbf{29.68 \pm 0.22}$ |

**Additional Evaluation.** Table 4 shows the experiments when we keep the number of parameters roughly the same across different choices of $\mathcal{G}$.

Table 4: 3D human pose prediction using $\mathcal{G}$-Net(gc+ew), where the models induced from each choice of $\mathcal{G}$ are set to have roughly the same number of parameters. $d$ denotes the number of hidden units.

| $\mathcal{G}$-Net | Number of Parameters | Number of Epochs | MPJPE | P-MPJPE |
|---|---|---|---|---|
| $\mathcal{S}_{16}$ | 0.27M ($d = 128$) | 50 | 43.48 | 34.96 |
| Relax-$\mathcal{S}_{16}$ | 0.27M ($d = 32$) | 20 | $\mathbf{40.08}$ | $\mathbf{32.08}$ |
| $\mathcal{A}_G = (\mathcal{S}_2)^2$ | 0.22M ($d = 16$) | 30 | 44.10 | 34.12 |
| Trivial | 0.22M ($d = 10$) | 30 | 45.05 | 34.79 |

