# A   Graph Neural Networks (GNNs)

We consider standard message-passing graph neural networks (MPNNs) [19–21] defined as follows. A $L$-layer MPNN maps input $X \in \mathbb{R}^{N \times d}$ to output $Y \in \mathbb{R}^{N \times k}$ following an iterative scheme: At initialization, $\mathbf{h}^{(0)} = X$; At each iteration $l$, the embedding for node $i$ is updated to

$$\mathbf{h}_i^{(l)} = \phi \left( \mathbf{h}_i^{(l-1)}, \sum_{j \in \mathcal{N}(i)} \psi \left( \mathbf{h}_i^{(l-1)}, \mathbf{h}_j^{(l-1)}, A_{[i,j]} \right) \right), \tag{7}$$

where $\phi, \psi$ are the update and message functions, $\mathcal{N}(i)$ denotes the neighbors of node $i$, and $A_{[i,j]}$ represents the $(i, j)$-edge weight. MPNNs typically have two key design features: (1) $\phi, \psi$ are *shared* across all nodes in the graph, typically chosen to be a linear transformation or a multi-layer perceptions (MLPs), known as *global weight sharing*; (2) the graph $A$ is used for (spatial) convolution.

# B   Parameterization of Linear Equivariant Maps

We consider a group $\mathcal{G}$ acting on spaces $\mathcal{X}$ and $\mathcal{Y}$ via representations $\phi$ and $\psi$, respectively. Our goal is to find the linear equivariant maps $f : \mathcal{X} \to \mathcal{Y}$ such that $f(\phi(g)x) = \psi(g)f(x)$ for all $g \in \mathcal{G}$ and $x \in \mathcal{X}$. The standard way to do this, used extensively in the equivariant machine learning literature (e.g. [40, 43]), is to decompose $\phi$ and $\psi$ in irreducibles and use Schur's lemma.

In a nutshell, a group representation $\varphi$ is an homomorphism $\mathcal{G} \to \mathrm{GL}(V)$ (sometimes mathematicians say that $V$ is a representation of $\mathcal{G}$, but we need to know the homomorphism $\varphi$ too). One way to interpret the group homomorphism (i.e. $\varphi(gh) = \varphi(g) \circ \varphi(h)$) is that the group multiplication corresponds to the composition of linear invertible maps (i.e. matrix multiplication). A linear subspace $W$ of $V$ is said to be a subrepresentation of $\varphi$ if $\varphi(\mathcal{G})(W) \subset W$. A irreducible representation is one that only has itself and the trivial subspace as subrepresentations.

Schur's lemma states that if $V$, $W$ are vector spaces over $\mathbb{C}$ and $\varphi_V$, $\varphi_W$ are irreducible representations, then either (1) $\varphi_V$ and $\varphi_W$ are not isomorphic as representations (and the only linear equivariant map between $V$, $W$ is the zero map), or (2) $\varphi_V$ and $\varphi_W$ are isomorphic and the only non-trivial equivariant maps are of the form $\lambda I$ where $\lambda \in \mathbb{C}$ and $I$ is the identity (See Chapter 1 of [60]).

Now given $\mathcal{G}$ acting on spaces $\mathcal{X}$ and $\mathcal{Y}$ via representations $\phi$ and $\psi$, respectively. Then one can decompose $\phi$ and $\psi$ in irreducibles over $\mathbb{C}$

$$\phi = \oplus_{k=1}^{\ell} a_k \mathcal{T}_k \quad \psi = \oplus_{k=1}^{\ell} b_k \mathcal{T}_k$$

(this notation assumes the same irreducibles appear in both decompositions, which can be done if we allow some of the $a_k$ and $b_k$ to be zero). And then one can parameterize the equivariant maps by having one complex parameter per irreducible that appears in both decompositions. These ideas can be applied to real spaces.

Then finding the linear equivariant maps reduces to decomposing the corresponding representations in irreducibles. In the next sections we explain in detail how to do this for the specific problems described in this paper. The appendix is organized as follows: We first show how to parameterize equivariant linear layers for Abelian group (Section B.1.1), and then provide the end-to-end design of equivariant graph networks $\mathcal{G}$-Net (Section B.3).

## B.1   Equivariant Linear Maps via Isotypical Decomposition

In this section, we assume that the graph adjacency matrix $A$ has distinct eigenvalues $\lambda_1 > \lambda_2 > \ldots > \lambda_n$. Then $\mathcal{A}_G$ is an Abelian group (Lemma 3.8.1, notes). Under this assumption, we present the construction of approximately equivariant graph networks using isotypical decomposition (i.e. decomposition into isomorphism classes of irreducible representations) and group characters. We remark that such construction extends to non-Abelian groups and refer the interested reader to [68], but we omit it here for the ease of exposition.

### B.1.1 Equivariant Linear Layers for Abelian Group

We consider the simplest setting where $f : \mathbb{R}^N \to \mathbb{R}^N$ is a linear function that maps signals on the node level. Let $x \in \mathbb{R}^N$ be the node features, then equivariance requires

$$f(g\,x) = g\,f(x) \quad \text{for all } g \in \mathcal{A}_G. \tag{8}$$

To construct linear equivariant functions $f$, our roadmap is outlined as follows:

1. Decompose the vector space $\mathcal{X} = \mathbb{R}^N$ into a sum of components such that different components cannot be mapped to each other equivariantly (also known as the isotypic decomposition);

2. Given $\mathcal{X} = \oplus_i \mathcal{X}_i$ an isotypic representation, we then parameterize $f$ by linear maps at each $\mathcal{X}_i$ such that for all $i$, $f(\mathcal{X}_i) \subseteq \mathcal{X}_i$.

To this end, we need the following definitions.

**Definition 5.** *($\mathcal{G}$-module, [68, Defn 1.3.1]) Let $\mathcal{X}$ be a vector space and $\mathcal{G}$ be a group. We say the vector space $\mathcal{X}$ is a $\mathcal{G}$-module or $\mathcal{X}$ carries a representation of $\mathcal{G}$ if there is a group homomorphism $\rho : \mathcal{G} \to GL(\mathcal{X})$, where $GL$ denotes the General Linear group. Equivalently, if the following holds:*

*1. $gv \in \mathcal{X}$,*

*2. $g(cv + dw) = c(gv) + d(gw)$,*

*3. $(gh)v = g(hv)$,*

*4. $ev = v$*

*for all $g, h \in \mathcal{G}; v, w \in \mathcal{X}$ and scalars $c, d \in \mathbb{C}$ ($e \in \mathcal{G}$ denotes the identity element).*

In what follows, we consider $\mathcal{X} = \mathbb{R}^N$ carries a representation of $G$.

**Definition 6.** *(Group characters) Let $X(g), g \in \mathcal{G}$ be a matrix representation of a group element. Then the character of $X$ is $\chi(g) := \operatorname{tr} X(

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

where $f_{kl}$ are block matrices, and $a_k, b_k, c_{kl}$ are scalars where $c_{kl} = c_{lk}$ if and only if the coarsened nodes $k, l \in G'$ are in the same group orbit. Figure [2] illustrates the block structure of $f$. This is due to (1) $f_{kk}$ is a linear permutation-equivariant function if and only if its diagonal elements are the same and its off-diagonal elements are the same ([[34], Lemma 3.]); (2) $f_{kl}$ for $k \neq l$ is a constant matrix since nodes within a cluster are indistinguishable, and $c_{kl}$ needs to satisfy the symmetry of $\overline{\mathcal{A}}_{G'}$.

Finally, we illustrate the linear equivariant layer for two-cluster graph coarsening. Without loss of generality, assume that the adjacency matrix $A$ and the node signals $X$ are ordered according to the cluster assignment (e.g., $X_{[1:|V_1|]}$ are node features for the first cluster, etc). Let $X_{(1)}, X_{(2)}$ denote the node features for the first and second cluster, $W_{(1)}^s, W_{(2)}^s$ denote the weights on the block diagonal for self-feature transformation, $W_{(1)}^n, W_{(2)}^n$ denote the weights on the block diagonal for within-cluster neighbors, and $W_{(12)}^n, W_{(21)}^n$ denote the weights off the block diagonal for across-cluster neighbors. Let $I$ denote the identity matrix, and $\mathbf{1}_{(1)}, \mathbf{1}_{(2)}$ denote the all-ones matrices with the same size as the corresponding cluster. Recall $\odot$ denotes the element-wise multiplication of two matrices. Then the linear equivariant layer is parameterized as

$$A \odot I \begin{bmatrix} X_{(1)} W_{(1)}^s \\ X_{(2)} W_{(2)}^s \end{bmatrix} + A \odot \left( \begin{bmatrix} \mathbf{1}_{(1)} & 0 \\ 0 & \mathbf{1}_{(2)} \end{bmatrix} - I \right) \begin{bmatrix} X_{(1)} W_{(1)}^n \\ X_{(2)} W_{(2)}^n \end{bmatrix} + A \odot \begin{bmatrix} 0 & \mathbf{1}_{(2)} \\ \mathbf{1}_{(1)} & 0 \end{bmatrix} \begin{bmatrix} X_{(1)} W_{(12)}^n \\ X_{(2)} W_{(21)}^n \end{bmatrix}. \tag{20}$$

## B.3 Equivariant Layer for Human Skeleton Graph

We now apply the constructions above to our human skeleton graph described in Section [5.1].

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

---

[2] In the main paper, the irreducible error term $\sigma_{\xi}^{2}$ is missing. We fix this in the Appendix and the revised version. The risk gain is of a factor $\frac{N^{2}dk-(\chi_{\psi}\mid\chi_{\phi})}{n-1}$.

*Proof.* Recall $X, Y$ denote the test sample. We denote the risk of the least-squares estimator *conditional on the training data* $\boldsymbol{X} \in \mathbb{R}^{n \times Nd}$ as $\mathcal{R}(\hat{\Theta} \mid \boldsymbol{X})$, which has the following bias-variance decomposition:

$$\mathcal{R}(\hat{\Theta} \mid \boldsymbol{X}) = \mathbb{E}\left[\|Y - \hat{\Theta}^\top X\|_F^2 \mid \boldsymbol{X}\right] \tag{40}$$

$$= \mathbb{E}\left[\|\Theta^\top X + \xi - \hat{\Theta}^\top X\|_F^2 \mid \boldsymbol{X}\right] \tag{41}$$

$$= \mathbb{E}\left[\|(\Theta - \hat{\Theta})^\top X\|_F^2 \mid \boldsymbol{X}\right] + \sigma_\xi^2, \tag{42}$$

where the last equality follows from $\xi$ being zero mean and independent with $X$. The second term $\sigma_\xi^2$ is also known as *irreducible error*. We decompose the first term into

$$\mathbb{E}\left[\|(\Theta - \hat{\Theta})^\top X\|_F^2 \mid \boldsymbol{X}\right] = \mathbb{E}\left[\|(\Theta - \mathbb{E}[\hat{\Theta}])^\top X\|_F^2 + \|(\mathbb{E}[\hat{\Theta}] - \hat{\Theta})^\top X\|_F^2 \mid \boldsymbol{X}\right]. \tag{43}$$

Recall that $\hat{\Theta} \overset{a.e.}{=} (\boldsymbol{X}^\top \boldsymbol{X})^{-1}\boldsymbol{X}^\top \boldsymbol{Y} = (\boldsymbol{X}^\top \boldsymbol{X})^{-1}\boldsymbol{X}^\top(\boldsymbol{X}\Theta + \xi) = \Theta + (\boldsymbol{X}^\top \boldsymbol{X})^{-1}\boldsymbol{X}^\top \xi$. Thus $\mathbb{E}[\hat{\Theta}] = \Theta$ and (43) simplifies to $\mathbb{E}\left[\|(\mathbb{E}[\hat{\Theta}] - \hat{\Theta})^\top X\|_F^2 \mid \boldsymbol{X}\right]$.

We finish computing the risk by taking expectation over $\boldsymbol{X}$, and using $\mathbb{E}[\hat{\Theta}] - \hat{\Theta} = (\boldsymbol{X}^\top \boldsymbol{X})^{-1}\boldsymbol{X}^\top \xi$,

$$\mathbb{E}\left[\|Y - \hat{\Theta}^\top X\|_F^2\right] = \mathbb{E}\left[\mathcal{R}(\hat{\Theta} \mid \boldsymbol{X})\right] \tag{44}$$

$$= \mathbb{E}_{\boldsymbol{X}}\left[\mathbb{E}_{X,\xi}\left[\|(\mathbb{E}[\hat{\Theta}] - \hat{\Theta})^\top X\|_F^2 \mid \boldsymbol{X}\right]\right] + \sigma_\xi^2 \tag{45}$$

$$= \mathbb{E}\left[\|\left((\boldsymbol{X}^\top \boldsymbol{X})^{-1}\boldsymbol{X}^\top \xi\right)^\top X\|_F^2\right] + \sigma_\xi^2 \tag{46}$$

$$= \sigma_\xi^2 \operatorname{tr}\left(\mathbb{E}[(\boldsymbol{X}^\top \boldsymbol{X})^{-1}]\sigma_X^2 I\right) + \sigma_\xi^2. \tag{47}$$

By [72, Lemma 2.3], for $n > Nd + 1$, $\mathbb{E}[(\boldsymbol{X}^\top \boldsymbol{X})^{-1}] = \frac{Nd}{n-Nd-1}I$. Putting this in (47) completes the proof. $\qquad\square$

## C.2  Proofs of Generalization with Approximate Symmetries

**Corollary 3** (Risk Gap via Graph Coarsening). Let $\mathcal{X} = \mathbb{R}^{N \times d}, \mathcal{Y} = \mathbb{R}^{N \times k}$ be the input and output graph signal spaces on a fixed graph $G$. Let $X \sim \mu$ where $\mu$ is a $\mathcal{S}_N$-invariant distribution on $\mathcal{X}$. Let $Y = f^*(X) + \xi$, where $\xi \in \mathbb{R}^{N \times k}$ is random, independent of $X$ with zero mean and finite variance, and $f^* : \mathbb{R}^{N \times d} \to \mathbb{R}^{N \times k}$ be an approximately equivariant mapping with equivariance rate $\kappa$. Then, for any $G'$ that coarsen $G$ up to error $\epsilon$, for any $f \in V$, we have

$$\Delta(f, \bar{f}_{\mathcal{G}_{G \to G'}}) = \underbrace{-2\langle f^*, f^\perp_{\bar{\mathcal{G}}_{G \to G'}}\rangle_\mu}_{\text{mismatch}} + \underbrace{\left\|f^\perp_{\bar{\mathcal{G}}_{G \to G'}}\

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

Next, we show that $f^*$ is approximately equivariant in a restricted sense, where we limit ourselves to a symmetry group

$$\mathcal{G}_{G \to G'} = \mathcal{S}_{c_1} \times \mathcal{S}_{c_2} \ldots \times \mathcal{S}_{c_M}$$

in Definition 3, without the symmetries of $\overline{\mathcal{A}}_{G'}$.

Equation (59) leads to

$$\|f^*(s_G) - \mathcal{Q}_{\mathcal{G}_{G \to G'}}(f^*)(s_G)\|_1 = \|f^*(s_G) - \frac{1}{|\mathcal{G}_{G \to G'}|} \sum_{g \in \mathcal{G}_{G \to G'}} g^{-1} f^*(g s_G)\|_1 \tag{60}$$

$$\leq \frac{1}{|\mathcal{G}_{G \to G'}|} \sum_{g \in \mathcal{G}_{G \to G'}} \|f^*(s_G) - g^{-1} f^*(g s_G)\|_1 < \epsilon'. \tag{61}$$

Since for any $q \in L^2[0, 1] \cap L^\infty[0, 1]$ we have $\|q\|_2^2 \leq \|q\|_\infty \|q\|_1$, we can bound

$$\|f^*(s_G) - \mathcal{Q}_{\mathcal{G}_{G \to G'}}(f^*)(s_G)\|_2 < \sqrt{2\|f^*(s_G)\|_\infty} \sqrt{\epsilon'}.$$

Denote $\|f^*\|_\infty := \int \|f^*(s_G)\|_\infty d\mu(s_G)$, and suppose that $\|f^*\|_\infty$ is finite. Hence, if $\mu$ is a probability measure, we have

$$\left\|f^* - \mathcal{Q}_{\mathcal{G}_{G \to G'}}(f^*)\right\|_\mu < \sqrt{2\|f^*\|_\infty} \sqrt{\epsilon'}.$$

This shows a modified version of approximate equivariance, where the approximation rate is also a function of the size of the graph $N$, and goes to zero as $N \to \infty$ and $\epsilon \to 0$.

In future work, we will extend this example to more general metric space $\mathcal{M}$ and to non-trivial symmetry groups $\overline{\mathcal{A}}_{G'}$. Intuitively, most random geometric graphs are "close to asymmetric." This means that for "most" $G'$, the symmetries of $\overline{\mathcal{A}}_{G'}$ can only permute between nodes connected by an edge, and so are the symmetries of $\mathcal{G}_{G \to G'}$. For this, we need to extend Definition 9 by treating $G'$ probabilistically.

# E  Experiment Details

The source code will be made available in the final version of the paper. All experiments were conducted on a server with 256 GB RAM and 4 NVIDIA RTX A5000 GPU cards.

## E.1 Application: Human Pose Estimation

**Data.** We use the standard benchmark dataset, Human3.6M [65], with the same protocol as in [66]: We train the models on 1.56M poses (from human subjects $S1, S5, S6, S7, S8$) and evaluate them on 0.54M poses (from human subjects $S9, S11$). We use the method described in [74] to normalize the inputs (2D joint poses) to $[-1, 1]$ and align the targets (3d joint poses) with the root joint.

**Graph Networks with Equivariant Modules.** We give detailed description of $\mathcal{G}$-Net and its variants used in the experiments. Figure inset illustrates the architecture of $\mathcal{G}$-Net. For the human skeleton graph with $N = 16$, we have $f_{\mathcal{G}} : \mathbb{R}^{16 \times d} \to \mathbb{R}^{16 \times k}$, where $d, k$ represent the input dimension and output dimension (for each layer). Let $f_{\mathcal{G}}[i,j] : \mathbb{R}^{16} \to \mathbb{R}^{16}$ denote its $(i,j)$-th slice.

1. $\mathcal{G}$-Net with strict equivariance using equivariant linear map $f_{\mathcal{G}}$ (see Table 5):

- $\mathcal{S}_{16}$: $f_{\mathcal{S}_{16}}[i,j] \in \mathbb{R}^{16 \times 16}$ is a diagonal matrix, with one learnable scalar $a$ on diagonal and another learnable scalar $b$ off diagonal.

- Relax-$\mathcal{S}_{16}$: We relax $f_{\mathcal{S}_{16}}[i,j]$ by having 16 different pairs of scalars $(a_i, b_i), i \in [16]$, such that each node $i$ can map to itself and communicate to its neighbors in a different way (controlled by $(a_i, b_i)$), while still treat all neighbors equally (by using the same $b_i$ for nodes $j \neq i$).

- $(\mathcal{S}_2)^6$: We use Algorithm 2 while replacing $\mathcal{A}_G$ with the symmetry group on a disconnected graph $G_0$ consists of the orbits in $G$, i.e. $G_0$ has the same nodes as $G$, but only retaining the edges among $(1,4), (2,5), (3,6), (10,13), (11,14), (12,15)$.

- $\mathcal{A}_G$: We use Algorithm 2.

- $\mathcal{S}_2$: We use Algorithm 2 while replacing $\mathcal{A}_G$ with $\mathcal{S}_2$ representing the bilateral symmetry on the human skeleton graph (i.e., the left arms and legs must flip together, similarly for the right arms and legs).

- Trivial: We allow $f[i,j] \in \mathbb{R}^{16 \times 16}$ to be arbitrary, i.e., it has $16 \times 16$ learnable scalars.

We remark that for $\mathcal{S}_{16}$ and Relax-$\mathcal{S}_{16}$, we implement them by tying weights; for $(\mathcal{S}_2)^6, \mathcal{A}_G, \mathcal{S}_2$, we implement them by projecting to isotypic component as shown in Algorithm 2.

2. $\mathcal{G}$-Net augmented with graph convolution $A f_{\mathcal{G}}(x)$, denoted as $\mathcal{G}$-Net(gc) (see Table 5): We apply the equivariant linear map $f_{\mathcal{G}}$ in 1. and obtain the output $f_{\mathcal{G}}(x) \in \mathbb{R}^{16 \times k}$; We then apply graph convolution by multiplication from the left, i.e., $A f_{\mathcal{G}}(x) \in \mathbb{R}^{16 \times k}$.

3. $\mathcal{G}$-Net augmented with graph convolution and learnable edge weights, denoted as $\mathcal{G}$-Net(gc+ew) (see Table 1[3]): We further learn the edge weights for the adjacency matrix $A$, by $\texttt{softmax}(M \odot A)$ where $M \in \mathbb{R}^{16}$ represents the learnable edge weights, and $M_{i,j}$ is nonzero when $A_{i,j} \neq 0$ and 0 elsewhere. This is inspired from SemGCN [66]. Besides the groups discussed in 1., we also implemented Relax-$(\mathcal{S}_6)^2$ which corresponds to tying weights among the coarsened graph orbits, consists of 4 spline nodes (singleton orbits) and 2 orbits for the left/right arm and leg nodes.

4. $\mathcal{G}$-Net augmented with graph locality constraints $(A \odot f_{\mathcal{G}})(x)$, denoted as $\mathcal{G}$-Net(pt) (see Table 5): We perform pointwise multiplication $A \odot f_{\mathcal{G}}[i,j]$ at each $(i,j)$-th slice of $f_{\mathcal{G}}$. In practice, we also allow learnable edge weights as done in 3.

**Experimental Set-up.** We design $\mathcal{G}$-Net to have 4 layers (with batch normalization and residual connections in between the hidden layers), 128 hidden units, and use ReLU nonlinearity. This allows $\mathcal{G}$-Net(gc+ew) to recover SemGCN [66] when choosing $\mathcal{G} = \mathcal{S}_{16}$. We train our models for maximally 30 epochs with early stopping. For comparison purpose, we use the same optimization routines as in SemGCN [66] and perform the hyper-parameter search of learning rates $\{0.001, 0.002\}$.

**Evaluation.** Table 5 shows results of $\mathcal{G}$-Net and its variants when varying the choice of $\mathcal{G}$. We observe that using the automorphism group $\mathcal{A}_G$ does not give the best performance, while imposing no symmetries (Trivial) or a relaxed version of $\mathcal{S}_{16}$ yields better results.

---

[3]There is a typo in Table 1, where $(\mathcal{S}_2)^6$ should be corrected to Relax-$\mathcal{S}_{16}$, and $(\mathcal{S}_6)^2$ should be corrected to Relax-$(\mathcal{S}_6)^2$.