# OpenReview forum: "Approximately Equivariant Graph Networks"
_NeurIPS.cc/2023/Conference — NeurIPS 2023 poster_

### Official Review · Reviewer_KAcz · 2023-06-28

**Soundness:** 3 good
**Presentation:** 2 fair
**Contribution:** 4 excellent
**Rating:** 6
**Confidence:** 3

**Summary:**

This work discusses a non-trivial case of equivariant graph networks. In this scenario, the input graph $G$ remains fixed, and therefore, the relevant permutations that act on the graph signals are the automorphisms of $G$. Under this assumption, the authors describe a bias-variance tradeoff with respect to the symmetry group to which the hypothesis class is restricted. Since the automorphism group tends to be trivial in large graphs, the authors introduce the concept of approximated symmetry by considering a symmetry group associated with a coarser version of the graph. The main result of the paper formalizes the bias-variance tradeoff in relation to the approximated symmetries.

**Strengths:**

* The paper introduces a novel perspective and analysis of equivariant graph networks, both in terms of the fixed graph scenario and the approximated equivariance via graph coarsening.

* The use of Graphons to define a metric over graph spaces is interesting and could have many applications.

* The theoretical claims are very clear and explained in a detailed manner.

**Weaknesses:**

* An explanation regarding the clustering methods is missing in the main text. The method section in the paper does not suggest how to coarsen a graph G, therefore it would be helpful to explain how it is done, specifically in the traffic flow experiment.

* Although the appendix provides some explanation, additional information about the architectures employed in the experimental section would be beneficial for a more comprehensive understanding of the empirical results.

* The empirical results presented in Table 2 do not provide strong evidence supporting the necessity of approximated equivariance.

**Questions:**

* Definition 1 - What does the J in summation stand for? should it be N?

*  The implementation of this method on an arbitrary graph, which lacks any inductive bias concerning the graph's structure, remains unclear. Are the authors able to offer any insights or intuition regarding a general approach or recipe for graph coarsening in such cases?

**Limitations:**

There is no potential negative impact to this work

---

> ### Author Rebuttal · Authors · 2023-08-08
>
> We thank the reviewer for the constructive feedback and appreciation of our work. We provide detailed responses to the each section point by point:
>
> ### Responses to Weaknesses:
> 1. (Graph coarsening details) We thank the reviewer for raising this point and will add more discussion of graph coarsening methods in the camera-ready version. The clustering choice of the traffic flow experiments are included in the supplementary information (Figure 5) due to the page limit. We also give a detailed explanation of the graph clustering choices in our new set of experiments. We note that theoretically, we assume that the coarse-grained graph approximates the fine graph in cut distance. However, in practice **we construct the coarse-grained versions of the graphs not by minimizing the cut-distance, but rather by using off-the-shelf clustering methods**. In this sense, the theory does not justify the implementation completely rigorously, but rather heuristically motivates it. We will clarify this point in the camera-ready version.
>
> 2. (Experimental details) We now provide a new set of experiments with detailed information of the architectures and set-up in the common response and in the attached PDF. We will also re-organize the main paper to include additional important information of the previous experiments.
>
> 3. (Experiments in Table 2 do not strongly support the theory)
> - To further support our theory, we now provide **a new set of experiments (see attached PDF) that clearly illustrates the benefits of approximate equivariance** (induced from graph coarsening). The details are thoroughly described in the common response, and we give a brief summary below.
> - We consider a regular 2D grid as our fixed graph domain, and images as our graph signals. The goal is to reconstruct the original images given masked images by learning a mapping $f: \mathbb R^N \to \mathbb R^N$, also known as image inpainting. We make use of standard image datasets (MNIST, FashionMNIST) and perform the symmetry model selection via graph coarsening: We cluster the grid into $d \times d$ patches, where $d$ ranges from $1$ (no clustering, corresponding to the trivial symmetry) to $N$ (one giant cluster, corresponding to the $\mathcal{S}\_N$ symmetry). Figure 6 in PDF shows the empirical risk first decreases and then increases as the group decreases, illustrating the bias-variance trade-off from our theory.
> - Note that the 2D grid as a graph has global reflection symmetries (from vertical and horizontal axis), but nodes from a local $d \times d$ patch are not symmetrical in the original grid. Yet enforcing approximate symmetries among nodes in local patches suitably can outperform trivial symmetries, see for example Figure 6 where using $2 \times 2$ patches with the coarsened symmetry ${\cal S}_{4}^{196}$ yields better test error than the trivial symmetry case.
> - We agree that for the traffic flow prediction task, empirical results in Table 2 don’t seem to show strong advantage of approximate equivariant. We want to remark that the traffic flow prediction concerns both the spatial graph domain and the temporal dimension, which is likely to downplay the gain in the spatial domain. We nevertheless include the experiment in the paper since it illustrates the potential to enhance standard GNNs architectures with approximately equivariant modules.
> - In light of the reviewer’s feedback, we plan to replace our original experiments with the new experiments in the main text to clearly illustrate our theory, and defer most of the original experiments to the Appendix.
>
> ### Responses to Questions:
> 1. (Notation) We thank the reviewer for catching this typo and indeed the summand should be N. We will correct this typo in the camera-ready version.
> 2. (Graph coarsening method)
> - Indeed we do not propose a computational method to derive a ground-truth optimal coarsening for general graphs and general tasks. **We envision our approach being used for model selection. Namely, one should implement different coarsening procedures and find the one that works best for the problem**. Our symmetry model selection perspective suggests that a natural coarsening recipe is to sweep from a few large coarsened nodes to many small coarsened nodes. We will clarify this motivation in the camera-ready version.
> - That said, graph clustering is a form of graph coarsening that is very widely studied in the literature and used in practice. For instance, if the graphs are instances of social networks, clustering is a natural approach. Under certain random graph model assumptions, spectral clustering, message-passing algorithms and semidefinite programs enjoy theoretical guarantees, see for example
>   - Lyzinski, Vince, et al. "Perfect clustering for stochastic blockmodel graphs via adjacency spectral embedding." (2014): 2905-2922.
>   - Qin, Tai, and Karl Rohe. "Regularized spectral clustering under the degree-corrected stochastic block model." Advances in neural information processing systems 26 (2013).
>   - Abbe, Emmanuel. "Community detection and stochastic block models: recent developments." The Journal of Machine Learning Research 18.1 (2017): 6446-6531
> - To summarize, in practice we construct the coarse-grained versions of the graphs not by minimizing the cut-distance, but rather by using off-the-shelf clustering methods. Hence, the theory heuristically motivates the implementation.

---

### Official Review · Reviewer_RNWs · 2023-07-01

**Soundness:** 2 fair
**Presentation:** 1 poor
**Contribution:** 3 good
**Rating:** 5
**Confidence:** 3

**Summary:**

This paper formalizes the notion of active symmetries and approximate symmetries of GNNs on a fixed graph domain. Furthermore, it theoretically characterizes the statistical risk of linear regression with symmetries and show a bias-variance tradeoff. For graph tasks, it utilize
coarsed graph for approximate symmetries. Experimental results on human pose estimation and traffic flow prediction valid the proposed method.

**Strengths:**

1. Novel theoretical results and methods. To my best knowledge, its the first risk gap with a combination of symmetries and graphon. The approximating symmetry by coarsened graph is also fancy.

**Weaknesses:**

1. The result in Section 3.1 seems trivial and useless.
2. The experiment section is hard to read as all implementations are put in Appendix.
3. Graph coarsening is vital for the proposed method. But coarsening method for general graph task in unknown.
4. The experiments is conducted on small and specific tasks only rather than general graph task, like link prediction [1].

[1] https://ogb.stanford.edu/docs/linkprop/

**Questions:**

1. As shown in the Weaknesses section, please add a more concrete description of your method in the maintext.
2. Please provide experimental results on more general graph tasks, like link prediction.

**Limitations:**

Yes

---

> ### Author Rebuttal · Authors · 2023-08-08
>
> We thank the reviewer for the comments and appreciation of our theoretical contribution. We provide detailed responses to the each section point by point:
>
> ### Responses to Weaknesses:
> 1. (Result in Section 3.1 seems trivial and useless) We want to remark that considering simplistic models to analyze complex phenomena is very common in machine learning research, and in science in general. **Insights from simplistic models often transfer to the real-life phenomenon. We hence do not see Section 3.1 as useless**. The set-up in Section 3.1 is indeed simple, using a linear regression model with white noise. Yet we hope to make a first step to analyze the symmetry model selection using a very simple setting, such that the results are clear to state and prove, and verifiable through simulation. Although linear regression setting is simple and may seem trivial compared to deep neural networks used in practice, **analysis in linear regression allows us to (1) precisely characterize the bias-variance tradeoff; (2) illustrate the phenomenon we observe in practice for non-linear models**. Moreover, we now added a new set of experiments (see attached PDF), where Figure 6 (left) shows that **the bias-variance tradeoff can be observed both in simple linear models and nonlinear models**.
>
> 2. (Experimental details) We thank the reviewer for the suggestions, and we have supplemented **a new set of experiments which solely use our proposed $\mathcal{G}$-Net without any further augmentation**. The details are thoroughly described in the common response, and we give a brief summary below. We consider a regular 2D grid as our fixed graph domain, and images as our graph signals. The goal is to reconstruct the original images given masked images by learning a mapping $f: \mathbb R^N \to \mathbb R^N$, also known as image inpainting. We make use of standard image datasets (MNIST, FashionMNIST) and perform the symmetry model selection via graph coarsening: We cluster the grid into $d \times d$ patches, where $d$ ranges from $1$ (no clustering, corresponding to the trivial symmetry) to $N$ (one giant cluster, corresponding to the $\mathcal{S}\_N$ symmetry). Figure 6 in PDF shows the empirical risk first decreases and then increases as the group decreases, illustrating the bias-variance trade-off from our theory.
>
> 3. (Graph coarsening method)
> - We agree with the reviewer that there is no ground-truth optimal coarsening for general tasks. Indeed we do not propose a computational method to derive a ground-truth optimal coarsening for general graphs and general tasks. **We envision our approach being used for model selection. Namely, one should implement different coarsening procedures and find the one that works best for the problem**. Our symmetry model selection perspective suggests that a natural coarsening recipe is to sweep from a few large coarsened nodes to many small coarsened nodes. We will clarify this motivation in the camera-ready version.
> - That said, graph clustering is a form of graph coarsening that is very widely studied in the literature and used in practice. For instance, if the graphs are instances of social networks, clustering is a natural approach. Under certain random graph model assumptions, spectral clustering, message-passing algorithms and semidefinite programs enjoy theoretical guarantees, see for example
>   - Lyzinski, Vince, et al. "Perfect clustering for stochastic blockmodel graphs via adjacency spectral embedding." (2014): 2905-2922.
>   - Qin, Tai, and Karl Rohe. "Regularized spectral clustering under the degree-corrected stochastic block model." Advances in neural information processing systems 26 (2013).
>   - Abbe, Emmanuel. "Community detection and stochastic block models: recent developments." The Journal of Machine Learning Research 18.1 (2017): 6446-6531
> - To summarize, in practice we construct the coarse-grained versions of the graphs not by minimizing the cut-distance, but rather by using off-the-shelf clustering methods. Hence, the theory heuristically motivates the implementation.
>
> 4. (More experiments including link prediction) Our experiments are intended to illustrate the learning task on a fixed graph setting. Concretely, we are given a dataset of input/output graph signals supported on a **fixed** graph domain, and aim to learn the best function to map the input signal to output signal. On the other hand, **link prediction** usually aims to predict unobserved edges given a partially observed graph, which **does not fit into our setting in a direct way**. However, we did extend our experiments to more tasks, like image inpainting as explained thoroughly in our common response.
>
> ### Responses to Questions:
> 1. Please refer to the previous Section, Responses 2 (Experimental details)
> 2. Please refer to the previous Section, Responses 4 (More experiments including link prediction)

---

> > ### Comment · Reviewer_RNWs · 2023-08-10
> >
> > Thank you for detailed reply. It solves my concerns with experimental details and graph coarsening methods.I am willing to raise my score to 5.

---

> > > ### Author Response · Authors · 2023-08-11
> > >
> > > Thank you for carefully reading our rebuttal. We are glad that your concerns have been addressed.
> > > We noted that there seemed to be a bug of editing review yesterday, but it has been resolved now.

---

### Official Review · Reviewer_van5 · 2023-07-06

**Soundness:** 3 good
**Presentation:** 3 good
**Contribution:** 3 good
**Rating:** 7
**Confidence:** 3

**Summary:**

The authors discuss the generalization of learning a map that is equivariant to one group, while the ground truth is (approximately) equivariant to another group. Their theoretical analysis first considers the case where the ground truth is exactly equivariant and finds a bias-variance trade-off: making the hypothesis equivariant to a larger group increases bias, while reducing variance. When the hypothesis equivariance group is smaller than the ground-truth group, the bias is 0. Next, the authors define a graph coarsening criterion via graphon theory, that results in a coarse graph $G'$ and whose nodes are a clustering of the nodes of the fine graph $G$. This gives a new symmetry group on the fine graph $G$, called $\mathcal G_{G \to G'}$, which equals the automorphisms of the coarse graph times the permutations of nodes in the clusters. The ground truth graph function is then assumed to be $\epsilon$ close to a function equivariant to $\mathcal G_{G \to G'}$. Theoretically, the authors find a trade-off in a lower bound the risk gap (the difference in test loss between a hypothesis and its symmetrization to the approximate symmetry group). In their experiments, on fixed graphs, the authors use neural networks that combine graph methods on $G$ with methods equivariant to permutations in clusters in the graph. They evaluate different cluster sizes and find an optimum between clusters between size 1 and all nodes clustered together.

**Strengths:**

- Very interesting formalization of approximate graph symmetry via graph coarsening with graphons and the group $\mathcal G_{G \to G'}$ .
- The authors develop novel theoretical results on generalization with approximate symmetries.
- The ideas give an interesting theoretical insight in using clustering and permutation equivariant networks on graphs.
- The paper mostly clearly written.

**Weaknesses:**

- The paper should explain better (in the main paper) why the risk gap is an important criterion for model selection. Lines 183-186 are hard to follow.
- There appears to be a substantial mismatch between the theory and the experiments: the experiments rely substantially on combining equivariant nets to $\mathcal G_{G \to G'}$ with the ( $G$ -automorphism equivariant) graph nets on $G$ . The theory does not discuss this combination of symmetries. Also, the graph cut distance doesn't appear to be computed in practice.
- The theoretical results talk about the symmetry group  $\mathcal G_{G \to G'}$ being a combination of cluster permutations and coarse graph $G'$ automorphisms. However, in their experiments the authors assume those automorphisms to be trivial, so that they only work with cluster permutations. This is a substantially less interesting class of groups, which should be more fairly stated in the paper.
- The coarsening group $\mathcal G_{G \to G'}$ appears quite similar to the groupoid of nodes used in [1], which contains the automorphism group of node neighbourhood (akin to the cluster permutations in this paper) and maps between similar neighbourhoods (akin to the automorphism group of $G'$ ). The authors should consider comparing to this paper.
- The neural network architecture $\mathcal G$ -Nets used in the experiments should be more fully defined in the main paper.

[1] P de Haan and TS Cohen. 2020. “Natural Graph Networks.”

**Questions:**

- Schur's lemma doesn't apply always naively on the real numbers (as they're not algebraically closed.) How do the authors apply the complex results to the real field, as mentioned in line 575 of the appendix? Do they ignore the possible additional intertwiners?
- In section 3, it is not so clear which results rely on the group acting via a permutation of rows (so related to graphs), and which parts could generalize to other groups and representations. Could the authors comment on that?
- In line 168, the authors say that the mismatch term goes below zero, presumably because an inner product goes above zero. However, it is unclear to me why the sign of the inner product is known in general.
- In line 178, the authors make a comment on an inner product of characters increasing when the group increases. Can the authors clarify why this holds in general, or make the comment more specific?

Minor points:
- The authors appear to be confusingly using both the symbols $\Psi$ and $\mathcal Q$ for the symmetrization operation.
- What are the two measures in (3)? Shouldn't these both be the Lebesgue measure? $\mu$ is already used for a measure on $\mathcal X$ .

**Limitations:**

The authors are not quite transparent about the mismatch between their theory and experiments.

---

> ### Author Rebuttal · Authors · 2023-08-08
>
> We thank the reviewer for the critical assessment and constructive feedback of our work. We provide detailed responses to the each section point by point:
>
> ### Responses to Weaknesses:
> 1. (why risk gap) We will add more motivation in Section 3 for the risk gap. To summarize: The risk quantifies how a given model performs on average on any potential input. The smaller the risk, the better the model performs. The risk gap computes the difference of the risk between two models (satisfying different symmetries), and thus allows us to **perform model selection**. Lines 183-186 intend to explain the significance of the risk gap (i.e., choosing different symmetry affects generalization nontrivially in the small $n$ regime).
>
> 2. (mismatch between theory and experiments)
> - We thank the reviewer for pointing this out, and we are providing **a new set of experiments which solely use our proposed $\mathcal{G}$-Net without any further augmentation**. The details are thoroughly described in the common response, and we give a summary below. We consider a regular 2D grid as our fixed graph domain, and images as our graph signals. The goal is to reconstruct the original images given masked images. We perform the symmetry model selection via graph coarsening: We cluster the grid into $d \times d$ patches, where $d$ ranges from $1$ (no clustering, corresponding to the trivial symmetry) to $N$ (one giant cluster, corresponding to the $\mathcal{S}\_N$ symmetry). Figure 6 in PDF shows the empirical risk first decreases and then increases as the group decreases, illustrating the bias-variance trade-off from our theory.
> - We also want to point out that the graph cut distance serves as a theoretical tool for our construction; in practice, we construct the coarse-grained versions of the graphs not by minimizing the cut-distance, but rather by using off-the-shelf clustering methods. In this sense, the theory motivates the implementation but does not justify it completely rigorously. We will clarify this in the camera-ready version.
>
> 3. (experiment assumes trivial coarsened graph symmetry)
> We thank the reviewer for raising this point. We added **new experiments with nontrivial $\mathcal{A}\_{G’}$ symmetries**. Concretely, we consider a global reflection symmetry for the coarsened grid graph of FashionMNIST. We compare the non-trivial coarsened graph symmetry with the trivial case in the inpainting problem (c.f. Figure 6 - right in the PDF). Our results show the utility of using the coarsened graph symmetry, which leads to better performance.
>
> 4. (comparison to NGN)
> We thank the reviewer for these remarks and will include a more detailed comparison to Haan and Cohen’s paper for the camera-ready version. Concretely:
> - Our set-up focuses on learning on a fixed graph and thus graph automorphism, whereas their set-up considers different graphs and thus graph isomorphism.
> - Our goal is to choose the best symmetry for generalization, whereas their goal is to design maximally expressive graph networks.
>
> 5. (experimental details) We now provide a new set of experiments with detailed information on the architectures and set-up in the common response and the attached PDF. We will also re-organize the main paper to include additional important information from the previous experiments.
>
> ### Responses to Questions:
> 1. (Construction of equivariant maps on the reals) We thank the reviewer for raising this question. We didn’t explain the general construction in the appendix of the paper because we didn't need it in the experimental setting. We will add the general construction in the appendix of the revised version for completeness, and outline the key ideas here. By Maschke’s theorem we can decompose the representation in irreducibles over $\mathbb R$. Then we can check further how to decompose these irreducibles over $\mathbb C$, and apply Schur's lemma. We have 3 cases for the decomposition:
> - The irreducible over $\mathbb R$ is also irreducible over $\mathbb C$: We can directly apply Schur’s lemma.
> - The irreducible over $\mathbb R$ decomposes in two different irreducibles over $\mathbb C$: We can send each $\mathbb C$-irreducible to their isomorphic counterpart.
> - The irreducibles over $\mathbb R$ decompose in two copies of the same irreducible over $\mathbb C$: We can send each irreducible to any isomorphic copy independently.
>
> 2. (Generalization to other groups/representations) Yes, our results are built on the results from Elsedy et al. (2021), which **can be used for general compact groups with orthogonal representations**. We believe it may be possible to extend the results to other representations of compact groups. Extending the results to non-compact groups seems much harder. We will comment on this in the camera-ready version.
>
> 3. (Sign of the inner product) We thank the reviewer for raising this point and agree that the original remark in line 168 is incorrect. Indeed, the inner product $\langle f^*, f_G^{\perp} \rangle$ can be negative. We fixed the discussion in line 167-168 to "...when $\cal{G} > \cal{A}_G$, the mismatch term can be positive, negative or zero (depending on $f^*$) whereas the constraint term increases with $\cal{G}$ ".
>
> 4. (Inner product of characters) We thank the reviewer for catching this confusing comment. The correct version of the remark holds in general and it reads as follows:
> The inner product of characters measures the dimension of the linear equivariant space (this is a standard result, but we will add the proof in the appendix for completeness). Thus it decreases as the group increases. Consequently in the risk gap formula that measures the generalization gain, the variance term increases, whereas the bias term decreases (since the projection to the orthogonal complement of $\mathcal{G}$-equivariant space increases, and negates by the minus sign).
>
> 5. Minor points: We apologize for the confusing notations and will fix them in the final version.

---

> > ### Comment · Reviewer_van5 · 2023-08-11
> > **Why only horizontal reflections?**
> >
> > I thank the authors for their rebuttal. Most of my points have been addressed.
> >
> > I have one remaining question though on the new experiment. Why isn't the automorphism group of the square graph, the dihedral group, chosen for $\mathcal A_{\mathcal G}$? Instead, the authors picks a subgroup thereof.

---

> > > ### Author Response · Authors · 2023-08-11
> > > **For simplicity of parameterisation, we consider Abelian groups and full permutation groups in the experiments**
> > >
> > > We thank the reviewer for the question. For simplicity of parameterization, we restrict ourselves to Abelian groups or full permutation groups in the experiments. Thus, we choose the horizontal reflection symmetry, a subgroup of the dihedral group which is *not Abelian*. That said, we note that
> > > - The horizontal reflection symmetry is more meaningful for our image signals (than other elements of the dihedral group such as the diagonal reflection).
> > > - The goal of the experiment aims to show utility of considering nontrivial coarsened graph symmetry (up to our choice, and do not have to be the ground-truth symmetry $\mathcal{A}\_{G'}$)
> > >
> > > We will remark this thoroughly in the camera-ready version.

---

### Official Review · Reviewer_YT16 · 2023-07-07

**Soundness:** 4 excellent
**Presentation:** 3 good
**Contribution:** 4 excellent
**Rating:** 8
**Confidence:** 3

**Summary:**

The authors observe that – while graphs considered as a class have global permutation symmetry – for specific problems, e.g. when learning on a fixed graph, the graph has a much smaller symmetry. Consequently, they attempt to answer the question of how symmetric the model should be in comparison with global permutation symmetry or the graph’s natural symmetries. They quantify this model selection problem as a bias-variance tradeoff, which they validate with numerical experiments.


**Strengths:**

To my knowledge, this paper contains several new ideas. The concept of choosing a supergroup of an object’s natural symmetry group as a statistical strategy motivated by bias-variance trade-off is a novel one in equivariant ML as well as in graph-ML. Similarly, the concept of explicitly considering the natural symmetry of a fixed graph in devising the learning algorithm rather than assuming global permutation symmetry is novel as well. Finally, the concept of using graphon distances to define a notion af approximate equivariance on graphs is yet another novelty that addresses the issue of defining approximate equivariance on a space of objects that is fundamentally discrete.

Methodologically, I found the mathematical approaches taken to be appropriate, and the experiments to be clean tests of the ideas presented. In particular, the authors demonstrate the existence of the proposed bias-variance trade-off in multiple settings.

Finally, I found the author’s argument that the use of overly-equivariant models should be viewed as a statistical strategy that purposefully introduces some systematic error to gain regularity to be a good perspective on the current state of graph neural networks, where people have seen good practical success with permutation-equivariant GNNs on problems that have much lower natural symmetry.


**Weaknesses:**

The paper could do with being run through a spell-check: there are a few obvious typos, e.g. “neighrborhood” in line 289. Moreover, I did not feel like the choice of using graphons to construct a distance between graphs was properly motivated: additional discussion on why the authors chose to use graphons to compare graphs would help motivate the direction of the paper. Is it only to embed the graphs in a continuous space?

**Questions:**

This question might reflect my unfamiliarity with a subset of the literature, but to what extent is the bias-variance tradeoff that the authors propose a perspective that has been used for more general equivariant neural networks where the symmetry of the model is greater than the symmetry of the data? Am I correct in thinking that this perspective on the use of overly-symmetric models is novel for general equivariant-ML, not just for graph ML? If so, maybe this should be noted in the text, e.g. by noting in the discussion that these results should translate to other groups and analyzing this as a future direction.

**Limitations:**

Limitations and potential negative societal impacts are appropriately addressed.

---

> ### Author Rebuttal · Authors · 2023-08-08
>
> We thank the reviewer for carefully examining our work and appreciating the novelty and the impactfulness of our paper. We provide detailed responses to the Weakness and Questions section.
>
> ### Responses to Weaknesses:
> 1. (spell check): We thank the reviewer for pointing this out and will fix all the typos for the camera-ready version.
>
> 2. (motivation of graphon): We thank the reviewer for the question. Graphons are generalizations of graphs in the sense that any graph induces a graphon (see Definition 1 of the main text, also Figure 1 for an illustration). Therefore we can use the cut distance (defined on graphons) as a graph metric. **The cut distance is a natural similarity measure for graphs of different sizes**, as we now explain. By the weak regularity lemma [1], the cut distance between a large graph and a small graph can be interpreted as follows. *Seeing the small graph as a stochastic block model (SBM), the distance is small if the large graph “looks like” it was randomly sampled from the SBM*. The meaning of “looks like” has a precise meaning via the regularity lemma. Namely, the number of edges between any two subsets of nodes of the large graph behave like the expected number of nodes between these sets, if the graph was sampled via the SBM. This is comuted exactly by the cut distance, which is the reason to consider cut distance when working with fine graphons and their coarse-grained versions. We will clarify this motivation in the camera-ready version of the paper and add the above reference.
>
> ### Responses to Questions:
> 1. (Extension to general equivariant ML): We thank the reviewer for raising this important point. Indeed, **our bias-variance tradeoff results can extend to general equivariant machine learning models**. Our current formulation requires that the group is compact and acts on the input/output spaces via an orthogonal representation. We believe the orthogonality requirement can be lifted straightforwardly, but the compactness requirement seems much harder to lift (see discussion in [2]). We will include this in the discussion section and outline interesting future directions.
>
> References:
> 1. Lovász, László, and Balázs Szegedy. "Szemerédi’s lemma for the analyst." GAFA Geometric And Functional Analysis 17 (2007): 252-270.
>
> 2. Villar, Soledad, et al. "Dimensionless machine learning: Imposing exact units equivariance." arXiv preprint arXiv:2204.00887 (2022).

---

### Author Rebuttal · Authors · 2023-08-08

## Common Response

We thank the reviewers for their detailed assessment of our work, and their appreciation for the novelty and the impactfulness of our paper. We are encouraged that all reviewers found that our theoretical results are novel and interesting, particularly on the symmetry bias-variance tradeoff perspective (YT16) and the formulation of approximate symmetries via graphon analysis (van5, RNWs, KAcz), and that our graphon analysis techniques “could have many applications” (KAcz). We are grateful for all the comments and constructive feedback, which will undoubtedly contribute to increasing the overall quality of the paper.

Along with detailed responses to each reviewer, we synthesize below the common questions and responses to all reviewers:
1. Experiments details (van5, RNWs, KAcz); mismatch between theory and practice (van5):

 - **We provided a new set of experiments that more accurately match our theory and illustrate our bias-variance tradeoff perspective**, together with a thorough explanation of the architecture and set-up; see explanations below and PDF attached.

2. Graph coarsening methods and guarantees (RNWs, KAcz):
- We added a more detailed discussion on existing graph coarsening methods based on clustering with proven guarantees (i.e., small cut-distance to the generative graphon)
- We want to highlight that **our symmetry selection perspective** (using symmetry induced from the coarsened graph) **allows us to compare different coarsening procedures, and choose the one that works the best for the target application** (which balances optimally the bias from coarsening error and the variance from constraining the hypothesis class).

3. Extension to general groups (YT16, van5):
- We explained the **applicability of our analysis to general compact groups with orthogonal representation**, and remarked on the possibility of lifting some of the requirements (van5).
- We added a discussion on the implications of our work to general equivariant ML beyond graph ML with pointers to related work (YT16).

### New Experiments

To illustrate our theory, we consider a $28 \times 28$ grid graph as the fixed domain, and grey-scale images as the graph signals. The learning task is to reconstruct the original images given masked} images as inputs (a.k.a *image inpainting*). We investigate the symmetry model selection problem by clustering the grid into $d \times d$ patches, where $d \in \\{28, 14, 7, 4, 2, 1 \\}$. Here $d = 28$ means one cluster (with full permutation symmetry); $d = 1$ is $784$ singleton clusters with no symmetry (trivial).

**Data.** We consider the datasets MNIST and FashionMNIST. For each dataset, we take $100$ training samples and $1000$ test samples via stratified random sampling. The input and output graph signals are $(m_i \odot x_i, x_i)$ ($\odot$ is entrywise multiplication). Here $x_i \in \mathbb R^{28 \times 28} \equiv  \mathbb R^{784}$ denotes the image signals and $m_i$ denotes a random mask (size $14 \times 14$ for MNIST and ${20 \times 20}$ for FashionMNIST). For experiments with *reflection symmetry on the coarsened graph*, we further transform each image in the FashionMNIST subset using horizontal flip with probability $0.5$ (FashionMNIST+hflip).

**Model.**  We consider $2$-layer $\mathcal{G}$-equivariant networks, a composition of $f_{\text{out}} \circ \texttt{ReLU} \circ f_{\text{in}}$, where $f_{\text{in}}, f_{\text{out}}$ denote the input/output linear equivariant layers. We use a hidden size $28$ for all models. $f_{\text{in}}, f_{\text{out}}$ are parameterized as eqn. (1) below. Concretely, any linear equivariant function $f: \mathbb R^N \to \mathbb R^N$ with respect to the symmetry group induced by the coarsening $\mathcal{G} = \mathcal{S}\_{c\_1} \\ldots \mathcal{S}\_{c\_M}  \times \mathcal{A}\_{G'}$ (c.f. Defn 3) admits the following block-matrix form (assuming the nodes are ordered by their cluster assignment) with $f_{kl}$ block matrices, and $a_k, b_k, c_{kl}$ scalars:

$$
 f = \\begin{bmatrix}
        f_{11} &  \\cdots & f_{1M} \\\\
        &   \\ddots &  & \\\\
        f_{M1} &  \\cdots & f_{MM}
       \\end{bmatrix},  \\, f_{kk} = a_k \\mathbf{I} + b_k \\mathbf{1} \\mathbf{1}^{\top}, \\, f_{kl} = c_{kl} \\mathbf{1} \\mathbf{1}^{\top} \\text{ for } k \\neq l.  \\quad (1)
$$

The coarsened graph symmetry $ \mathcal{A}\_{G'}$ induces constraints on $a_k, b_k, c_{kl}$. If $ \mathcal{A}\_{G'}$ is trivial, then these scalars are unconstrained. In the experiment, we consider a reflection symmetry on the coarsened grid graph, i.e., $ \mathcal{A}\_{G'}= \mathcal{S}\_2$ which acts by reflecting the left (coarsened) patches to the right (coarsened) patches. Suppose the reflected patch pairs are ordered consecutively, then $a_k = a_{k+1}, b_k = b_{k+1}$ for $k \in \\{1, 3, \ldots, M-1 \\}$, and $c_{kl} = c_{k+1, l-1}$ for $k \in \\{1, 3, \ldots, M-1 \\}, l \in \\{2, 4, \ldots, M \\}$ (see Figure 8 in PDF for an illustration).

In practice, we extend the formulation to $f: \mathbb R^{N \times d} \to \mathbb R^{N \times k}$.
We train the models with ADAM (learning rate $0.01$, no weight decay, at most $1000$ epochs). We report the best test accuracy at the model checkpoint selected by the best validation accuracy (with a $80/20$ training-validation split).

Figure 6 in PDF shows the **empirical risk first decreases and then increases as the group decreases, illustrating the bias-variance trade-off from our theory**. We perform further ablation studies on the effect of the $\mathcal{G}$-Net architecture and the coarsened graph symmetry $\mathcal{A}\_{G'}$. Figure 6 (left) compares $2$-layer and $1$-layer linear $\mathcal{G}$-Net, **demonstrating that the tradeoff occurs in both linear and nonlinear models**. Figure 6 (right) shows that **using reflection symmetry of the coarsened graph outperforms the trivial symmetry baseline**, highlighting the utility of modeling coarsened graph symmetries.

---

### Decision · Program_Chairs · 2023-09-21

**Decision:**

Accept (poster)

**Comment:**

This paper introduces several novelties in equivariant fixed graph learning: (i) using hypothesis with equivariance group that is super/sub-group of the target equivariance group to adjust a tradeoff between bias and variance; (ii) Using graphon theory to define an approximate symmetry group using coarsening. (iii) Providing risk bounds for approximate equivariant linear regression. (iv) Experiments aiming to validate the theoretical findings.

Some points that would be good to address (to some extent at-least) in the final version: Better illustration of the connection of the theory and experiments; adding missing detail; and provide validation to the practical necessity of certain parts of the theory (i.e., approximate equivariance).